# Fractons, dipole symmetries and curved spacetime

**Leo Bidussi, Jelle Hartong,**[1] **Emil Have,**[2] **Jørgen Musaeus,**[3] **Stefan Prohazka**[4]

*School of Mathematics and Maxwell Institute for Mathematical Sciences,*
*University of Edinburgh, Peter Guthrie Tait Road, Edinburgh EH9 3FD, UK*

*E-mail:* l.bidussi@sms.ed.ac.uk, j.hartong@ed.ac.uk,
emil.have@ed.ac.uk, j.s.musaeus@sms.ed.ac.uk,
stefan.prohazka@ed.ac.uk

ABSTRACT: We study complex scalar theories with dipole symmetry and uncover a no-go theorem that governs the structure of such theories and which, in particular, reveals that a Gaussian theory with linearly realised dipole symmetry must be Carrollian. The gauging of the dipole symmetry via the Noether procedure gives rise to a scalar gauge field and a spatial symmetric tensor gauge field. We construct a world-line theory of mobile objects that couple gauge invariantly to these gauge fields. We systematically develop the canonical theory of a dynamical symmetric tensor gauge field and arrive at scalar charge gauge theories in both Hamiltonian and Lagrangian formalism. We compute the dispersion relation of the modes of this gauge theory, and we point out an analogy with partially massless gravitons. It is then shown that these fractonic theories couple to Aristotelian geometry, which is a non-Lorentzian geometry characterised by the absence of boost symmetries. We generalise previous results by coupling fracton theories to curved space and time. We demonstrate that complex scalar theories with dipole symmetry can be coupled to general Aristotelian geometries as long as the symmetric tensor gauge field remains a background field. The coupling of the scalar charge gauge theory requires a Lagrange multiplier that restricts the Aristotelian geometries.

KEYWORDS: Fractons, non-Lorentzian geometry, Carroll, Aristotelian, Scalar charge gauge theory, Dipole symmetry

---
[1]ORCID: 0000-0003-0498-0029
[2]ORCID: 0000-0001-8695-3838
[3]ORCID: 0000-0003-4869-0293
[4]ORCID: 0000-0002-3925-3983

# 1   Introduction

Fractons [1, 2] are exotic quasiparticles with the distinctive feature of having only limited mobility. They therefore constitute an unfamiliar and fundamental new (theoretical) phase of matter [3, 4]. The bizarre trait of not being able to freely move offers a novel window to widen our understanding of physical (quantum field) theories, gravitational physics [5, 6], holography [7], and might even have applications in the context of quantum information storage [2, 8–10]. For further details and references we refer to the reviews [11, 12].

For some theories the restricted mobility of isolated fracton particles can be viewed as a consequence of conservation of their dipole moment: a point particle of (constant) charge $q$ with a conserved dipole moment $\vec{d} = q\vec{x}$ must remain stationary, $\dot{\vec{x}}(t) = 0$.

For continuum scalar field theories a conserved dipole moment arises from global dipole symmetry which acts infinitesimally on a complex scalar $\Phi(t, \vec{x})$ as $\delta\Phi = i\vec{\beta} \cdot \vec{x}\Phi$. Such a transformation admits an interpretation as a higher moment generalisation of global $U(1)$ invariance, which acts as $\delta\Phi = i\alpha\Phi$. In this language, the dipole moment is the first moment as the transformation is linear in $\vec{x}$. Including even higher moments in the symmetry transformation leads to multipole symmetries [13]. Concretely, a complex scalar theory that describes fracton phases of matter and enjoys dipole symmetry is [14]

$$\mathcal{L} = \dot{\Phi}\dot{\Phi}^{\star} - m^2 \left|\Phi\right|^2 - \lambda(\partial_i\Phi\partial_j\Phi - \Phi\partial_i\partial_j\Phi)(\partial_i\Phi^{\star}\partial_j\Phi^{\star} - \Phi^{\star}\partial_i\partial_j\Phi^{\star})\,. \qquad (1.1)$$

A similar non-Gaussian theory was also encountered in the context of the X-cube model of fracton topological order [15], where they employ a lattice description.

Complex scalar theories with dipole symmetry – including the theory of (1.1) – have two distinctive features: a non-Gaussian term, like the last term in (1.1), and the absence of a $\partial_i \Phi \partial_i \Phi^\star$ term in the action. The absence of the term $\partial_i \Phi \partial_i \Phi^\star$ implies that the free theory, i.e., the one containing only $\dot{\Phi} \dot{\Phi}^\star - m^2 |\Phi|^2$, has no notion of particles in the usual sense. Indeed, we will show that the excitations of the free ungauged theory can be understood as Carrollian particles [16–19], which, like isolated fractons, have the peculiar property that they cannot move. The non-Gaussian term breaks the infinite multipole symmetry of the free theory down to the dipole symmetry and, furthermore, breaks Carroll boost invariance, which means that the Carrollian spacetime symmetry reduces to Aristotelian spacetime symmetry. In fact, as we will demonstrate, a Lagrangian that is polynomial in fields and their derivatives cannot simultaneously be Gaussian, contain spatial derivatives, and have a linearly realised dipole symmetry: assuming two of those properties implies that the third will not hold. For a linearly realised dipole symmetry this leaves on the one hand the case containing spatial derivatives, which is non-Gaussian, like (1.1), and, on the other hand, the case without spatial derivatives which is Gaussian. As discussed above, the latter theories are Carrollian due to a symmetry enhancement that arises when spatial derivatives are absent. Finally, if we demand that the theory is both Gaussian and contains spatial derivatives, the dipole symmetry can no longer be linearly realised and the resulting theories are special cases of Lifshitz field theories with polynomial shift symmetry [20].

Gauging this dipole symmetry requires a purely spatial symmetric tensor gauge field $A_{ij}$ and a scalar $\phi$, which we demonstrate by employing the Noether procedure to gauge the dipole symmetry. This symmetric tensor gauge field can be made dynamical by introducing a suitable gauge invariant action [21, 22], known generically as scalar charge gauge theories. We elucidate the gauge structure of these theories using cohomological tools and we calculate their asymptotic charges. A particularly interesting special case of the scalar charge gauge theory is the *traceless* theory [21, 22], which is independent of the trace of the symmetric tensor gauge field, $\delta_{ij} A_{ij}$. We derive this theory from a new perspective by using a Faddeev–Jackiw type approach (which sidesteps the more elaborate Dirac approach of treating constrained systems) [23, 24]. We also show that the traceless theory only exists in spatial dimensions $d > 2$. Moreover, it turns out that the gauge structure of the symmetric tensor gauge field shares certain similarities with so-called (linearised) partially massless gravitons [25, 26].

The spacetime symmetries of the theories described above are those of absolute spacetime: they are Aristotelian [27]. Aristotelian symmetries consists of spacetime translations and spatial rotations, but do *not* include boosts that mix time and

space. If we were to include a boost symmetry, Aristotelian geometry becomes either Lorentzian, Galilean, or Carrollian, depending on the type of boost that is included in the description.

Coupling field theories to arbitrary geometric backgrounds has the advantage that it allows us to extract currents by varying the geometry (see, e.g., [28–30]). While the coupling of scalar charge gauge theories to curved space (without time) has previously been considered in [31] (see also [32, 33]), the coupling of the complex scalar theory (such as the theory of (1.1)) to curved spacetime has remained an open problem. As we will show, the geometric framework for coupling such fracton theories to curved spacetime is that of Aristotelian geometry [34]. An Aristotelian geometry is described not by a metric but by a 1-form $\tau_\mu$ and a symmetric corank-1 tensor $h_{\mu\nu}$, which respectively measure time and space, as well as their "inverses" $v^\mu$ and $h^{\mu\nu}$.

The coupling of the scalar charge gauge theory to curved spacetime has been studied in the literature [31] for the special case where only the geometry on constant time slices is curved. We repeat this analysis and we find that for $d > 2$ the scalar charge gauge theory can only couple to a curved Riemannian geometry on constant time slices provided that its magnetic sector is traceless, but contrary to claims in the literature, the electric sector does not need to be traceless. Furthermore, we point out that the Riemannian geometry on constant time slices that these special scalar charge gauge theories can couple to are spaces of constant sectional curvature, i.e., they are described by a Riemann tensor of the form

$$R_{ijkl} = \frac{R}{d(d-1)}(h_{ik}h_{jl} - h_{il}h_{jk}),$$  (1.2)

which implies for $d > 2$ (via Schur's lemma) that the Ricci scalar must be constant.[1] We generalise these results to a curved Aristotelian spacetime whose intrinsic torsion vanishes. We find that this is possible if the Riemann tensor of the Aristotelian geometry obeys equation (7.45), which is the Aristotelian generalisation of (1.2). The complex scalar theories, on the other hand, can be coupled to *any* Aristotelian geometry, with the caveat that the (now covariant) symmetric tensor gauge field $A_{\mu\nu}$ and $\phi$ must be background fields, i.e., non-dynamical.

*Note added*: As this manuscript was nearing completion we were made aware of the work [35] which also studies fractons on curved spacetime.

**Organisation**    As an aid to the reader, we here provide an overview of the structure of this document.

In Section 2, we consider a complex field with dipole symmetry. In Section 2.1, we study the global symmetries of a theory with dipole symmetry and work out

---

[1]This follows from the covariant constancy of the Einstein tensor. We thank José Figueroa-O'Farrill for useful discussions on this point.

the general expressions for the associated Noether currents. We then discuss the classification of Lagrangians with linearly realised dipole symmetry in Section 2.2, assuming that the Lagrangian is polynomial in the field and its derivatives. Following this, we derive a no-go theorem in Section 2.4 that tells us that a theory with dipole symmetry cannot simultaneously have linearly realised dipole symmetry, contain spatial derivatives, and be Gaussian. We then discuss the symmetry algebra for a concrete complex scalar theory with dipole symmetry that is very similar to (1.1). We elaborate in Section 2.6 on the connection of these symmetries to a (static) Aristotelian spacetime and discuss some coincidental isomorphisms to Carroll and Bargmann algebras. We end this section with Section 2.7, where we work out the gauging of the global dipole symmetry using the Noether procedure, which shows how the symmetric tensor gauge field emerges.

In Section 3, we couple a worldline action to the scalar charge gauge theory, which we show gives rise to a vanishing total dipole charge (see also [36] for an alternative approach to fracton worldline theories).

In Section 4, we develop the scalar charge gauge theory using a cohomological analysis. Starting in Section 4.1, we work out the Poisson brackets and the generator of gauge transformations, followed by an analysis of the gauge structure in Section 4.2 using generalised differentials. We find it convenient to employ Young tableaux to elucidate the gauge structure. Following this, we work out the the Hamiltonian for scalar charge gauge theory in Section 4.3, which we convert from a phase-space formulation to a configuration space formulation by integrating out the canonical momentum in Section 4.4. We then consider the special case of 3+1 dimensions in Section 4.5. Of special interest is the traceless scalar charge theory, which is independent of the trace of the symmetric tensor gauge field. We develop this from a novel perspective in Section 4.7 using a Faddeev–Jackiw type approach, which a priori suggests the existence of two novel scalar charge gauge theories, which, however, turn out to be field redefinitions of either the traceless or the original theory. We then compute the spectrum of scalar charge gauge theory in Section 4.8. Finally, we comment on similarities between the scalar charge gauge theory and the theory of partially massless gravitons in Section 4.9.

In Section 5, we describe Aristotelian geometry. In Section 5.1, we describe the geometric data that takes the place of a metric in Aristotelian geometry, while connections for Aristotelian geometry are discussed in Section 5.2. Finally, we discuss the procedure of coupling generic field theories to Aristotelian geometry in Section 5.3

We then present one of our main results in Section 6: the coupling of the complex scalar theory with dipole symmetry to an arbitrary Aristotelian geometry.

This is followed by the coupling of the scalar charge gauge theory to Aristotelian geometry in Section 7 which is less straightforward than for the scalar fields. We start this analysis by considering Aristotelian geometries with absolute time for which the

geometry on leaves of constant time are time-independent but further arbitrary Riemannian geometries. In Sections 7.1.1 and 7.1.2, we show how to couple the magnetic and electric sectors to Aristotelian backgrounds that have a curved time-independent Riemannian geometry on leaves of constant time. It is shown that a generic magnetic Lagrangian can only couple gauge invariantly if the Riemannian geometry on constant time slices is flat. If we demand that the magnetic Lagrangian is traceless (i.e., independent of the trace of the symmetric tensor gauge field) then we show that it can be coupled to Riemannian geometries of constant sectional curvature. Furthermore we show that for $d = 2$ spatial dimensions the magnetic sector cannot be traceless as in that case the magnetic Lagrangian vanishes. The conditions on the spatial geometry can be enforced with Lagrange multipliers. Finally, we demonstrate that there are no restrictions on the electric sector, i.e., this part of the Lagrangian can couple to any Riemannian geometry on the leaves of constant absolute time. In Section 7.2 we generalise these results by considering any torsion-free Aristotelian geometry. We end the paper with a discussion in Section 8.

Furthermore, we include three appendices: in Appendix A, which is intended as an aid to the reader, we derive electrodynamics in a similar fashion to how the scalar charge gauge theory is derived in the main text. In Appendix B we provide the details behind our conclusion that the analysis that led to the traceless scalar theory does not lead to any further new scalar charge gauge theories.Finally, in Appendix C, we show that introducing an additional gauge field in an attempt to couple the scalar charge gauge theory to any curved background breaks the dipole symmetry.

**Notation & conventions**   Throughout the manuscript, we employ the following notation: we use $i, j, k, \ldots$ as flat spatial indices, which run from $1, \ldots, d$, where $d$ is the number of spatial dimensions. The index position of the spatial components can be raised and lowered with a Kronecker delta, and we are often cavalier with their position. Greek indices, $\mu, \nu, \rho, \ldots$ are used for curved spacetime indices and run from $0, \ldots, d$. These cannot be raised and or lowered in general. We (anti)symmetrise with weight one, i.e., $T_{(ab)} = \frac{1}{2}(T_{ab} + T_{ba})$ and $T_{[ab]} = \frac{1}{2}(T_{ab} - T_{ba})$. Furthermore, "c.c." stands for complex conjugation and will appear frequently in expressions involving complex scalar fields. The Riemann tensor of an affine connection $\nabla$ is defined via the Ricci identity

$$[\nabla_\mu, \nabla_\nu]X_\rho = R_{\mu\nu\rho}{}^\sigma X_\sigma - 2\Gamma^\rho_{[\mu\nu]}\nabla_\rho X_\sigma \,, \tag{1.3}$$

where $X_\mu$ is any 1-form. The components of the Riemann tensor are

$$R_{\mu\nu\rho}{}^\sigma = -\partial_\mu\Gamma^\rho_{\nu\sigma} + \partial_\nu\Gamma^\rho_{\mu\sigma} - \Gamma^\rho_{\mu\lambda}\Gamma^\lambda_{\nu\sigma} + \Gamma^\rho_{\nu\lambda}\Gamma^\lambda_{\mu\sigma} \,. \tag{1.4}$$

The Ricci tensor is defined as $R_{\mu\rho} = R_{\mu\sigma\rho}{}^\sigma$.

## 1.1 Summary of main results

This is a fairly lengthy paper, so to help guide the reader we provide here a summary of some of our main results. The archetypal field theory with a dipole symmetry (see, for example, the review [12]) consists of a complex scalar field $\Phi$, the dynamics of which is described by the Lagrangian

$$\mathcal{L} = \dot{\Phi}\dot{\Phi}^{\star} - m^2 |\Phi|^2 - \lambda X_{ij} X_{ij}^{\star} , \tag{1.5}$$

where $m$ is the mass of the scalar, and $\lambda$ is a coupling constant. The quantity $X_{ij}$ is given by

$$X_{ij} = \partial_i \Phi \partial_j \Phi - \Phi \partial_i \partial_j \Phi . \tag{1.6}$$

This theory is invariant under the following infinitesimal transformations

$$\delta_\alpha \Phi = i\alpha \Phi \qquad \delta_\beta \Phi = i\beta_i x^i , \tag{1.7}$$

where $\alpha$ and $\beta_i$ are constants. The dipole symmetry may be gauged via the introduction of a symmetric tensor gauge field $A_{ij}$ and a scalar gauge field $\phi$ that transform as $\delta\phi = \dot{\Lambda}$ and $\delta A_{ij} = \partial_i \partial_j \Lambda$, where $\Lambda(t,x)$ is the parameter of the gauge transformation. This leads to the gauge invariant Lagrangian

$$\mathcal{L} = (\partial_t - i\phi)\Phi(\partial_t + i\phi)\Phi^{\star} - m^2 |\Phi|^2 - \lambda \hat{X}_{ij} \hat{X}_{ij}^{\star} , \tag{1.8}$$

where $\hat{X}_{ij} = \partial_i \Phi \partial_j \Phi - \Phi \partial_i \partial_j \Phi + iA_{ij}\Phi^2$ and where the gauge fields are background fields. As we demonstrate in Section 2, the spacetime symmetries are *Aristotelian*: there is no boost symmetry, leaving only spacetime translations and spatial rotations. The appropriate curved geometry to which these theories couple realises the Aristotelian transformations as local tangent space symmetries and is called Aristotelian geometry, which we discuss in detail in Section 5. This geometry is described by geometric fields $(\tau_\mu, h_{\mu\nu}, v^\mu, h^{\mu\nu})$ that satisfy the relations

$$v^\mu \tau_\mu = -1 \qquad v^\mu h_{\mu\nu} = \tau_\mu h^{\mu\nu} = 0 \qquad - v^\mu \tau_\nu + h^{\mu\rho} h_{\rho\nu} = \delta_\nu^\mu . \tag{1.9}$$

From this Aristotelian structure, we can construct a compatible connection $\nabla$ (see equation (5.17) for the connection coefficients of $\nabla$). In terms of this geometry, we may write down the curved generalisation of (1.1), where the complex scalar is coupled to a non-dynamical symmetric tensor gauge field $A_{\mu\nu}$, satisfying $v^\mu A_{\mu\nu} = 0$, as well as a non-dynamical scalar gauge field $\phi$, as

$$\mathcal{L} = e\left[ (v^\mu \partial_\mu \Phi^{\star} - i\phi\Phi^{\star})(v^\nu \partial_\nu \Phi + i\phi\Phi) - m^2 |\Phi|^2 - \lambda h^{\mu\nu} h^{\rho\sigma} \hat{X}_{\mu\rho} \hat{X}_{\nu\sigma}^{\star} \right] , \tag{1.10}$$

where

$$\hat{X}_{\mu\nu} = P^\rho_{(\mu} P^\sigma_{\nu)} (\partial_\rho \Phi \partial_\sigma \Phi - \Phi \nabla_\rho \partial_\sigma \Phi) + iA_{\mu\nu}\Phi^2 . \tag{1.11}$$

In these expressions, $e$ is the Aristotelian analogue of the familiar $\sqrt{-g}$ from Lorentzian geometry, while $P^{\mu}_{\nu} = h^{\mu\rho}h_{\rho\nu}$ is a spatial projector. The curved spacetime Lagrangian is gauge invariant with respect to the curved gauge transformations $\delta\phi = -v^{\mu}\partial_{\mu}\Lambda$, $\delta A_{\mu\nu} = P^{\rho}_{(\mu}P^{\sigma}_{\nu)}\nabla_{\rho}\partial_{\sigma}\Lambda$ and $\delta\Phi = i\Lambda\Phi$.

Until now, the gauge fields have been background fields. To make them dynamical we first introduce the gauge invariant field strengths $F_{ijk} = \partial_i A_{jk} - \partial_j A_{ik}$ and $F_{0ij} = \dot{A}_{ij} - \partial_i\partial_j\phi$. The class of Lagrangians describing these gauge fields are known as scalar charge gauge theories and are the topic of Section 4. The coupling of the traceless scalar charge gauge theory to curved space (but not space*time*) was considered in [31], where they found that the background must be Einstein in $d = 3$ space dimensions. This in turn implies that the 3-dimensional geometry must be a space of constant (sectional) curvature. We generalise their result by showing that it is not necessary for the electric sector of the theory to be traceless in order to couple it to curved space. The restriction to backgrounds that are Einstein implies that, unlike for the complex scalar, we can no longer perform arbitrary background variations. As we explicitly discuss for the case of $(3 + 1)$-dimensional curved space in Section 7.1.1, we can however couple the scalar charge gauge theories to *arbitrary* backgrounds by introducing a Lagrange multiplier $\mathcal{X}_{ij}$ that constrains the spatial geometry to satisfy the Einstein condition. This allows us to perform arbitrary variations of the background while maintaining gauge invariance at the cost of having an additional field in the description. The resulting Lagrangian for $d = 3$ has the form

$$\mathcal{L} = \sqrt{h}\left[\frac{1}{2g_1}h^{ik}h^{jl}F_{0ij}F_{0kl} - \frac{g_2}{g_1(g_1 + 3g_2)}(h^{ij}F_{0ij})^2 \right.$$
$$\left. -\frac{h_1}{4}\left(h^{jm}h^{kn} - h^{jk}h^{mn}\right)h^{il}F_{ijk}F_{lmn} + h_1\left(R^{ij} - \frac{R}{3}h^{ij}\right)\mathcal{X}_{ij}\right], \qquad (1.12)$$

where $g_1, g_2$ and $h_1$ are coupling constants. A few remarks are in order. For $d = 2$ spatial dimensions, the magnetic sector cannot be traceless because if it were it would vanish identically (due to the symmetry properties of $F_{ijk}$). For $d \geq 3$ the traceless magnetic sector can only couple to spaces of constant curvature. In any dimension, if the magnetic sector is not traceless we can only couple to flat space. In any dimension, the electric sector can couple to any Riemannian geometry. We summarised our findings in Table 1.

The coupling to curved space*time* requires the use of Aristotelian geometry. For simplicity, we will restrict to Aristotelian geometries that are torsion-free. Here, the field strengths $F_{0ij}$ and $F_{ijk}$ combine into the following covariant field strength

$$F_{\mu\nu\rho} = \nabla_{\mu}A_{\nu\rho} - \nabla_{\nu}A_{\mu\rho} - 2P^{\sigma}_{\rho}\tau_{[\mu}\nabla_{\nu]}\nabla_{\sigma}\phi. \qquad (1.13)$$

This field strength is not gauge invariant and transforms under gauge transformations as $\delta F_{\mu\nu\rho} = R_{\mu\nu\rho}{}^{\sigma}\partial_{\sigma}\Lambda$, where $R_{\mu\nu\rho}{}^{\sigma}$ is the Riemann tensor of $\nabla$. Provided that the

$(d+1)$-dimensional Aristotelian geometry satisfies a special condition, given in (7.45), the coupling of the Lagrangian (1.12) to such backgrounds takes the form

$$\mathcal{L} = e \left[ \left( \frac{1}{2g_1} h^{\rho\lambda} h^{\sigma\kappa} - \frac{g_2}{g_1(g_1 + dg_2)} h^{\rho\sigma} h^{\lambda\kappa} \right) v^\mu v^\nu F_{\mu\rho\sigma} F_{\nu\lambda\kappa} \right. \tag{1.14}$$
$$\left. + h_1 \left( -\frac{1}{4} h^{\nu\lambda} h^{\rho\kappa} + \frac{1}{2(d-1)} h^{\nu\rho} h^{\lambda\kappa} \right) h^{\mu\sigma} F_{\mu\nu\rho} F_{\sigma\lambda\kappa} \right] .$$

We need to supplement this Lagrangian with the appropriate Lagrange multiplier term that enforces (7.45). We summarised our findings in Table 2.

In addition to the coupling to curved spacetime, we also obtain the following new results:

- We provide a classification of (polynomial) Lagrangians with dipole symmetry by deriving a condition (see (2.20)) that must be satisfied order-by-order in the number of spatial derivatives.

- We remark that a Gaussian theory of a complex scalar with dipole symmetry is also Carrollian.

- We derive a no-go theorem that states that a theory of a complex scalar with a linearly realised dipole symmetry cannot be simultaneously Gaussian and contain gradient terms. For a non-linearly realised dipole symmetry, it is possible to have a theory that is both Gaussian and such that it contains spatial derivatives.

- We derive a novel worldline action that couples the symmetric tensor gauge field to dipoles. This coupling has the form

$$S_{\text{int}} = -q \int_{\lambda_i}^{\lambda_f} d\lambda \left[ \dot{T} \left( \phi - X^i \partial_i \phi \right) - X^i \dot{X}^j A_{ij} \right] , \tag{1.15}$$

  where $q$ is the $U(1)$ charge, and $X^\mu(\lambda) = (T(\lambda), X^i(\lambda))$ are the embedding fields describing the worldline. We expect this to be relevant for the study of Wilson loops of the scalar charge gauge theory.

- Using cohomology we highlight the gauge structure of the scalar gauge theories and provide an exact sequence similar to the gauge structure of electrodynamics and linearised gravity (see Section 4.2). We also point out some similarities and differences with partially massless gravitons (Section 4.9).

- We derive the most general quadratic scalar charge gauge theories whose Hamiltonian is bounded from below. In Hamiltonian form, this Lagrangian reads (in generic dimension)

$$\mathcal{L}[A_{ij}, E_{ij}, \phi] = E_{ij} \dot{A}_{ij} - \mathcal{H} - \phi \partial_i \partial_j E_{ij} , \tag{1.16}$$

where

$$\mathcal{H} = \frac{g_1}{2} E_{ij} E_{ij} + \frac{g_2}{2} E_{ii}^2 + \frac{h_1}{4} F_{ijk} F_{ijk} + \frac{h_2}{2} F_{ijj} F_{ikk} \,, \qquad (1.17)$$

with $g_1 > 0$, $g_1 + dg_2 > 0$, $h_1 \geq 0$ and $h_1 + (d-1)h_2 \geq 0$.

- We determine the modes of the scalar charge gauge theories. For the generic traceful theory, we find $d(d+1)/2 - 1$ independent modes with three characteristic velocities given by

$$\begin{aligned} v_1^2 &= (g_1 + (d-1)g_2)\,(h_1 + (d-1)h_2) \\ v_2^2 &= \frac{1}{2} g_1 (h_1 + h_2) \\ v_3^2 &= g_1 h_1 \,. \end{aligned}$$

## 2 Complex scalar theories with dipole symmetry

A complex scalar field with dipole symmetry describes the fracton phase of matter [14]. The requirement of dipole symmetry restricts the form of the action governing the dynamics of the scalar field, and leads generically to non-Gaussian theories. As we will show, it is possible to obtain Gaussian theories at the expense of linearly realised dipole symmetry or the presence of spatial derivatives. The latter case is an example of a Carrollian theory, while the former is a special case of a Lifshitz field theory with polynomial shift symmetries.

We will then compute and discuss the symmetry algebra for the prototypical complex scalar field theory with dipole symmetry [12, 14, 37], which appears in (2.29). We show using these symmetries that the underlying homogeneous space is a static Aristotelian spacetime.

Finally, we will discuss the Noether procedure for Lagrangians with linearly realised dipole symmetry and explicitly show how the gauging of the dipole symmetry leads to a symmetric tensor gauge field $A_{ij}$ and a scalar gauge field $\phi$.

### 2.1 Symmetries and Noether currents

In this section we begin by studying the Noether currents for a generic complex scalar Lagrangian $\mathcal{L}[\Phi, \dot{\Phi}, \partial_i \Phi, \partial_i \partial_j \Phi, \text{c.c.}]$ (see equation (2.43) for a concrete model). We require the Lagrangian to have $U(1)$ and dipole symmetries which are associated with the following transformations

$$\Phi'(x) = e^{i\alpha} \Phi(x) \qquad\qquad (2.1a)$$

$$\Phi'(x) = e^{i\beta_i x^i} \Phi(x) \,. \qquad\qquad (2.1b)$$

In addition, we require the Lagrangian to be symmetric under temporal translations, spatial translations and spatial rotations given respectively by

$$t' = t + c \qquad x'^i = x^i \qquad \Phi'(x') = \Phi(x) \qquad (2.2a)$$
$$x'^i = x^i + a^i \qquad t' = t \qquad \Phi'(x') = \Phi(x) \qquad (2.2b)$$
$$x'^i = R^i{}_j x^j \qquad t' = t \qquad \Phi'(x') = \Phi(x)\,, \qquad (2.2c)$$

where $R^i{}_j$ is a rotation matrix. While we will not require it, some Lagrangians are also invariant under anisotropic scale transformations

$$t' = b^z t \qquad x'^i = b x^i \qquad \Phi'(x') = b^{D_\Phi} \Phi(x)\,, \qquad (2.3)$$

where the real parameter $z$ is known as the dynamical critical exponent, and $D_\Phi$ is the scaling dimension of $\Phi$. For the first three transformations the Lagrangian transforms as $\mathcal{L}'(x') = \mathcal{L}(x)$ while under scaling it should transform as $\mathcal{L}'(x') = b^{-d-z}\mathcal{L}(x)$ where $d$ is the number of spatial dimensions.

In order to compute the Noether currents we need to work with the infinitesimal version of these transformations. If we take $x'^\mu = x^\mu + \varepsilon \xi^\mu(x) + O(\varepsilon^2)$ and we take $\Phi$ to transform as $\Phi'(x') = \exp\left(\varepsilon f(x)\right)\Phi(x)$ where $f$ is any complex function, then we obtain

$$\delta\Phi(x) = -\xi^\mu(x)\partial_\mu\Phi(x) + f(x)\Phi(x)\,, \qquad (2.4)$$

where we defined $\Phi'(x) = \Phi(x) + \varepsilon\delta\Phi(x) + O(\varepsilon^2)$. Using that the Lagrangian transforms as a density and is defined up to a total derivative term we have a symmetry provided that

$$\delta\mathcal{L} = \partial_\mu\left(-\mathcal{L}\xi^\mu + K^\mu\right) \qquad (2.5)$$

for some vector $K^\mu$. For our set of symmetry transformations the expressions for $\xi^\mu$ and $f$ are

$$
\begin{array}{llll}
\xi^t = 1 & \xi^i = 0 & f = 0 & \text{time translation} \\
\xi^t = 0 & \xi^i = \delta^i_k & f = 0 & \text{space translation in } x^k\text{-dir.} \\
\xi^t = 0 & \xi^i = x^k\delta^i_l - x^l\delta^i_k & f = 0 & \text{rotation in } (x^k, x^l)\text{-plane} \\
\xi^t = zt & \xi^i = x^i & f = D_\Phi & \text{anisotropic dilatation} \\
\xi^t = 0 & \xi^i = 0 & f = i & \text{phase rotation} \\
\xi^t = 0 & \xi^i = 0 & f = ix^k & \text{dipole symmetry in } x^k\text{-dir.}
\end{array} \qquad (2.6)
$$

The indices $k, l$ on the right hand side are fixed and end up as additional indices on the Noether currents.

We now want to compute the conserved currents for each of these symmetries. An arbitrary variation the Lagrangian $\mathcal{L}[\Phi, \dot{\Phi}, \partial_i\Phi, \partial_i\partial_j\Phi, \text{c.c.}]$ is given by

$$\delta\mathcal{L} = \delta\Phi\left[\frac{\partial\mathcal{L}}{\partial\Phi} - \partial_t\frac{\partial\mathcal{L}}{\partial\dot{\Phi}} - \partial_i\frac{\partial\mathcal{L}}{\partial\partial_i\Phi} + \partial_i\partial_j\frac{\partial\mathcal{L}}{\partial\partial_i\partial_j\Phi}\right] + \partial_t\left(\frac{\partial\mathcal{L}}{\partial\dot{\Phi}}\delta\Phi\right)$$
$$+ \partial_i\left[\frac{\partial\mathcal{L}}{\partial\partial_i\Phi}\delta\Phi + \frac{\partial\mathcal{L}}{\partial\partial_i\partial_j\Phi}\partial_j\delta\Phi - \partial_j\frac{\partial\mathcal{L}}{\partial\partial_i\partial_j\Phi}\delta\Phi\right] + \text{c.c.} . \tag{2.7}$$

In this equation the terms in the first bracket are the equation of motion for the Lagrangian. A symmetry transformation leaves the Lagrangian invariant up to a total derivative, i.e.,

$$\delta\mathcal{L} = \partial_\mu\left(-\xi^\mu\mathcal{L} + K^\mu\right) . \tag{2.8}$$

Hence, for variations that are symmetries, and for fields that are on-shell, the Noether current $J^\mu = (J^0, J^i)$ obeys the conservation equation

$$\partial_0 J^0 + \partial_i J^i = 0 \tag{2.9}$$

where

$$J^0 = \left[\frac{\partial\mathcal{L}}{\partial\dot{\Phi}}\delta\Phi + \text{c.c.}\right] + \xi^t\mathcal{L} - K^t \tag{2.10a}$$

$$J^i = \left[\frac{\partial\mathcal{L}}{\partial\partial_i\Phi}\delta\Phi + \frac{\partial\mathcal{L}}{\partial\partial_i\partial_j\Phi}\partial_j\delta\Phi - \partial_j\frac{\partial\mathcal{L}}{\partial\partial_i\partial_j\Phi}\delta\Phi + \text{c.c.}\right] + \xi^i\mathcal{L} - K^i \tag{2.10b}$$

and the c.c. only applies to the terms on the left within the square brackets. The corresponding conserved charge is then given by

$$Q = \int d^d x \; J^0 . \tag{2.11}$$

Since for the Lagrangians that we will end up working with we find that $K^\mu = 0$ for all symmetries, we drop $K^\mu$ from now on.

The energy-momentum tensor is denoted by $T^\mu{}_\nu$. The $\nu = 0$ component corresponds to the Noether current for time translation invariance and the $\nu = k$ component corresponds to the Noether current for space translations invariance in the $x^k$-direction. Under translations we have $\delta\Phi = -\xi^\mu\partial_\mu\Phi = -\delta^\mu_\nu\partial_\mu\Phi = -\partial_\nu\Phi$ and so we find that

$$T^0{}_\nu = -\left[\frac{\partial\mathcal{L}}{\partial\dot{\Phi}}\partial_\nu\Phi + \text{c.c.}\right] + \delta^0_\nu\mathcal{L} \tag{2.12a}$$

$$T^i{}_\nu = -\left[\frac{\partial\mathcal{L}}{\partial\partial_i\Phi}\partial_\nu\Phi + \frac{\partial\mathcal{L}}{\partial\partial_i\partial_j\Phi}\partial_j\partial_\nu\Phi - \partial_j\frac{\partial\mathcal{L}}{\partial\partial_i\partial_j\Phi}\partial_\nu\Phi + \text{c.c.}\right] + \delta^i_\nu\mathcal{L} . \tag{2.12b}$$

We note that the expression for the stress tensor, $T^{ij}$, is in general not symmetric in $i$ and $j$. However, it is well known that Noether currents are only defined up to

improvement terms. In general we are allowed to add any term $X^\mu_{\ \nu}$ satisfying the following off-shell condition

$$\partial_\mu X^\mu_{\ \nu} = 0 \,, \tag{2.13}$$

such that the new current $\tilde{T}^\mu_{\ \nu} = T^\mu_{\ \nu} + X^\mu_{\ \nu}$ still satisfies $\partial_\mu \tilde{T}^\mu_{\ \nu} = 0$. For Lagrangians that can be coupled to curved space, we will always be able to find an $X^\mu_{\ \nu}$ such that $\tilde{T}^{[ij]} = 0$. This is because the stress tensor can be found as the response to varying the Lagrangian with respect to the spatial metric $h_{ij}$ of the curved geometry (in ADM variables) that these theories couple to, and this response is automatically symmetric. We will discuss this coupling to a background geometry in Section 5.

Let us use $\Theta^\mu_{\ \nu}$ to denote the specific choice of improved energy momentum tensor for which $\Theta^{[ij]} = 0$ (on flat space spatial indices are raised and lowered with $\delta^{ij}$ and $\delta_{ij}$). We can then construct a new set of conserved currents $J^\mu_{\ jk}$ given by

$$J^0_{\ jk} = x^j \Theta^0_{\ k} - x^k \Theta^0_{\ j} \tag{2.14a}$$

$$J^i_{\ jk} = x^j \Theta^i_{\ k} - x^k \Theta^i_{\ j} \,, \tag{2.14b}$$

where $\partial_\mu J^\mu_{\ jk} = 0$ follows from the conservation of $\Theta^\mu_{\ \nu}$ as well as $\Theta^{[ij]} = 0$. This will be the conserved current associated with rotations in the $(jk)$-plane.

If the theory under scrutiny is also invariant under anisotropic scale transformations (2.3), the $z$-deformed trace of the appropriately improved energy-momentum tensor $\Theta^\mu_{\ \nu}$ vanishes

$$z\Theta^0_{\ 0} + \Theta^k_{\ k} = 0 \,. \tag{2.15}$$

In section 6 we will show that the coupling to curved space can be done in such a manner that the resulting theory enjoys an anisotropic Weyl symmetry. The Ward identity for this gauge symmetry is given in (5.31), and on flat space this becomes (2.15) on-shell.

This allows us to construct yet another conserved dilatation current $J^\mu_D$ given by

$$J^0_D = zt\Theta^0_{\ 0} + x^k \Theta^0_{\ k} \tag{2.16a}$$

$$J^i_D = zt\Theta^i_{\ 0} + x^k \Theta^i_{\ k} \,, \tag{2.16b}$$

where $\partial_\mu J^\mu_D = 0$ follows from the conservation of $\Theta^\mu_{\ \nu}$ as well as the condition in (2.15). This is the conserved current corresponding to the anisotropic scale symmetry.

The $U(1)$ Noether current for our generic scalar Lagrangian is given by

$$J^0_{(0)} = i\frac{\partial \mathcal{L}}{\partial \dot{\Phi}}\Phi + \text{c.c.} \tag{2.17a}$$

$$J^i_{(0)} = i\frac{\partial \mathcal{L}}{\partial(\partial_i \Phi)}\Phi + i\frac{\partial \mathcal{L}}{\partial(\partial_i \partial_j \Phi)}\partial_j \Phi - i\partial_j\left(\frac{\partial \mathcal{L}}{\partial(\partial_i \partial_j \Phi)}\right)\Phi + \text{c.c.} \,. \tag{2.17b}$$

The Noether current associated with the dipole symmetry can then be expressed as follows

$$J_{(2)}^{0j} = x^j J_{(0)}^0 \tag{2.18a}$$

$$J_{(2)}^{ij} = x^j J_{(0)}^i - \tilde{J}^{ij}, \tag{2.18b}$$

where we defined

$$\tilde{J}^{ij} = \left[ -i \frac{\partial \mathcal{L}}{\partial(\partial_i \partial_j \Phi)} \Phi + \text{c.c.} \right]. \tag{2.19}$$

The current conservation tells us that $J_{(0)}^i = \partial_j \tilde{J}^{ji}$. The latter equation is equivalent to

$$J_{(0)}^j - \partial_i \tilde{J}^{ij} = i\Phi \frac{\partial \mathcal{L}}{\partial(\partial_j \Phi)} + 2i\partial_i \Phi \frac{\partial \mathcal{L}}{\partial(\partial_i \partial_j \Phi)} + \text{c.c.} = 0. \tag{2.20}$$

It can be shown[2] that the latter equation is nothing but the condition that the Lagrangian viewed as a function of $\rho$ and $\theta$, where $\Phi = \frac{1}{\sqrt{2}} \rho e^{i\theta}$, does not depend on $\partial_i \theta$.

It can be shown that equation (2.20) holds off shell. The Lagrangian is invariant under both a global $U(1)$ transformation and a dipole transformation, i.e., under $\delta \Phi = i \left( \alpha + \beta_k x^k \right) \Phi$. This means that we have

$$\delta \mathcal{L} = \left( \alpha + \beta_k x^k \right) \left[ i\Phi \frac{\partial \mathcal{L}}{\partial \Phi} + i\dot{\Phi} \frac{\partial \mathcal{L}}{\partial \dot{\Phi}} + i\partial_i \Phi \frac{\partial \mathcal{L}}{\partial \partial_i \Phi} + i\partial_i \partial_j \Phi \frac{\partial \mathcal{L}}{\partial \partial_i \partial_j \Phi} + \text{c.c} \right]$$
$$+ \beta_i \left[ i\Phi \frac{\partial \mathcal{L}}{\partial(\partial_i \Phi)} + 2i\partial_j \Phi \frac{\partial \mathcal{L}}{\partial(\partial_i \partial_j \Phi)} + \text{c.c} \right] = 0. \tag{2.21}$$

Using that this must vanish off shell for $\beta_i = 0$ and $\alpha \neq 0$ as well as for $\alpha = 0$ and $\beta_i \neq 0$ we obtain equation (2.20).

## 2.2   Classification of Lagrangians with linear dipole symmetry

We will assume that the Lagrangian is polynomial in the fields and derivatives of the fields. The classification problem for such theories with linear dipole symmetry amounts to finding the most general polynomial solution to (2.21).

---

[2]To show this consider

$$\mathcal{L}(\Phi, \dot{\Phi}, \partial_i \Phi, \partial_i \partial_j \Phi, \text{c.c}) = \tilde{\mathcal{L}}(\rho, \dot{\rho}, \dot{\theta}, \partial_i \rho, \partial_i \partial_j \rho, \partial_i \partial_j \theta),$$

and vary both sides

$$\frac{\partial \mathcal{L}}{\partial \Phi} \delta \Phi + \frac{\partial \mathcal{L}}{\partial \dot{\Phi}} \delta \dot{\Phi} + \frac{\partial \mathcal{L}}{\partial \partial_i \Phi} \delta \partial_i \Phi + \frac{\partial \mathcal{L}}{\partial \partial_i \partial_j \Phi} \delta \partial_i \partial_j \Phi + \text{c.c.} =$$

$$\frac{\partial \tilde{\mathcal{L}}}{\partial \rho} \delta \rho + \frac{\partial \tilde{\mathcal{L}}}{\partial \dot{\rho}} \delta \dot{\rho} + \frac{\partial \tilde{\mathcal{L}}}{\partial \dot{\theta}} \delta \dot{\theta} + \frac{\partial \tilde{\mathcal{L}}}{\partial \partial_i \rho} \delta \partial_i \rho + \frac{\partial \tilde{\mathcal{L}}}{\partial \partial_i \partial_j \rho} \delta \partial_i \partial_j \rho + \frac{\partial \tilde{\mathcal{L}}}{\partial \partial_i \partial_j \theta} \delta \partial_i \partial_j \theta.$$

Next use $\Phi = \frac{1}{\sqrt{2}} \rho e^{i\theta}$ in the variations on the left hand side and collect all terms proportional to $\delta \partial_i \theta$. Since the right hand side, by assumption, does not contain such terms these terms must add up to zero. This is precisely the condition (2.20).

For theories that are second order in time derivatives we find Lagrangians of the form

$$\mathcal{L} = \dot{\Phi}\dot{\Phi}^\star - V(|\Phi|^2) + \mathcal{L}^{[2]} + \mathcal{L}^{[4]} + \cdots , \tag{2.22}$$

where $\mathcal{L}^{[2]}$ and $\mathcal{L}^{[4]}$ contain the most general terms that are quadratic and quartic in spatial derivatives, respectively. The dots denote terms that are higher order in spatial derivatives. If we wish to consider theories that are first order in time derivatives we need to replace the kinetic term with $i\Phi^\star\dot{\Phi} + $ c.c.

For example at second order in spatial derivatives we can make the ansatz

$$\mathcal{L}^{[2]} = c_1\Phi^{\star 2}\partial_i\Phi\partial_i\Phi + c_1^\star\Phi^2\partial_i\Phi^\star\partial_i\Phi^\star + c_2\partial_i\Phi\partial_i\Phi^\star + c_3\Phi^\star\partial_i\partial_i\Phi + c_3^\star\Phi\partial_i\partial_i\Phi^\star , \tag{2.23}$$

where $c_1$ and $c_3$ are complex-valued functions of $|\Phi|^2$ and $c_2$ is a real-valued function of $|\Phi|^2$. This Lagrangian is manifestly $U(1)$ invariant. Solving (2.20) leads to

$$c_3 = -c_1|\Phi|^2 + \frac{c_2}{2} . \tag{2.24}$$

Hence we find

$$\mathcal{L}^{[2]} = \left[c_1\Phi^{\star 2}\left(\partial_i\Phi\partial_i\Phi - \Phi\partial_i\partial_i\Phi\right) + \text{c.c.}\right] + c_2\left(\partial_i\Phi\partial_i\Phi^\star + \frac{1}{2}\Phi^\star\partial_i\partial_i\Phi + \frac{1}{2}\Phi\partial_i\partial_i\Phi^\star\right) . \tag{2.25}$$

If we take $c_2$ to be a constant then the $c_2$-term is a total derivative. Hence, $c_2$ must be of order $|\Phi|^2$, while $c_1$ is $O(1)$. It follows that $\mathcal{L}^{[2]}$ is not Gaussian. Using partial integration the $c_2$ term can be written as

$$-\frac{1}{2}c_2'\partial_i|\Phi|^2\partial_i|\Phi|^2 , \tag{2.26}$$

where the prime denotes differentiation with respect to $|\Phi|^2$. Looking at the Hamiltonian we see that the $c_1$ term is not bounded from below while the $c_2'$ term is bounded from below.

Using similar methods, we can write down an expression for the most general expression that is quartic in spatial derivatives. Instead of working out this most general expression, we will work with the following expression

$$\mathcal{L}^{[4]} = -\lambda X_{ij}X_{ij}^\star - \tilde{\lambda}X_{ii}X_{jj}^\star , \tag{2.27}$$

where we defined

$$X_{ij} = \partial_i\Phi\partial_j\Phi - \Phi\partial_i\partial_j\Phi , \tag{2.28}$$

and where $\lambda$ and $\tilde{\lambda}$ are real parameters. This Lagrangian satisfies (2.20) for any values of $\lambda$ and $\tilde{\lambda}$, and the associated Hamiltonian is bounded from below for $\lambda \geq 0$ and $\lambda + d\tilde{\lambda} \geq 0$.

.

Combining this choice of $\mathcal{L}^{[4]}$ with the kinetic term above leads to Lagrangians that are reminiscent of some that have previously been considered in the literature [12, 14]

$$\mathcal{L} = \dot\Phi\dot\Phi^\star - m^2\,|\Phi|^2 - \lambda X_{ij}X_{ij}^\star - \tilde\lambda X_{ii}X_{jj}^\star\,, \tag{2.29}$$

where $m$ is the mass of the complex scalar.

## 2.3 Symmetry enhancement

An interesting sub-case for the class of Lagrangians described in section 2.1 is when there is additional symmetry in the form of the transformation $\delta\Phi = \frac{i}{2}\gamma x^2\Phi$, where $\gamma$ is the transformation parameter. This gives rise to the conservation of the trace of the quadrupole moment. Later on we will see that the gauging of this type of Lagrangians will lead to a symmetric and traceless tensor gauge field $A_{ij}$, where the tracelessness is due to this extra symmetry.

Using equations (2.10a) and (2.10b) we find the following expression for the Noether current associated with the $\gamma$-transformation

$$J_{(4)}^0 = \frac{1}{2}x^2 J_{(0)}^0 \tag{2.30a}$$

$$J_{(4)}^i = \frac{1}{2}x^2 J_{(0)}^i - x^j\,\tilde J^{ij}\,. \tag{2.30b}$$

From this we indeed see that the Noether charge associated with this is the trace of the quadrupole moment. Furthermore, if we write out the current conservation equation we get the following condition

$$0 = \partial_0 J_{(4)}^0 + \partial_i J_{(4)}^i = -\tilde J^{ii}\,, \tag{2.31}$$

where we used the conservation of the $U(1)$-current as well as the condition in (2.20).

It can be shown that the condition in (2.31) is the condition for the Lagrangian to have this additional $\gamma$-symmetry and thus holds off-shell. Specifically, if we require the Lagrangian to be invariant under $\delta\Phi = i(\alpha + x^k\beta^k + \frac{1}{2}\gamma x^2)$ we get

$$\begin{aligned}
\delta\mathcal{L} &= \left(\alpha + \beta_k x^k + \frac{1}{2}\gamma x^2\right)\left[i\Phi\frac{\partial\mathcal{L}}{\partial\Phi} + i\dot\Phi\frac{\partial\mathcal{L}}{\partial\dot\Phi} + i\partial_i\Phi\frac{\partial\mathcal{L}}{\partial\partial_i\Phi} + i\partial_i\partial_j\Phi\frac{\partial\mathcal{L}}{\partial\partial_i\partial_j\Phi} + \text{c.c.}\right] \\
&\quad + \left(\beta_i + \gamma x^i\right)\left[i\Phi\frac{\partial\mathcal{L}}{\partial(\partial_i\Phi)} + 2i\partial_j\Phi\frac{\partial\mathcal{L}}{\partial(\partial_i\partial_j\Phi)} + \text{c.c.}\right] \\
&\quad + \gamma\left[i\Phi\frac{\partial\mathcal{L}}{\partial(\partial_i\partial_i\Phi)} + \text{c.c.}\right] = 0\,.
\end{aligned} \tag{2.32}$$

Using that this must vanish off-shell for arbitrary $\alpha$, $\beta_i$ and $\gamma$ we recover (2.20) as well as the condition (2.31).

## 2.4 No-go theorem

So far we have discussed non-Gaussian theories with spatial derivatives and linearly realised dipole symmetries, c.f., (2.1b). A theory is Gaussian if its Lagrangian is quadratic in the fields whose kinetic terms are canonically normalised. In our case this is the field $\Phi$. Additionally, we restrict our attention to Lagrangians that are polynomial in $\Phi$ and derivatives thereof. In this case, a Gaussian complex scalar with linearly realised dipole symmetry is either of the form

$$\mathcal{L} = \frac{i}{2}(\Phi\dot{\Phi}^\star - \Phi^\star\dot{\Phi}) - V(|\Phi|^2) \tag{2.33}$$

or

$$\mathcal{L} = \dot{\Phi}\dot{\Phi}^\star - V(|\Phi|^2) \tag{2.34}$$

depending on whether one wants first or second order time derivatives in the equations of motion. If we demand that the theory be Gaussian the potential is, up to an insignificant constant, a mass term $V = m^2\Phi\Phi^\star$. Due to the linearly realised dipole symmetry a gradient term $(\partial_i\Phi)(\partial_i\Phi^\star)$ is disallowed. These Gaussian models with linearly realised dipole symmetry are Carrollian. This means their spacetime symmetries are enhanced by a Carroll boost symmetry

$$t' = t + b_i x^i \qquad\qquad x'^i = x^i \qquad\qquad \Phi'(x') = \Phi(x) \tag{2.35}$$

which, with $\delta\Phi(x) = -\xi^\mu\partial_\mu\Phi(x) + f(x)\Phi(x)$, is given infinitesimally by

$$\xi^t = x^k \quad\quad \xi^i = 0 \quad\quad f = 0 \quad\quad \text{Carroll boost in direction } x^k. \tag{2.36}$$

These Carroll boosts are actually part of the more general symmetries $\delta\Phi(x) = \xi^t(x^i)\partial_t\Phi(x)$, where $\xi^t(x^i)$ is an arbitrary real function of the spatial coordinates. Additionally, we have a second tower of infinite-dimensional symmetries whereby we can rotate $\Phi$ with a phase that, again, is any local function of the spatial coordinates.[3] If we expand $\Phi$ in Fourier modes (assuming a quadratic potential $V = m^2\Phi\Phi^\star$) then the modes have a fixed energy $E = m$, i.e., no dispersion relation, so these modes (particles) are not propagating in space. To show this we compute the retarded

---

[3]This means that this Gaussian (or free) theory without coupling admits an infinite dimensional BMS-like [38, 39] symmetry algebra, c.f., also [13]. More precisely, this algebra is a semidirect sum of an Euclidean algebra spanned by the rotations and translations extended by two infinite dimensional abelian Lie algebras, the "supertranslations". They have the unfamiliar feature that the two "supertranslations" do not commute with the actual translations, but their polynomial order gets reduced by them. For example, the action of the translations $P_i$ on the first order Carroll boosts $B_i$ leads to the zeroth order time translations, $\{P_i, B_j\} = \delta_{ij}H$. We remark that a point particle with a conserved dipole moment $\vec{d} = q\vec{x}$ also has conserved higher-pole moments and thus an infinite symmetry.

propagator for the second order time derivative theory with $V = m^2 \Phi \Phi^\star$ which is proportional to

$$\lim_{\epsilon \to 0} \int dE d^d \vec{p} \, \frac{e^{i(Et - \vec{p} \cdot \vec{x})}}{(E - i\epsilon)^2 - m^2} \,. \tag{2.37}$$

It is of the form $f(t)\delta(\vec{x})$ since there is no momentum dependence in the denominator. Hence there is only propagation in time and not in space. This result is Carroll boost invariant [18].

The above shows that any theory of a complex scalar with a linearly realised dipole symmetry cannot simultaneously be Gaussian and contain spatial derivatives (i.e., gradient terms). If we allow for a non-linearly realised dipole symmetry, it is possible to build theories that are both Gaussian and contain spatial derivatives, as we illustrate with the following example. It is well-known that the Lagrangian (2.29) is non-Gaussian. Consider now the case where the potential in (2.34) is of Mexican hat form

$$V = g \left( |\Phi|^2 - \frac{v^2}{2} \right)^2 \,, \tag{2.38}$$

where $g$ and $v$ are real constants. Around the false vacuum $\Phi = 0$ the theory is non-Gaussian but if we expand around the true vacuum $|\Phi| = v/\sqrt{2}$ and ignore higher order terms the theory becomes Gaussian with a non-linearly realised dipole symmetry. To see this consider

$$\mathcal{L} = \dot{\Phi}\dot{\Phi}^\star - \lambda X_{ij} X_{ij}^\star - g \left( |\Phi|^2 - \frac{v^2}{2} \right)^2 \tag{2.39}$$

$$= \frac{1}{2}\dot{\rho}^2 + \frac{1}{2}\rho^2 \dot{\theta}^2 - \frac{\lambda}{4} \partial_i \partial_j \rho \partial_i \partial_j \rho$$

$$+ \frac{\lambda}{2} \rho \partial_i \rho \partial_j \rho \partial_i \partial_j \rho - \frac{\lambda}{4} \rho^2 \partial_i \partial_j \rho \partial_i \partial_j \rho - \frac{\lambda}{4} \rho^4 \partial_i \partial_j \theta \partial_i \partial_j \theta - \frac{g}{4} \left( \rho^2 - v^2 \right)^2 \,, \tag{2.40}$$

where $X_{ij}$ is defined in (2.28), and where we used $\Phi = \frac{1}{\sqrt{2}}\rho e^{i\theta}$ and expanded around $\rho = v > 0$ by defining $\rho = v + \eta$. If we keep only quadratic terms in $\eta$ and $\theta$ we find

$$\mathcal{L} = \frac{1}{2}\dot{\eta}^2 + \frac{1}{2}v^2 \dot{\theta}^2 - \frac{\lambda}{4}v^2 \partial_i \partial_i \eta \partial_j \partial_j \eta - \frac{\lambda}{4}v^4 \partial_i \partial_i \theta \partial_j \partial_j \theta - g v^2 \eta^2 \,, \tag{2.41}$$

where we performed some partial integrations. The fields $\eta$ and $\theta$ now have canonically normalised kinetic terms. This is a theory of Lifshitz type with polynomial shift symmetries which can be seen as a non-linear realisation of the dipole symmetry. The field $\theta$ is a Lifshitz Goldstone boson and the field $\eta$ is a massive Lifshitz scalar. The non-linear (in $\theta$) symmetry is explicitly given by

$$\delta\theta = \alpha + \beta_i x^i \,, \tag{2.42}$$

where $\alpha$ is the constant shift symmetry of conventional Goldstone bosons and $\beta_i$ parametrises the dipole symmetry which is, up to the exclusion of the time dimension, also reminiscent of the symmetries of the Galileon [40].

Based on the result of this section and Section 2.1, we conclude that the following three properties cannot all hold at the same time (for Lagrangians that are polynomial in the fields and their derivatives):

1. linear dipole symmetry

2. spatial derivatives

3. Gaussian

If you assume any two of these the remaining property does not hold. To summarise: if 1. and 2. hold we have linear dipole symmetry and spatial derivatives at the expense of obtaining non-Gaussian theories, like the fractonic ones of this work, see, e.g., (2.43). When 1. and 3. hold we have Gaussian theories with linear dipole symmetry, however spatial derivatives are forbidden, and the theory acquires a Carrollian symmetry. For the case where 2. and 3. hold we have a Gaussian theory and spatial derivatives however in that case we cannot have a linear dipole symmetry. What is still possible is for the dipole symmetry to be nonlinearly realised. These are special cases of Lifshitz field theories with polynomial shift symmetry, like the one we have discussed.

## 2.5   Symmetry algebra

In this section we want to compute the symmetry algebra for the following anisotropic scale invariant Lagrangian

$$\mathcal{L} = \dot{\Phi}\dot{\Phi}^\star - \lambda X_{ij} X_{ij}^\star - \tilde{\lambda} X_{ii} X_{jj}^\star , \qquad (2.43)$$

where $X_{ij}$ is defined in (2.28). This theory has scale symmetry (2.3) with dynamical critical exponent $z = \frac{d+4}{3}$ and the scaling dimension of $\Phi$ given by $D_\Phi = \frac{2-d}{3}$. Unless both $\lambda$ and $\tilde{\lambda}$ vanish it is also non-Gaussian.

To obtain the charges for the Lagrangian in (2.43) and compute their Poisson brackets we use the Hamiltonian formulation. We start by defining the canonical momenta $\Pi$ and $\Pi^\star$ of $\Phi$ and $\Phi^\star$ by

$$\Pi = \frac{\partial \mathcal{L}}{\partial \dot{\Phi}} = \dot{\Phi}^\star \qquad\qquad \Pi^\star = \frac{\partial \mathcal{L}}{\partial \dot{\Phi}^\star} = \dot{\Phi} \qquad (2.44)$$

which we use to obtain the canonical Hamiltonian density

$$\mathcal{H} = \Pi\Pi^\star + \lambda X_{ij} X_{ij}^\star + \tilde{\lambda} X_{ii} X_{jj}^\star , \qquad (2.45)$$

which is bounded from below for $\lambda \geq 0$ and $\lambda + d\tilde{\lambda} \geq 0$. The Lagrangian density in Hamiltonian form is then

$$\mathcal{L}^\mathcal{H} = \Pi\dot{\Phi} + \Pi^\star\dot{\Phi}^\star - \mathcal{H} \qquad (2.46)$$

from which we can read off the equal time Poisson brackets

$$\{\Phi(x), \Pi(y)\} = \delta(x - y) \qquad \{\Phi^\star(x), \Pi^\star(y)\} = \delta(x - y) \,. \tag{2.47}$$

Next we want to compute the Noether charges associated with the symmetries of the Lagrangian. We use the expression we found in equation (2.10a) to find the following set of Noether charges

$$Q^{(0)} = \int d^d x \, J^0_{(0)} \tag{2.48a}$$

$$Q^{(2)}_i = \int d^d x \, x^i J^0_{(0)} \tag{2.48b}$$

$$P_i = \int d^d x \, \mathcal{P}_i \tag{2.48c}$$

$$M_{ij} = \int d^d x \, (x^i \mathcal{P}_j - x^j \mathcal{P}_i) \tag{2.48d}$$

$$H = \int d^d x \, \mathcal{H} \tag{2.48e}$$

$$D = \int d^d x \, (zt\mathcal{H} + x^i \mathcal{P}_i - D_\Phi \, (\Phi\Pi + \Phi^\star\Pi^\star)) \,, \tag{2.48f}$$

where

$$J^0_{(0)} = i \, (\Phi\Pi - \Phi^\star\Pi^\star) \tag{2.49a}$$

$$\mathcal{P}_i = \Pi\partial_i\Phi + \Pi^\star\partial_i\Phi^\star \,, \tag{2.49b}$$

are the charge density and momentum density, respectively. Starting from the top we have the $U(1)$ charge, the dipole charge, the momentum, the angular momentum, the energy and the dilatation charge.

It is interesting to note that for general values of $D_\Phi$ and $d$, the Poisson bracket $\{D, H\}$ is given by the following expression

$$\{D, H\} = \int d^d x \big[ -2(d - 2 + 3D_\Phi)\Pi\Pi^\star + (d - 4 + 4D_\Phi)\mathcal{H} \big]. \tag{2.50}$$

In order for this to be proportional to $H$, and thus for the algebra to close, we need $D_\Phi = -\frac{d-2}{3}$, and so we obtain

$$\{D, H\} = -\frac{d+4}{3} H \,. \tag{2.51}$$

The prefactor of $\frac{d+4}{3}$ is exactly the value of $z$ for which the theory is scale invariant. Ultimately one finds the following nonzero Poisson brackets

$$\{M_{ij}, M_{kl}\} = -4\delta_{[k[i}M_{j]l]} \qquad \{M_{jk}, P_i\} = -2\delta_{i[j}P_{k]} \tag{2.52a}$$

$$\{P_i, Q^{(2)}_j\} = \delta_{ij}Q^{(0)} \qquad \{M_{jk}, Q^{(2)}_i\} = -2\delta_{i[j}Q^{(2)}_{k]} \tag{2.52b}$$

$$\{D, H\} = -zH \qquad \{D, P_i\} = -P_i \qquad \{D, Q^{(2)}_i\} = Q^{(2)}_i \,, \tag{2.52c}$$

where $z = \frac{d+4}{3}$.

If we set $\tilde{\lambda} = -\lambda/d$ we get a symmetry enhancement. Namely, the Lagrangian is invariant under $\delta\Phi = \frac{i}{2}\gamma x^2 \Phi$ which leads to the following Noether charge

$$Q^{(4)} = \frac{1}{2}\int d^d x \, x^2 J^0_{(0)} \,. \tag{2.53}$$

This can be thought of as the trace of the quadrupole moment. It has the following nonzero Poisson brackets

$$\{P_i, Q^{(4)}\} = -Q^{(2)}_i \qquad\qquad \{D, Q^{(4)}\} = 2Q^{(4)} \,. \tag{2.54}$$

Let us remark that the charges (2.48a)–(2.48e), and by extension the algebra in (2.52a) and (2.52b), always take this form for any complex scalar theory that is second order in time derivatives with dipole and $U(1)$ symmetries. Within this class of theories, some Lagrangians, such as (2.43), will also have dilatation symmetry.

**Other symmetries?**

In this subsection, we work out the most general conditions that a manifest, i.e., linearly realised (in field space) symmetry of the form (2.4) must satisfy. For a variation of this form to be a symmetry, it must be such that the Lagrangian varies as in (2.8), which for the specific Lagrangian (2.43) leads to the condition

$$\begin{aligned}
\partial_\mu K^\mu = \frac{1}{2}&(X_{ij}X^\star_{kl} + X_{kl}X^\star_{ij}) \tag{2.55}\\
&\times \left(\lambda\delta_{jl}\left(4\partial_{(i}\xi_{k)} - \delta_{ik}(4\mathrm{Re}f + \partial_\mu\xi^\mu)\right) + \tilde{\lambda}\delta_{kl}\left(4\partial_{(i}\xi_{j)} - \delta_{ij}\left(4\mathrm{Re}f + \partial_\mu\xi^\mu\right)\right)\right)\\
&- \dot{\xi}^i(\dot{\Phi}^\star\partial_i\Phi + \dot{\Phi}\partial_i\Phi^\star) + \dot{\Phi}^\star\dot{\Phi}\left(\partial_\mu\xi^\mu - 2\dot{\xi}^t + 2\mathrm{Re}f\right) + \Phi\dot{\Phi}^\star\dot{f} + \dot{\Phi}\Phi^\star\dot{f}^\star\\
&+ (\Phi\partial_\mu\Phi X^\star_{ij} + \Phi^\star\partial_\mu\Phi^\star X_{ij})\left(\lambda\partial_i\partial_j\xi^\mu + \tilde{\lambda}\delta_{ij}\partial^2\xi^\mu\right)\\
&+ \Phi^2 X^\star_{ij}(\lambda\partial_i\partial_j f + \tilde{\lambda}\delta_{ij}\partial^2 f) + \Phi^{\star 2}X_{ij}(\lambda\partial_i\partial_j f^\star + \tilde{\lambda}\delta_{ij}\partial^2 f^\star)\\
&+ 2\left((\dot{\Phi}\partial_j\Phi - \Phi\partial_j\dot{\Phi})X^\star_{kl} + (\dot{\Phi}^\star\partial_j\Phi^\star - \Phi^\star\partial_j\dot{\Phi}^\star)X_{kl}\right)(\lambda\delta_{jl}\partial_k\xi^t + \tilde{\lambda}\delta_{kl}\partial_j\xi^t) \,.
\end{aligned}$$

A symmetry with a nonzero $K^\mu$ transforms the action into a boundary term, and we will not consider this case. As we saw above, all the transformations in (2.6) led to

$K^\mu = 0$. Using equation (2.55) we find that a symmetry with $K^\mu = 0$ must satisfy

$$0 = \lambda \left( 2\delta_{(l(j}\partial_{i)}\xi_{k)} + 2\partial_{(k}\xi_{(i}\delta_{j)l)} - \delta_{(l(j}\delta_{i)k)}(4\text{Re}f + \partial_\mu\xi^\mu) \right)$$
$$+ \tilde{\lambda}\delta_{kl}\left( 4\partial_{(i}\xi_{j)} - \delta_{ij}(4\text{Re}f + \partial_\mu\xi^\mu) \right) \tag{2.56a}$$

$$0 = \partial_i\xi^i - \dot{\xi}^t + 2\text{Re}f \tag{2.56b}$$

$$0 = \lambda\partial_i\partial_j\xi^\mu + \tilde{\lambda}\delta_{ij}\partial^2\xi^\mu \tag{2.56c}$$

$$0 = \lambda\partial_i\partial_j f + \tilde{\lambda}\delta_{ij}\partial^2 f \tag{2.56d}$$

$$0 = \dot{f} \tag{2.56e}$$

$$0 = \dot{\xi}^i \tag{2.56f}$$

$$0 = \lambda\delta_{j(l}\partial_{k)}\xi^t + \tilde{\lambda}\delta_{kl}\partial_j\xi^t \,. \tag{2.56g}$$

The solutions to these equations split into two cases: when $\lambda + d\tilde{\lambda} \neq 0$, the equations above tells us that the most general symmetry is such that

$$f = -\frac{d-2}{3}f_0 + i\alpha + i\beta_i x^i \tag{2.57a}$$

$$\xi^i = \xi_0^i + \omega^i{}_j x^j + f_0 x^i \tag{2.57b}$$

$$\xi^t = \xi_0^t + \frac{d+4}{3}f_0 t \,, \tag{2.57c}$$

where $\{f_0, \alpha, \beta_i, \xi_0^i, \xi_0^t\}$ are real constants, and $\omega^i{}_j$ is a real antisymmetric matrix. We see that this exactly reproduces the symmetries of (2.6). If, on the other hand, $\lambda + d\tilde{\lambda} = 0$, we find that

$$f = -\frac{d-2}{3}f_0 + i\alpha + i\beta_i x^i + \frac{i}{2}\gamma x^2 \tag{2.58a}$$

$$\xi^i = \xi_0^i + \omega^i{}_j x^j + f_0 x^i \tag{2.58b}$$

$$\xi^t = \xi_0^t + \frac{d+4}{3}f_0 t \,. \tag{2.58c}$$

This means that we obtain the additional trace-quadrupole symmetry when $\lambda + d\tilde{\lambda} = 0$, just as we observed around (2.53). There are thus no additional symmetries.

## 2.6 Fracton, Carroll and Bargmann symmetries

The typical structure of the symmetry algebra of a complex scalar theory with a dipole symmetry is of the form

| | | | |
|---|---|---|---|
| $\{M_{ij}, M_{kl}\} = -4\delta_{[k[i}M_{j]l]}$ | $\{M_{jk}, P_i\} = -2\delta_{i[j}P_{k]}$ | | Arist. static |
| $\{M_{jk}, Q_i^{(2)}\} = -2\delta_{i[j}Q_{k]}^{(2)}$ | $\{P_i, Q_j^{(2)}\} = \delta_{ij}Q^{(0)}$ | | dipole sym. |
| $\{D, H\} = -zH$ | $\{D, P_i\} = -P_i$ | $\{D, Q_i^{(2)}\} = Q_i^{(2)}$ | dilatations |
| $\{P_i, Q^{(4)}\} = -Q_i^{(2)}$ | $\{D, Q^{(4)}\} = 2Q^{(4)}$ | | quadrupole |

where we included a dilatation generator $D$ with dynamical critical exponent $z$ and a quadrupole symmetry $Q$, but keeping in mind that these latter symmetries are not always present. We thus have generators of spatial rotations $M_{ij}$, spatial and temporal translations $P_i$ and $H$, the electric charge $Q^{(0)}$, the dipole charge vector $Q_i^{(2)}$, the quadrupole scalar $Q^{(4)}$ and the dilatations $D$. The first line spans the Aristotelian static symmetries which get accompanied by the symmetries of the second line once dipoles are conserved. When we have scale symmetry the third line gets added. For quadrupole symmetries one adds the commutation relations of the last line (only the first term when there is no dilation symmetry).

The subalgebra spanned by $\langle M_{ij}, H, P_i \rangle$ is naturally interpreted as an Aristotelian homogeneous space due to the absence of boost symmetries. A homogeneous space is, up to global considerations, characterised by a Lie algebra $\mathfrak{g} = \mathfrak{h} + \mathfrak{m}$ and a Lie subalgebra $\mathfrak{h}$, where $\mathfrak{m}$ is spanned by the remaining generators (the $+$ is a vector space direct sum and should not be understood as a Lie algebra direct sum). For the case at hand $\mathfrak{g}_{\text{Arist}} = \langle M_{ij}, H, P_i \rangle$ and $\mathfrak{h}_{\text{Arist}} = \langle M_{ij} \rangle$ giving rise to a $(d+1)$-dimensional manifold which is closely tied to the fact that we have $d+1$ remaining generators $\mathfrak{m} = \langle H, P_i \rangle$. This homogeneous space is Aristotelian – more precisely, the static Aristotelian spacetime [27]. We refer to [27, 41] for more details and a classification of Aristotelian algebras and spacetimes. Having specified the homogeneous space one can introduce exponential coordinates as $\sigma(t, x) = e^{tH + x^i P_i}$ in terms of which the invariants of low rank are given by a 1-form $\tau = dt$, a degenerate metric $h = \delta_{ij} dx^i dx^j$ and their duals $v = \frac{\partial}{\partial t}$ and $\delta^{ij} \frac{\partial}{\partial x^i} \frac{\partial}{\partial x^j}$, which will play a prominent role once we curve our manifold, c.f., Section 5. The action of the symmetries of the subalgebra $\mathfrak{h}$ on the coordinates is determined by $[\mathfrak{h}, \mathfrak{m}]$ quotiented by $\mathfrak{h}$, i.e., $[\mathfrak{h}, \mathfrak{m}] \bmod \mathfrak{h}$. For example, the rotations have a nontrivial action on the coordinates precisely as given in (2.2).

The relevant part for fractonic physics is the addition of the dipole charge vector $Q_i^{(2)}$ and the charge $Q^{(0)}$. In particular, the existence of a conserved dipole charge and its nontrivial commutation relation with the translations distinguishes these theories from non-fractonic theories. The geometry of the enlarged algebra, spanned by $\mathfrak{g}_{\text{Frac}} = \langle M_{ij}, H, P_i, Q^{(0)}, Q_i^{(2)} \rangle$, is still naturally interpreted as the $(d+1)$-dimensional static Aristotelian spacetime when we quotient by $\mathfrak{h}_{\text{Frac}} = \langle M_{ij}, Q^{(0)}, Q_i^{(2)} \rangle$. This is the case since the action generated by the charges $Q^{(0)}$ and $Q_i^{(2)}$ acts trivially on $\mathfrak{m} = \langle H, P_i \rangle$ and consequently on the spacetime manifold. This is in perfect agreement with (2.1) where these symmetries only act on the field. To see this consider

$$[Q_j^{(2)}, P_i] \bmod \mathfrak{h}_{\text{Frac}} = -\delta_{ij} Q^{(0)} \bmod \mathfrak{h}_{\text{Frac}} = 0 \bmod \mathfrak{h}_{\text{Frac}} , \qquad (2.60)$$

or, in other words, the commutation relations of $\langle Q^{(0)}, Q_i^{(2)} \rangle$ with $H$ and $P_i$ do not lead to elements in $\langle H, P_i \rangle$. The same arguments apply upon introducing the conserved

quadrupole moments. In both cases it is natural from the point of view of the underlying homogeneous space to quotient by the trivial symmetries, whereby we land again at our original Aristotelian geometry.

The situation slightly differs upon the introduction of the dilatations. Like for the other cases we enlarge the quotient $\mathfrak{h}$, but stick to $\mathfrak{m} = \langle H, P_i \rangle$ connected to the fact that our manifold stays $(d+1)$-dimensional. However, the action $D$ on $\mathfrak{m}$ leads again to elements in $\mathfrak{m}$ and to the action (2.3) on the coordinates. Therefore the homogeneous space and its invariants differ in this case (one could call it a Lifshitz–Weyl spacetime [42]).

Let us finally comment on two Lie algebra isomorphisms. The algebra spanned by $\langle M_{ij}, P_i, Q_i^{(2)}, Q^{(0)} \rangle$ is isomorphic to the Carroll algebra and if $Q_i^{(2)}$ is interpreted as boosts and $Q^{(0)}$ as time translations this would indeed also lead to the (flat) Carroll spacetime. However, as can be seen from (2.1) the action of $Q_i^{(2)}$ is not naturally interpreted as a Carroll boost (2.35). Since Carroll boosts are not a symmetry of the action this observation is merely a coincidental equivalence of Lie algebras and not of the underlying homogeneous spaces.

Similar remarks apply for the case with an additional conserved quadrupole symmetry for which the algebra turns our to be isomorphic to the Bargmann algebra [13], the unique central extension of the Galilei algebra that exists in any dimension. To obtain Bargmann spacetime symmetries we would interpret $Q^{(4)}$ as the time translation generator, $Q_i^{(2)}$ would generate translations, $P_i$ Bargmann boosts and $Q^{(0)}$ would be the central extension which is sometimes interpreted as mass.

While in the current setup the interpretation in terms of Carrollian and Galilean symmetries seems to be non-conventional, it might still be interesting to see if there is something to be learned by thinking of them from this other perspective.

## 2.7 Noether procedure for gauging dipole symmetry

A Lagrangian for which (2.21) vanishes for any $\alpha$ and $\beta_i$ is a complex scalar theory with dipole symmetry. If we make $\alpha$ and $\beta_i$ local[4] for such a theory we obtain

$$\delta\mathcal{L} = J_{(0)}^0 \partial_t \left( \alpha + \beta_k x^k \right) - \tilde{J}^{ij} \partial_i \partial_j \left( \alpha + \beta_k x^k \right) , \tag{2.61}$$

as can be explicitly verified.

When we apply the Noether procedure the original matter Lagrangian is called $\mathcal{L}^{(0)}$ (which is zeroth order in gauge fields). To counter the non-invariance of $\mathcal{L}^{(0)}$

---

[4]Making $\beta_i$ local is in a way taken care of by making $\alpha$ local. The functions $\alpha$ and $\beta_i$ must always appear in the combination $\alpha + \beta_i x^i$. The role of $\beta_i$ is in the second line of (2.21) ensuring that the Lagrangian has dipole symmetry which is why, from the point of view of an ordinary $U(1)$ gauging, the spatial part of the $U(1)$ current obeys (2.20). We can derive (2.61) by using that the local $U(1)$ variation with parameter $\Lambda$ of a Lagrangian with a global $U(1)$ symmetry always takes the form $\delta\mathcal{L} = J_{(0)}^0 \partial_t \Lambda + J_{(0)}^i \partial_i \Lambda$ and by using that for a theory with a dipole symmetry we have (2.20). Performing a partial integration and setting $\Lambda = \alpha + \beta_k x^k$ we obtain the desired result.

we add to it an $\mathcal{L}^{(1)}$ which is first order in a set of gauge fields whose variation gives us the objects $J^0_{(0)}$ and $\tilde{J}^{ij}$ (which are the building blocks of the $U(1)$ and dipole currents). Since the latter are fully generic we need a scalar field $\phi$ and a symmetric tensor gauge field $A_{ij}$. The expression for $\mathcal{L}^{(1)}$ is then given by

$$\mathcal{L}^{(1)} = -J^0_{(0)}\phi + \tilde{J}^{ij}A_{ij} \,, \tag{2.62}$$

where the gauge fields transform as

$$\delta\phi = \partial_t\Lambda \qquad\qquad \delta A_{ij} = \partial_i\partial_j\Lambda \,, \tag{2.63}$$

where $\Lambda = \alpha + \beta_k x^k$. The new Lagrangian is now $\mathcal{L}^{(0)} + \mathcal{L}^{(1)}$ and we need to check that this is gauge invariant. This is not guaranteed because the objects $J^0_{(0)}$ and $\tilde{J}^{ij}$ need not be gauge invariant. If they are not we add an $\mathcal{L}^{(2)}$ (which is second order in gauge fields) etc. until the procedure stops which happens when $\mathcal{L}^{(0)} + \mathcal{L}^{(1)} + \dots$ is gauge invariant. For (non-)Abelian symmetries (and polynomial Lagrangians) this always happens after a finite number of steps.

Now, suppose we only assume that $\mathcal{L}$ is $U(1)$ invariant, i.e., the first line of (2.21) vanishes but we do not assume that there is also a dipole symmetry, so that the second line of (2.21) does not need to vanish, then varying $\mathcal{L}$ for local $\alpha$ and $\beta_i$ leads to

$$\delta\mathcal{L} = J^0_{(0)}\partial_t\left(\alpha + \beta_k x^k\right) + J^i_{(0)}\partial_i\left(\alpha + \beta_k x^k\right) \,. \tag{2.64}$$

Up to a total derivative, this can be rewritten as

$$\delta\mathcal{L} = J^0_{(0)}\partial_t\left(\alpha + \beta_k x^k\right) - \tilde{J}^{ij}\partial_i\partial_j\left(\alpha + \beta_k x^k\right) + \left(J^i_{(0)} - \partial_j\tilde{J}^{ji}\right)\partial_i\left(\alpha + \beta_k x^k\right) \,. \tag{2.65}$$

Applying the Noether procedure to the latter leads to an $\mathcal{L}^{(1)}$ given by

$$\mathcal{L}^{(1)} = -J^0_{(0)}\phi + \tilde{J}^{ij}A_{ij} - \left(J^i_{(0)} - \partial_j\tilde{J}^{ji}\right)B_i \,, \tag{2.66}$$

where the gauge fields transform as

$$\delta\phi = \partial_t\left(\alpha + \beta_k x^k\right) \qquad \delta B_i = \partial_i\left(\alpha + \beta_k x^k\right) \qquad \delta A_{ij} = \partial_i\partial_j\left(\alpha + \beta_k x^k\right) \,. \tag{2.67}$$

If the theory really only has a $U(1)$ symmetry and no dipole symmetry then we can write $A_{ij} = \partial_{(i}B_{j)}$ as in that case the Noether current is just given by $(J^0_{(0)}, J^i_{(0)})$. If however the theory has a dipole symmetry we need to ensure this which can be achieved by assigning to $B_i$ the additional Stückelberg transformation

$$\delta B_i = -\Sigma_i \,. \tag{2.68}$$

The $\Sigma_i$ transformation is there to enforce equation (2.20). Using partial integration we can rewrite $\mathcal{L}^{(1)}$ as

$$\mathcal{L}^{(1)} = -J^0_{(0)}\phi - J^i_{(0)}B_i + \tilde{J}^{ij}\tilde{A}_{ij} \,, \tag{2.69}$$

where we defined

$$\tilde{A}_{ij} = A_{ij} - \partial_{(i} B_{j)} \,. \tag{2.70}$$

In this latter formulation the gauge fields transform as

$$\delta\phi = \partial_t \left( \alpha + \beta_k x^k \right) \qquad \delta B_i = \partial_i \left( \alpha + \beta_k x^k \right) - \Sigma_i \qquad \delta\tilde{A}_{ij} = \partial_{(i}\Sigma_{j)} \,. \tag{2.71}$$

At the level of the currents the situation is as follows: we have the following responses,

$$- J^0_{(0)}\delta\phi - J^i_{(0)}\delta B_i + \tilde{J}^{ij}\delta\tilde{A}_{ij} \,, \tag{2.72}$$

where $\tilde{J}^{ij} = \tilde{J}^{ji}$. This leads to the following Ward identities

$$0 = \partial_t J^0_{(0)} + \partial_i J^i_{(0)} \tag{2.73a}$$

$$0 = J^i_{(0)} - \partial_j \tilde{J}^{ij} \tag{2.73b}$$

for the $\Lambda = \alpha + \beta_k x^k$ and $\Sigma_i$ gauge parameters, respectively. This in turn gives rise to the charge and dipole conservation equations

$$0 = \partial_t J^0_{(0)} + \partial_i\partial_j \tilde{J}^{ij} \tag{2.74a}$$

$$0 = \partial_t \left( x^i J^0_{(0)} \right) + \partial_j \left( x^i J^j_{(0)} - \tilde{J}^{ij} \right) \,. \tag{2.74b}$$

The gauge field $B_i$ is now a Stückelberg field and can thus be gauged away entirely. Setting both $B_i$ and its total gauge transformation to zero, i.e., $\delta B_i = \partial_i \Lambda - \Sigma_i = 0$ tells us that the residual gauge transformations in the gauge $B_i = 0$ are described by $\Sigma_i = \partial_i \Lambda$, and thus in this gauge $\tilde{A}_{ij} = A_{ij}$ which transforms as $\delta A_{ij} = \partial_i\partial_j\Lambda$.

Lastly, we want to comment on the Noether procedure for the case where the Lagrangian has the additional $\gamma$-symmetry described in section 2.3. In this case, if we make $\alpha$, $\beta$ and $\gamma$ local, the variation of the Lagrangian becomes

$$\delta\mathcal{L}^{(0)} = J^0_{(0)}\partial_t \left( \alpha + \beta_k x^k + \frac{1}{2}\gamma x^2 \right)$$
$$- \tilde{J}^{ij} \left[ \partial_i\partial_j \left( \alpha + \beta_k x^k + \frac{1}{2}\gamma x^2 \right) - \frac{1}{d}\delta_{ij}\partial_k\partial_k \left( \alpha + \beta_k x^k + \frac{1}{2}\gamma x^2 \right) \right] \,, \tag{2.75}$$

where we used that $\tilde{J}^{ii} = 0$. We therefore need to introduce a scalar field $\phi$ and a symmetric traceless tensor gauge field $A_{ij}$. The expression for $\mathcal{L}^{(1)}$ is then given by

$$\mathcal{L}^{(1)} = - J^0_{(0)}\phi + \tilde{J}^{ij} A_{ij} \,, \tag{2.76}$$

where the gauge fields transform as

$$\delta\phi = \partial_t \left( \alpha + \beta_k x^k + \frac{1}{2}\gamma x^2 \right) \tag{2.77a}$$

$$\delta A_{ij} = \partial_i\partial_j \left( \alpha + \beta_k x^k + \frac{1}{2}\gamma x^2 \right) - \frac{1}{d}\delta_{ij}\partial_k\partial_k \left( \alpha + \beta_k x^k + \frac{1}{2}\gamma x^2 \right) \,. \tag{2.77b}$$

Thus, it is clear that the $\gamma$-symmetry leads to a tracelessness condition on $A_{ij}$ in the Noether procedure.

# 3 Worldline actions and coupling to scalar charge gauge theory

Now that we have identified the gauge fields involved in gauging the dipole symmetry we can ask ourselves what is the form a worldline action would take that couples to these fields in a gauge invariant fashion. This would be the analogue of the coupling of a charged point particle as we are familiar with from electrodynamics where such couplings lead to the Lorentz force, i.e., a coupling of the form $q \int d\lambda A_\mu \dot{X}^\mu$. The general form of the action we are looking for is

$$S_{\text{tot}} = S_{\text{SCGT}} + S_{\text{int}} + S_{\text{kin}}, \tag{3.1}$$

where $S_{\text{SCGT}}$ is the action for the scalar charge gauge theory involving the fields $\phi$ and $A_{ij}$ (which we will discuss in detail in Section 4) and where $S_{\text{kin}}$ is some yet to be determined kinetic term for the embedding scalars $X^i$ (see further below). The interaction action is

$$S_{\text{int}} = -q \int_{\lambda_i}^{\lambda_f} d\lambda \left[ \dot{T} \left( \phi - X^i \partial_i \phi \right) - X^i \dot{X}^j A_{ij} \right], \tag{3.2}$$

where the dot denotes differentiation with respect to $\lambda$, the parameter along the worldline and where $\lambda_i$ and $\lambda_f$ denote the endpoints of the worldline parameter. The embedding coordinates are $T$ and $X^i$. The gauge fields and its derivatives are evaluated along the worldline where $t = T$ and $x^i = X^i$. The interaction action is worldline reparametrisation invariant. Under the gauge transformation $\delta\phi = \partial_t \Lambda$ and $\delta A_{ij} = \partial_i \partial_j \Lambda$ the combination $\dot{T} \left( \phi - X^i \partial_i \phi \right) - X^i \dot{X}^j A_{ij}$ transforms as

$$\delta \left[ \dot{T} \left( \phi - X^i \partial_i \phi \right) - X^i \dot{X}^j A_{ij} \right] = \dot{T} \partial_t \left( \Lambda - X^i \partial_i \Lambda \right) + \dot{X}^j \partial_j \left( \Lambda - X^i \partial_i \Lambda \right) \tag{3.3a}$$

$$= \frac{d}{d\lambda} \left( \Lambda - X^i \partial_i \Lambda \right), \tag{3.3b}$$

so that $S_{\text{int}}$ remains invariant up to boundary terms (endpoints of the worldline). The gauge variation is precisely zero for the target space symmetries $\partial_\mu \left( \Lambda - x^i \partial_i \Lambda \right) = 0$, i.e., $\Lambda = \alpha + \beta_i x^i$ with $\alpha$ and $\beta_i$ constant.

In (3.2) the fields are evaluated along the worldline. In order to compute the spacetime currents associated with the flow of these objects we write (3.2) as follows

$$S_{\text{int}} = -q \int dt d^d x \int_{\lambda_i}^{\lambda_f} d\lambda \delta(t - T(\lambda)) \delta(x - X(\lambda)) \left[ \dot{T} \left( \phi - X^i \partial_i \phi \right) - X^i \dot{X}^j A_{ij} \right], \tag{3.4}$$

where the integrand of the $\lambda$-integral is no longer restricted to the worldline, so for example $\phi$ is now a function of $t, x^i$ and not of $T(\lambda), X^i(\lambda)$ as was the case in (3.2).

Let us define

$$\delta_A S_{\text{int}} = \int dt d^d x \left( -J^0_{(0)} \delta\phi + \tilde{J}^{ij} \delta A_{ij} \right). \tag{3.5}$$

This leads to

$$J^0_{(0)} = q \int_{\lambda_i}^{\lambda_f} d\lambda \delta(t - T(\lambda))\dot{T}\left[\delta(x - X(\lambda)) + X^i \partial_i \delta(x - X(\lambda))\right], \qquad (3.6a)$$

$$\tilde{J}^{ij} = \frac{q}{2} \int_{\lambda_i}^{\lambda_f} d\lambda \delta(t - T(\lambda))\delta(x - X(\lambda))\left(X^i \dot{X}^j + X^j \dot{X}^i\right). \qquad (3.6b)$$

We can fix worldline reparametrisation invariance by setting $T = \lambda$. If we do this we obtain

$$J^0_{(0)} = q\left[\delta(x - X(t)) + X^i \partial_i \delta(x - X(t))\right], \qquad (3.7a)$$

$$\tilde{J}^{ij} = \frac{q}{2}\delta(x - X(t))\left(X^i \dot{X}^j + X^j \dot{X}^i\right), \qquad (3.7b)$$

where the dot now denotes differentiation with respect to $t$.

Gauge invariance of $S_{\text{int}}$ tells us that we have the identically conserved equation

$$\int dt d^d x \left[\partial_t J^0_{(0)} + \partial_i \partial_j \tilde{J}^{ij}\right]\Lambda = 0, \qquad (3.8)$$

for all $\Lambda$ that are at most linear in $X^i$ at the endpoints $\lambda_i = t_i$ and $\lambda_f = t_f$. This implies that

$$\partial_t J^0_{(0)} + \partial_i \partial_j \tilde{J}^{ij} = 0, \qquad (3.9)$$

as can be explicitly verified by using $\dot{X}^i \partial_i \delta(x - X(t)) = -\partial_t \delta(x - X(t))$. The current is identically conserved because for the worldline theory there are no other fields (other than the gauge fields) transforming under the gauge transformation with parameter $\Lambda$. We can construct $d$ additional (identically) conserved equations, namely the currents

$$J^{0j}_{(2)} = x^j J^0_{(0)}, \qquad J^{ij}_{(2)} = x^j \partial_k \tilde{J}^{ik} - \tilde{J}^{ij}, \qquad (3.10)$$

which obey

$$\partial_t J^{0j}_{(2)} + \partial_i J^{ij}_{(2)} = 0, \qquad (3.11)$$

by virtue of (3.9).

We can define a $U(1)$ and dipole charge in the sense of distributions, i.e., let $\varepsilon(x)$ be a test function then we define

$$Q^{(0)}[\varepsilon] := \int d^d x \varepsilon(x) J^0_{(0)} = q\varepsilon(X(t)) - qX^i(t)\left(\partial_i \varepsilon(x)\right)\Big|_{x=X(t)}, \qquad (3.12a)$$

$$Q^{(2)}_j[\varepsilon] := \int d^d x \varepsilon(x) J^{0j}_{(2)} = -qX^j(t)X^i(t)\left(\partial_i \varepsilon(x)\right)\Big|_{x=X(t)} = X^j\left(Q^{(0)}[\varepsilon] - q\varepsilon(X(t))\right). \qquad (3.12b)$$

For $\varepsilon = 1$, we obtain the total $U(1)$ and total dipole charge, which are $q$ and zero, respectively[5].

---

[5]The expression for $Q^{(0)}[\varepsilon]$ contains the first two terms in the Taylor expansion of $q\varepsilon(x)$ around $x = X(t)$ evaluated at $x = 0$, i.e., $\varepsilon(x) = \varepsilon(X(t)) + (x^i - X^i(t))(\partial_i \varepsilon(x))\Big|_{x=X(t)} + \cdots$ evaluated at $x = 0$.

The kinetic term is of the form

$$S_{\text{kin}} = \int d\lambda \dot{T} f\left(\frac{|\dot{\vec{X}}|^2}{\dot{T}^2}\right). \tag{3.13}$$

This is dictated by translation invariance of $T$ and $X^i$ and rotational symmetry of the $X^i$. These target space symmetries become global symmetries of the worldline theory. Finally, the form of the Lagrangian is such that we have worldline reparametrisation invariance for any $f$. Well-known examples of such a function $f$ are

$$\frac{|\dot{\vec{X}}|^2}{2\dot{T}^2} \qquad \text{or} \qquad -\sqrt{1 - \frac{|\dot{\vec{X}}|^2}{\dot{T}^2}}, \tag{3.14}$$

where the first expression for $f$ is for theories with Galilei invariance and the second expression for $f$ is for theories with Lorentz invariance. In the case we are dealing with there is no boost symmetry and hence $f$ is not uniquely fixed.

Let us come back to the fact that the total dipole charge is zero. For a point particle a nonzero dipole charge is proportional to $qX^i$ (with respect to some chosen origin). For this to be conserved the particle cannot move unless the total dipole charge is zero. For a point particle this would imply $q = 0$, but our worldline theory does not describe a point particle because the charge distribution (3.7a) involves a derivative of a delta function and so the above argument about immobility does not apply. Here we have an example of a worldline theory for which the total dipole charge is zero while the total charge is $q$ and there is no mobility restriction. It would be interesting to investigate these mobile and dipole-like objects in more detail.

If the scalar charge gauge theory is traceless, the gauge transformations instead read $\delta\phi = \partial_t\Lambda$ and $\delta A_{ij} = \partial_i\partial_j\Lambda - \frac{1}{d}\delta_{ij}\partial^2\Lambda$. In this case, the gauge invariant interaction term is

$$S_{\text{int}} = -q \int_{\lambda_i}^{\lambda_f} d\lambda \left[\dot{T}\left(\phi - X^i\partial_i\phi + \frac{1}{2d}X^kX^k\partial_j\partial_j\phi\right) - \left(X^i\dot{X}^j - \frac{1}{d}\delta^{ij}X^k\dot{X}^k\right)A_{ij}\right.$$
$$\left. + \frac{1}{2(d-1)}X^kX^k\left(\dot{X}^j\partial_iA_{ij} - \frac{1}{d}\delta_{ij}\dot{X}^l\partial_lA_{ij}\right)\right]. \tag{3.15}$$

Under a gauge transformation the Lagrangian transforms into a total derivative term that is proportional to

$$\frac{d}{d\lambda}\left(\Lambda - X^i\partial_i\Lambda + \frac{1}{2d}X^kX^k\partial_j\partial_j\Lambda\right). \tag{3.16}$$

It would be interesting to study this action in more detail and to generalise these results to higher order symmetries.

# 4 Scalar charge gauge theory

The scalar charge gauge theory was the first continuum model proposed to describe fracton behaviour [21, 22, 43] (see also the review [12]).

In this section, we develop the scalar charge gauge theory by making dynamical the gauge fields obtained by gauging the dipole symmetry using the Noether procedure, c.f., Section 2.7. We analyse the gauge sector and cohomology of the theory. It is useful to contrast this discussion with electrodynamics, which we have added for convenience in Appendix A, and linearised general relativity which it perfectly mirrors, see, e.g., the introduction of [44].

By modifying the pre-symplectic potential, we show how the traceless theory emerges from a Faddeev–Jackiw type Hamiltonian analysis of this modified theory, and, in particular, we demonstrate that the traceless scalar charge gauge theory with a nontrivial magnetic sector only exists for $d \geq 3$. After computing the spectrum of the scalar charge gauge theory, we conclude with some observations regarding the similarities between the scalar charge gauge theory and the theory of partially massless gravitons.

## 4.1 Poisson bracket and gauge generator

The fundamental fields of scalar charge gauge theory are the symmetric fields

$$A_{ij} \sim \boxed{i\,j} \, , \tag{4.1}$$

and their canonical conjugate momenta $E_{ij}$. Boxes after the $\sim$ symbol denote Young tableaux that describe the symmetries of the indices. The indices $i, j, \ldots$ are spatial, i.e., they run from 1 to $d$. The fundamental fields satisfy the equal time Poisson bracket

$$\{A_{ij}(\vec{x}), E_{kl}(\vec{y})\} = \delta_{i(k}\delta_{l)j}\delta(\vec{x} - \vec{y}) \, , \tag{4.2}$$

and, by assumption, we have a gauge symmetry

$$\delta_\Lambda A_{ij} = \partial_i \partial_j \Lambda \quad \text{and} \quad \delta_\Lambda E_{ij} = 0 \, , \tag{4.3}$$

for fixed time $t$. The gauge parameter is a scalar $\Lambda = \Lambda(t, \vec{x}) \sim \bullet$, where the bullet is the Young tableaux for a scalar.

The gauge symmetries are generated canonically via $\delta_\Lambda F = \{F, \tilde{G}[\Lambda]\}$ with the gauge generator $\tilde{G}$. It must have a well-defined functional derivative, i.e., $\delta\tilde{G}$ should not lead to boundary terms upon integration by parts. This means that the gauge generator consists of two parts $\tilde{G}[\Lambda] = G[\Lambda] + Q[\Lambda]$ [45, 46]

$$G[\Lambda] = \int \mathrm{d}^d x \, (\Lambda \partial_i \partial_j E_{ij}) \tag{4.4}$$

$$Q[\Lambda] = \int \mathrm{d}^d x \, \partial_i \left[ \partial_j \Lambda E_{ij} - \Lambda \partial_j E_{ij} \right] \, , \tag{4.5}$$

where $G[\Lambda]$ is a bulk and $Q[\Lambda]$ a boundary term. The charge $Q[\Lambda]$ does not necessarily vanish on-shell for gauge parameters $\Lambda$ that are nonzero on the boundary. On the other hand, the bulk term $G[\Lambda]$ vanishes on-shell (more precisely on the constraint surface) and only the boundary term remains, $\tilde{G}[\Lambda] \approx Q[\Lambda]$. In this sense gauge transformations with nonzero $Q[\Lambda]$ actually generate physical symmetries and change the physical state of the system. As we will show next, they also lead to nontrivial conserved charges. They are called improper gauge transformations [45, 46]. When $Q[\Lambda]$ vanishes the gauge symmetries are proper and are nothing but the redundancies inherent in our description [45, 46].

Let us now couple sources to our theory. The charge conservation equation for our dipole symmetry takes the form (2.73). For sufficient fall-offs of $J^i_{(0)}$, this leads to conservation of the charge

$$Q^{(0)} = \int d^d x \; J^0_{(0)} \qquad\qquad \dot{Q}^{(0)} = 0 \,, \qquad\qquad (4.6)$$

as well as the conservation of the dipole charge

$$Q^{(2)}_i = \int d^d x \; x^i J^0_{(0)} \,. \qquad\qquad (4.7)$$

The boundary charges (4.5) are compatible with these conserved $U(1)$ and dipole charges. Verifying this requires the generalised Gauss constraint

$$\partial_i \partial_j E_{ij} = -J^0_{(0)} \,, \qquad\qquad (4.8)$$

which follows from coupling our theory to matter as we will show at the start of Section 4.3. Setting $\Lambda = \alpha$, where $\alpha$ is constant, we obtain from the boundary charges

$$Q[\alpha] = \alpha \int d^d x \; J^0_{(0)} = \alpha Q^{(0)} \,. \qquad\qquad (4.9)$$

If we instead set $\Lambda = \beta_i x^i$, we get the dipole charge after using (4.8)

$$Q[\beta_i x^i] = \beta_i \int d^d x \; x^i J^0_{(0)} = \beta_i Q^{(2)}_i \,. \qquad\qquad (4.10)$$

## 4.2 The gauge sector

Using cohomology [44, 47, 48] (see Appendix C.I in [49] for a concise summary which is sufficient for the following arguments) we will now construct a gauge invariant "curvature" or "magnetic field" tensor which has the important property that it fully characterises the gauge symmetries and satisfy a Bianchi identity. It is useful to contrast the following discussion with electrodynamics, which we provide for convenience in Appendix A.

As a first step it is useful to rewrite the gauge transformation as a generalised differential

$$(d_2 d_1 \Lambda)_{ij} = \partial_j (d_1 \Lambda)_i = 2\partial_i \partial_j \Lambda \sim \boxed{i\,j}\,. \tag{4.11}$$

where $d_1$ acts with a derivative on the first and $d_2$ on the second column of the Young tableaux symmetries and we afterwards Young project accordingly to have the right index symmetries. We can represent these operations as

$$\tag{4.12}$$

We want to emphasise that $d_2$ does not exist for equal height tableaux. These operations imply that

$$(d_1)^2 = 0 = (d_2)^2\,. \tag{4.13}$$

We refer to potentials of the form $A = d_2 d_1 \Lambda$ as being pure gauge.

We define the gauge invariant "curvature" or "magnetic field" $F_{ijk}$ by

$$F_{ijk} = (d_1 A)_{ijk} := 2\partial_{[i} A_{j]k} \sim \begin{array}{|c|c|}\hline i & k \\\hline j \\\cline{1-1}\end{array}\,. \tag{4.14}$$

This tensor is an irreducible $GL(d)$ representation and has mixed symmetry, i.e., it is neither totally symmetric nor totally antisymmetric. These curvatures are a subset of all "hook" symmetric tensors, as denoted by the Young tableaux on the right hand side of (4.14). In general hook symmetry means that the first two indices are antisymmetric $F_{[ij]k} = F_{ijk}$ and an antisymmetrisation over all indices vanishes $F_{[ijk]} = 0$. A useful relation, which follows from these symmetries, is

$$F_{i[jk]} = -\frac{1}{2} F_{jki}\,. \tag{4.15}$$

By construction the curvature (4.14) vanishes when the potential is pure gauge

$$F_{ijk} = 4\partial_{[i} \partial_{j]} \partial_k \Lambda = 0\,. \tag{4.16}$$

In other words, the curvatures do not see the irrelevant pure gauge potentials, something we can also write as

$$d_1 d_2 d_1 \Lambda = d_2 (d_1)^2 \Lambda = 0\,. \tag{4.17}$$

Conversely, a vanishing curvature of an arbitrary potential $A_{ij}$ implies that this potential is pure gauge, i.e.,

$$F_{ijk} = 2\partial_{[i} A_{j]k} = 0 \quad \Longrightarrow \quad A_{ij} = 2\partial_i \partial_j \Lambda\,, \tag{4.18}$$

or in short $d_1 A = 0 \implies A = \frac{1}{2} d_2 d_1 \Lambda$. This shows that only the irrelevant pure gauge potentials get lost when going to curvatures, i.e., the curvatures fully capture the gauge symmetries. The relation (4.18) can be shown using the Poincaré lemma and the symmetry properties of the involved tensors.

The final class of tensors we introduce are tensors with the following Young tableaux

$$T_{ijkl} \sim \begin{array}{|c|c|} \hline i & l \\ \hline j \\ \cline{1-1} k \\ \cline{1-1} \end{array}. \tag{4.19}$$

This means that $T_{ijkl} = T_{[ijk]l}$ and $T_{[ijkl]} = 0$. A subset of these tensors are differentials of hook symmetric tensors $F_{jkl}$ of the form

$$(d_1 F)_{ijkl} = 3 \partial_{[i} F_{jk]l} \,. \tag{4.20}$$

If the hook symmetric tensor is the curvature of a potential, see (4.14), it follows that

$$\partial_{[i} F_{jk]l} = 2 \partial_{[i} \partial_j A_{k]l} = 0 \qquad (\Leftrightarrow d_1 F = (d_1)^2 A = 0) \tag{4.21}$$

which is the generalised differential Bianchi identity.

Conversely, $\partial_{[i} F_{jk]l} = 0$, where $F_{ijk}$ is a generic hook symmetric tensor, implies that $F_{jkl}$ is the curvature of a potential. To see this we start by

$$\partial_{[i} F_{jk]l} = 0 \quad \implies \quad F_{ijk} = 2 \partial_{[i} M_{j]k} \tag{4.22}$$

where we can decompose $M_{ij}$ into a symmetric tensor $\tilde{A}_{ij} = \tilde{A}_{ji}$ and an antisymmetric tensor $\tilde{B}_{ij} = -\tilde{B}_{ji}$ as

$$M_{ij} = \tilde{A}_{ij} + \tilde{B}_{ij} \,. \tag{4.23}$$

We still have to enforce that $F_{ijk}$ is a hook symmetric tensor, $F_{[ijk]} = 2 \partial_{[i} \tilde{B}_{jk]} = 0$ leads then via the Poincaré lemma to $\tilde{B}_{ij} = \partial_{[i} B_{j]}$. It follows that

$$\partial_{[i} F_{jk]l} = 0 \implies F_{ijk} = 2 \partial_{[i} \tilde{A}_{j]k} - \partial_k \partial_{[i} B_{j]} \quad (\Leftrightarrow d_1 F = 0 \implies F = d_1(\tilde{A} - \frac{1}{2} d_2 B)) \tag{4.24}$$

where

$$A_{ij} = \tilde{A}_{ij} - \partial_{(i} B_{j)} \qquad (\Leftrightarrow A = \tilde{A} - \frac{1}{2} d_2 B) \,. \tag{4.25}$$

We have the gauge freedom parametrised by $\Sigma_i$ and $\Lambda$,

$$\tilde{A}_{ij} \mapsto \tilde{A}_{ij} + \partial_{(i} \Sigma_{j)} \qquad (\tilde{A} \mapsto \tilde{A} + \frac{1}{2} d_2 \Sigma) \tag{4.26a}$$
$$B_i \mapsto B_i + \Sigma_i - \partial_i \Lambda \qquad (B \mapsto B + \Sigma - d_1 \Lambda) \tag{4.26b}$$
$$A_{ij} \mapsto A_{ij} + \partial_i \partial_j \Lambda \qquad (A \mapsto A + \frac{1}{2} d_2 d_1 \Lambda) \tag{4.26c}$$
$$F_{ijk} \mapsto F_{ijk} \qquad (F \mapsto F) \,. \tag{4.26d}$$

We can partially gauge fix by demanding that $B_i = 0$, which can be reached by the gauge transformation $\Sigma_i = -B_i$ and $\Lambda = 0$. The residual gauge transformation leaving this constraint unaltered are then given by $\Sigma_i - \partial_i \Lambda = 0$. This means the partial gauge fixed version of our statement above is $A_{ij} = \tilde{A}_{ij}$. This shows that the Bianchi identity characterises the curvatures that come from gauge potentials.

What we have described is a generalisation of the gauge structure of electrodynamics and linearised gravity. With the differential operators given in (4.11), (4.14), and (4.20), respectively, we have also shown that we obtain an exact sequence that we can schematically depict as

$$\bullet \xrightarrow{d_2 d_1} \boxed{\phantom{x}\phantom{x}} \xrightarrow{d_1} \boxed{\begin{array}{cc} & \\ & \end{array}} \xrightarrow{d_1} \boxed{\begin{array}{c} \\ \\ \end{array}} \,. \tag{4.27}$$

This subsection should be contrasted with Section 2.7 starting around (4.18), where the Noether procedure led to a similar structure.

## 4.3 Hamiltonian for scalar charge gauge theory

The phase space Lagrangian (up to total derivatives) is schematically of the form $p\dot{q} - H + \text{constraints}$. For our theory the only constraint is the (generalised) Gauss law and the phase space variables are $A_{ij}$ and $E_{ij}$. This leads to

$$\mathcal{L}[A_{ij}, E_{ij}, \phi] = E_{ij}\dot{A}_{ij} - \mathcal{H} - \phi \partial_i \partial_j E_{ij} \,, \tag{4.28}$$

where $\phi$ is the Lagrange multiplier for the Gauss constraint. This is the Lagrangian for the source-free part of the theory. If we include matter fields that couple to our gauge fields then we use that the total variation of the gauge invariant matter Lagrangian $\mathcal{L}_{\mathrm{mat}}[A_{ij}, \phi, \Phi]$, where the matter fields are collectively denoted by $\Phi$, is given by

$$\delta \mathcal{L}_{\mathrm{mat}}[A_{ij}, \phi, \Phi] = -J^0_{(0)}\delta\phi + \tilde{J}^{ij}\delta A_{ij} \,, \tag{4.29}$$

where the variations are arbitrary and where we have omitted the terms proportional to $\delta\Phi$.

The first term in (4.28) contains the (pre)symplectic potential from which we can derive the (pre)symplectic form whose inverse gives the Poisson brackets (4.2) (see, e.g., [23, 24]). The Lagrange multiplier $\phi$ is the same field we encountered in the Noether procedure in Section 2.7.

We now want to define a Hamiltonian $\mathcal{H}$. We demand that $\mathcal{H}$ is:

- $\mathfrak{so}(d)$-rotation invariant: this means we use $\delta_{ij}$ and $\epsilon_{i_1 \cdots i_d}$ to contract all indices.

- Gauge invariant: this means we build $\mathcal{H}$ out of only gauge invariant objects $E_{ij}$ and $F_{ijk}$, the analogues of the electric and magnetic field strengths. The Hamiltonian then commutes with the Gauss constraint and the latter Poisson commutes with itself so that the Gauss constraint is first-class.

- Quadratic in $E_{ij}$, so that we can integrate out $E_{ij}$ and obtain a Lagrangian that is second order in time derivatives.

- At most quadratic in $F_{ijk}$ (for simplicity).

- Bounded from below.

Up to total derivatives there are no linear terms that one can write. The only candidate is $E_{ii}$ but this is a total derivative term in the Lagrangian when expressed in terms of the gauge potentials. These requirements lead in generic dimension to the Hamiltonian

$$\mathcal{H} = \frac{g_1}{2} E_{ij} E_{ij} + \frac{g_2}{2} E_{ii}{}^2 + \frac{h_1}{4} F_{ijk} F_{ijk} + \frac{h_2}{2} F_{ijj} F_{ikk} \, . \tag{4.30}$$

Let us discuss these terms:

- The $g_1$ and $h_1$ terms are the terms that are commonly discussed in the literature and mimic electrodynamics, c.f., (A.12).

- The $g_2$ and $h_2$ terms can be added because of the possibility to treat the trace of $A_{ij}$ separately.

- We could have added a term proportional to $F_{ijk} F_{ikj}$ but using the identity $2 F_{ijk} F_{ikj} = F_{ijk} F_{ijk}$, (which follows from (4.15)) it does not give anything new.

What remains to be done is to analyse the ranges of the parameters $g_1, g_2, h_1, h_2$. We start with the electric sector. In order that the electric sector with coupling constants $g_1$ and $g_2$ is bounded from below we need that $g_1 \geq 0$ and $g_1 + d g_2 \geq 0$. This follows from writing $E_{ij}$ in a traceless and traceful part and demanding that the traceless and traceful parts contribute each non-negatively to the Hamiltonian. Next, in order to be able to solve for $E_{ij}$ after varying the phase space Lagrangian with respect to $E_{ij}$, so that we can integrate it out and obtain the Lagrangian expressed in terms of the gauge potentials, we must require that $g_1 > 0$ and $g_1 + d g_2 > 0$.

In Section 4.7, we will show that the case $g_1 + d g_2 = 0$, which needs to be treated separately, plays an important role in the so-called traceless scalar charge theory, which is a theory with a slightly different gauge transformation for the field $A_{ij}$.

Due to the hook symmetry of $F_{ijk}$ it is more difficult to find the necessary conditions for the magnetic part of the Hamiltonian to be bounded from below. We will solve this problem for $d \geq 3$ by expressing the Lagrangian in terms of the magnetic field which we define as follows

$$B_{Ij} := \frac{1}{2} \epsilon_{Ilm} F_{lmj} \, , \tag{4.31}$$

where the capital letter $I = i_1, ..., i_{d-2}$ denotes a multi-index. It follows from this definition that the magnetic field is completely traceless. From (4.31) we learn that

$$F_{ijk} = \frac{1}{(d-2)!} \epsilon_{ijN} B_{Nk} \,. \tag{4.32}$$

Using this we can rewrite the magnetic part of the Hamiltonian as follows

$$\mathcal{H}_{\text{mag}} = \frac{h_1 + h_2}{2(d-2)!} B_{Ij} B_{Ij} - \frac{h_2}{2(d-3)!} B_{i_1...i_{d-3}i_{d-2}j} B_{i_1...i_{d-3}ji_{d-2}} \,. \tag{4.33}$$

Splitting the last two indices of the magnetic field into its symmetric and antisymmetric parts, $B_{i_1...i_{d-2}j} = B_{i_1...(i_{d-2}j)} + B_{i_1...[i_{d-2}j]}$, we find that $h_1 \geq 0$ as well as $h_1 + (d-1)h_2 \geq 0$ in order for the magnetic part of Hamiltonian to be bounded from below. All in all, this means that we get the following conditions for the Hamiltonian to be bounded from below

$$g_1 > 0 \qquad g_1 + dg_2 > 0 \qquad h_1 \geq 0 \qquad h_1 + (d-1)h_2 \geq 0 \,. \tag{4.34}$$

### 4.4 Lagrangian of scalar charge gauge theory

The phase space action in a generic dimension is defined by

$$S[A_{ij}, E_{ij}, \phi] = \int dt d^d x \left( E_{ij} \dot{A}_{ij} - \mathcal{H} - \phi \partial_i \partial_j E_{ij} + \partial_i K_i \right) + S_{\text{bdry}} \tag{4.35a}$$

$$= \int dt d^d x \left( E_{ij} (\dot{A}_{ij} - \partial_i \partial_j \phi) - \mathcal{H} \right) + S_{\text{bdry}} \,, \tag{4.35b}$$

where $\mathcal{H}$ is given in (4.30). The Lagrangian is invariant under the gauge transformations $\delta A_{ij} = \partial_i \partial_j \Lambda$ and $\delta \phi = \partial_0 \Lambda$. The term $S_{\text{bdry}}$ is a suitable boundary action that depends on the type of variational problem we consider. The term $K_i$, which is closely related to the charge (4.5), is given by

$$K_i = -\partial_j \phi E_{ij} + \phi \partial_j E_{ij} \,. \tag{4.36}$$

The Lagrange multiplier $\phi$ enforces the "generalised Gauss constraint"

$$\partial_i \partial_j E_{ij} = -J^0_{(0)} \,, \tag{4.37}$$

where we included a source term $J^0_{(0)}$ (which is the response to varying $\phi$) and has undetermined time evolution, i.e., it is a redundancy of our theory. The variation of the phase space action of the scalar charge gauge theory coupled to some matter sector leads to

$$\delta S[A_{ij}, E_{ij}, \phi, \Phi] = \int dt d^d x \left[ \left( \dot{A}_{ij} - \partial_i \partial_j \phi - g_1 E_{ij} - g_2 \delta_{ij} E_{kk} \right) \delta E_{ij} \right. \tag{4.38a}$$

$$+ \left( -\dot{E}_{ij} + h_1 \partial_m F_{m(ij)} + h_2 \delta_{ij} \partial_m F_{mnn} - h_2 \partial_{(i} F_{j)nn} + \tilde{J}_{ij} \right) \delta A_{ij} \tag{4.38b}$$

$$\left. + \left( -\partial_i \partial_j E_{ij} - J^0_{(0)} \right) \delta \phi + \partial_\mu \theta^\mu \right] + \delta S_{\text{bdry}} \,, \tag{4.38c}$$

where we omitted the variation of the matter fields that we collectively denote by $\Phi$ and where we furthermore defined

$$\theta^0 = E_{ij}\delta A_{ij}\,, \tag{4.39a}$$

$$\theta^i = \partial_j E_{ij}\delta\phi - E_{ij}\partial_j\delta\phi - h_1 F_{ijk}\delta A_{jk} - 2h_2 F_{jmm}\,\delta_{i[j}\delta A_{k]k}\,. \tag{4.39b}$$

A well-posed variational problem means that the variation of the action vanishes on-shell and for suitable boundary conditions for the variations. We would like to consider a Dirichlet problem where we keep the fields $\phi$ and $A_{ij}$ fixed at the boundaries. However we are dealing with a theory that depends on second order spatial derivatives of $\phi$ and so we also need to say something about what we do with $\partial_i\phi$ at the boundary. Since $\phi$ is kept fixed on the boundary the same is true for its tangential derivatives. So we only need to say something about the normal derivative of $\phi$ at the boundary, i.e., $n^i\partial_i\phi$ where $n^i$ is the outward pointing unit normal at the boundary. We will keep this fixed as well. Hence for a Dirichlet variational problem we do not need to choose a nonzero $S_{\mathrm{bdry}}$.

The degrees of freedom are given by half the total amount of canonical variables $\{d(d+1)/2\}$ minus the amount of first class constraints $\{1\}$, leading to $d(d+1)/2-1$ degrees of freedom in $d$ spatial dimensions.

We now want to solve for the momenta $E_{ij}$ to write the action in configuration space, i.e., in terms of $A_{ij}$. The variation of $E_{ij}$ tells us that

$$g_1 E_{ij} + g_2\delta_{ij}E_{kk} = \dot{A}_{ij} - \partial_i\partial_j\phi =: F_{0ij}\,. \tag{4.40}$$

We first take the trace of this quantity

$$(g_1 + dg_2)E_{ii} = \dot{A}_{ii} - \partial_i\partial_i\phi = F_{0ii}\,. \tag{4.41}$$

When $g_1 + dg_2$ is nonzero, we can algebraically solve for $E_{ij}$

$$g_1 E_{ij} = F_{0ij} - \frac{g_2}{g_1 + dg_2}\delta_{ij}F_{0kk}\,. \tag{4.42}$$

We discuss the case when $g_1 + dg_2 = 0$ in Section 4.7. Having solved for $E_{ij}$ using its equation of motion, we may substitute it back into the phase space Lagrangian associated with the action (4.35b) to obtain

$$\mathcal{L}[A_{ij},\phi] = \frac{g_1}{2}E_{ij}E_{ij} + \frac{g_2}{2}E_{ii}{}^2 - \frac{h_1}{4}F_{ijk}F_{ijk} - \frac{h_2}{2}F_{ijj}F_{ikk} \tag{4.43a}$$

$$= \frac{1}{2g_1}F_{0ij}F_{0ij} - \frac{g_2}{2g_1(g_1+dg_2)}(F_{0ii})^2 - \frac{h_1}{4}F_{ijk}F_{ijk} - \frac{h_2}{2}F_{ijj}F_{ikk}\,. \tag{4.43b}$$

This is the analogue of the Maxwell Lagrangian $\frac{1}{2}F_{0i}F_{0i} - \frac{c^2}{4}F_{ij}F_{ij}$ where $c$ is the speed of light.

We can learn a few simple facts from dimensional analysis. Both $g_1 E_{ij} E_{ij}$ and $E_{ij} \dot{A}_{ij}$ have dimensions of energy density. Furthermore, $g_1 E_{ij}$ and $\dot{A}_{ij}$ have the same dimension. Only the dimension of the product of $g_1^{1/2}$ and $E_{ij}$ is determined. Without loss of generality we can take $g_1$ to be dimensionless. Then so must be $g_2$. The dimensions of $h_1$ and $h_2$ are then velocity squared. We can write equation (4.43b) as

$$\mathcal{L}[A_{ij}, \phi] = \frac{1}{g_1} \left[ \frac{1}{2} \left( F_{0ij} - \frac{1}{d} \delta_{ij} F_{0ii} \right)^2 + \frac{g_1}{2d(g_1 + dg_2)} (F_{0ii})^2 \right.$$
$$\left. - \frac{g_1 h_1}{4} F_{ijk}^2 - \frac{g_1 h_2}{2} F_{ijj} F_{ikk} \right], \tag{4.44}$$

where we factored out the parameter $1/g_1$. We can think of $g_1$ as a charge. When we gauged the complex scalar we fixed the charge by saying that $\delta\Phi = i\Lambda\Phi$. We could have said it has charge $e$ and $\delta\Phi = ie\Lambda\Phi$ with $g_1 = 1$. Alternatively we keep $\delta\Phi = i\Lambda\Phi$ but then $g_1 = e^2$. The two perspectives are related by rescaling the gauge fields $\phi$ and $A_{ij}$.

From the kinetic terms we find that the scaling dimensions of $\phi$ and $A_{ij}$ are $(d + z - 4)/2$ and $(d - z)/2$, respectively. The scaling dimension of the magnetic terms $F_{ijk} F_{ijk}$ and $F_{ijj} F_{ikk}$ is $2 + d - z$. The magnetic terms are relevant for $z > 1$. We can add quartic terms in $F_{ijk}$ as relevant terms when $2(2 + d - z) < d + z$, i.e., when $z > (4 + d)/3$. When $z = (4 + d)/3$ these terms are marginal which incidentally is the value of $z$ for which the $\lambda$, $\tilde{\lambda}$ scalar field theory (2.43) is scale invariant.

## 4.5   $3 + 1$ dimensions

In three spatial dimensions the magnetic field introduced in (4.31) is given by

$$B_{ij} = \frac{1}{2} \epsilon_{imn} F_{mnj} = \epsilon_{imn} \partial_m A_{nj}. \tag{4.45}$$

For $d = 3$, our result (4.33) implies that the Hamiltonian becomes

$$\mathcal{H} = \frac{1}{2} \left( g_1 E_{ij} E_{ij} + g_2 E_{ii}^2 + \tilde{h}_1 B_{ij} B_{ij} + \tilde{h}_2 B_{ij} B_{ji} \right), \tag{4.46}$$

where $\tilde{h}_1 = h_1 + h_2$ and $\tilde{h}_2 = -h_2$.

However, in three dimensions we can also write down an additional term that fulfills our requirements (as listed in section 4.3), namely

$$\mathcal{H}^\theta = \theta E_{ij} B_{ij}. \tag{4.47}$$

We will now show that this term is related to the $\theta$ term of [50] which is relevant for a higher spin Witten effect, however we arrive at this term from a complementary

perspective. The following discussion mirrors again the one of electrodynamics, c.f., Appendix A.2.

We start by adding the $\mathcal{H}$ and $\mathcal{H}^\theta$ term and by completing a square we arrive at

$$\mathcal{H}^{d=3} = \frac{1}{2}\left[\left(\sqrt{g_1}E_{ij} + \frac{\theta}{\sqrt{g_1}}B_{ij}\right)^2 + g_2 E_{ii}^2 + \left(\tilde{h}_1 - \frac{\theta^2}{g_1}\right)B_{ij}B_{ij} + \tilde{h}_2 B_{ij}B_{ji}\right]. \tag{4.48}$$

Next we apply the canonical transformation

$$P_{ij} = E_{ij} + \frac{\theta}{g_1}B_{ij}[A] \qquad\qquad Q_{ij} = A_{ij}, \tag{4.49}$$

where the square bracket indicates that the magnetic tensor is the one of the $A_{ij}$ fields. It is of the schematic form of a canonical transformation $p\dot{q} - H(p,q) = P\dot{Q} - K(Q,P) + \dot{F}$ with generating function $F$ as can be seen from

$$E_{ij}(\dot{A}_{ij} - \partial_i\partial_j\phi) - \mathcal{H}^{D=3} = P_{ij}(\dot{Q}_{ij} - \partial_i\partial_j\phi) - \mathcal{K} - \frac{\theta}{2g_1}(\partial_0 F^0 + \partial_i F^i). \tag{4.50}$$

The new Hamiltonian is given by

$$\mathcal{K} = \frac{1}{2}\left[g_1 P_{ij}P_{ij} + \left(\tilde{h}_1 - \frac{\theta^2}{g_1}\right)B_{ij}[Q]B_{ij}[Q] + \tilde{h}_2 B_{ij}[Q]B_{ji}[Q] + g_2 P_{ii}^2\right]. \tag{4.51}$$

and the boundary term is of the form

$$F^0 = Q_{ij}B_{ij} \qquad\qquad F^i = \epsilon_{ijk}Q_{jl}\dot{Q}_{kl} - 2B_{ij}\partial_j\phi, \tag{4.52}$$

which we can also write as $\partial_0 F^0 + \partial_i F^i = 2B_{ij}(\dot{Q}_{ij} - \partial_i\partial_j\phi)$ as was already shown in [50].

## 4.6   2 + 1 dimensions

While we have not studied the case of $2+1$ dimensions in detail, we would like to mention the possibility of fracton Chern–Simons like theories,

$$\mathcal{L} = \frac{k}{4\pi}\epsilon_{ij}\left(A_{ik}\dot{A}_{jk} + \phi\partial_k F_{ijk}\right), \tag{4.53}$$

where $i,j,k = 1,2$. These are not actual Chern–Simons theories, though, as their coupling to curved backgrounds requires the introduction of metric data. Note that (4.53) is independent of the trace of $A_{ij}$. See, e.g., [12, Section II.B.3] and references therein for more details.

Furthermore, we note that the magnetic part of the Hamiltonian (4.30) in $2+1$ dimensions can be written as

$$\mathcal{H}_{\text{mag}} = \frac{h_1}{4}F_{ijk}F_{ijk} + \frac{h_2}{2}F_{ijj}F_{ikk} = \frac{1}{2}(h_1 + h_2)(F_{122}^2 + F_{121}^2). \tag{4.54}$$

Thus, the magnetic theory only has one coupling constant, which must satisfy

$$h_1 + h_2 > 0 \,. \tag{4.55}$$

for the theory to be non-trivial and bounded from below. As we show in the section below, this rules out the existence of the traceless theory in $d = 2$.

## 4.7 Traceless scalar charge gauge theory

As we will show in this section, rotational symmetry allows for additional terms in the Lagrangian that describes the scalar charge gauge theory. These terms modify the Poisson brackets and thus the gauge transformations and, depending on the details of these terms, a priori result in three general classes of theories. The first and most important such class is the traceless scalar charge gauge theory, so called because it is independent of $A_{ii}$, i.e.,

$$\delta_{ij} \frac{\delta \mathcal{L}[A_{ij}, E_{ij}, \phi]}{\delta A_{ij}} = 0 \,. \tag{4.56}$$

This theory has an additional conserved quantity in the form of the trace of the quadrupole moment, and it has played a prominent role in the fracton literature; in particular, it was shown in [31] that these theories can be put on curved space where the geometry on constant time slices is some space of constant sectional curvature, and where time is absolute (for more details see the next section). The second class generalises the traceless theory by allowing for trace-dependence while the gauge transformation is identical to the one of the traceless theory. This theory depends on one parameter that measures the dependence on the trace, which has the interpretation of an additional scalar. Thus, as we discuss in Appendix B, this theory is just the traceless theory of case 1 coupled to a scalar in the guise of the trace. The third and final class of theories have what at first glance appears to be a different set of gauge symmetries than the other cases, that depend on two parameters, but as we show in Appendix B, this third case is equivalent to the original theory (4.43b).

The Lagrangian we considered above may be generalised by including two additional terms parameterised by two real constants $c_1$ and $c_2$:

$$\begin{aligned}
\mathcal{L}[A_{ij}, E_{ij}, \phi] &= (E_{ij} + c_1 \delta_{ij} E_{kk})\dot{A}_{ij} - \mathcal{H} - \phi(\partial_i \partial_j E_{ij} + c_2 \partial_i \partial_i E_{jj}) + \partial_i K_i \\
&= E_{ij}(\dot{A}_{ij} + c_1 \delta_{ij} \dot{A}_{kk} - \partial_i \partial_j \phi - c_2 \delta_{ij} \partial_k \partial_k \phi) - \mathcal{H} \,, \tag{4.57}
\end{aligned}$$

where the boundary term $K_i$ is

$$K_i = \phi \partial_j E_{ij} - \partial_j \phi E_{ij} + c_2(\phi \partial_i E_{jj} - \partial_i \phi E_{jj}) \,, \tag{4.58}$$

and where $\mathcal{H}$ is an appropriately chosen invariant Hamiltonian, which has the same functional form as (4.30) when written in terms of the electric field $E_{ij}$ and the magnetic field strengths $F_{ijk}$.

The new phase space Lagrangian (4.57) differs in two respects from the one discussed previously in (4.35a). The parameter $c_1$ modifies the Poisson brackets and the parameter $c_2$ modifies the Gauss constraint. Both deformations are compatible with the underlying Aristotelian symmetries (time and space translations and spatial rotations).

The term $(E_{ij} + c_1\delta_{ij}E_{kk})\dot{A}_{ij}$ modifies the pre-symplectic potential and hence the symplectic form on phase space and by inverting this new symplectic form we obtain the modified Poisson brackets. The "symplectic" term in the phase space Lagrangian can be written as

$$E_{kl}(\delta_{i(k}\delta_{l)j} + c_1\delta_{ij}\delta_{kl})\dot{A}_{ij}\,. \tag{4.59}$$

To determine the Poisson bracket, we need to invert the quantity in parentheses in the expression above, see, e.g., [23, 24]. This produces the bracket

$$\{A_{ij}(\vec{x}), E_{kl}(\vec{y})\} = \left(\delta_{i(k}\delta_{l)j} - \frac{1}{d}\delta_{ij}\delta_{kl}\right)\delta(\vec{x} - \vec{y}) \tag{4.60}$$

for $c_1 = -1/d$ and

$$\{A_{ij}(\vec{x}), E_{kl}(\vec{y})\} = \left(\delta_{i(k}\delta_{l)j} - \frac{c_1}{1 + dc_1}\delta_{ij}\delta_{kl}\right)\delta(\vec{x} - \vec{y}) \tag{4.61}$$

for $c_1 \neq -1/d$.

The Gauss constraint is the generator of gauge transformations. The gauge transformation generated by the constraint imposed by $\phi$ of some function $F$ on phase space is given by

$$\delta_\Lambda F = \{F, \int d^d x \ \Lambda(\partial_i\partial_j E_{ij} + c_2\partial_i\partial_i E_{jj})\}\,, \tag{4.62}$$

so that when $c_1 \neq -1/d$ and $c_2 \neq -1/d$ we have

$$\delta_\Lambda A_{ij} = \partial_i\partial_j\Lambda + \delta_{ij}\frac{c_2 - c_1}{1 + dc_1}\partial^2\Lambda\,, \qquad \delta_\Lambda E_{ij} = 0\,, \tag{4.63}$$

while for $c_1 = -1/d$ with $c_2$ arbitrary, as well as for the case $c_1 \neq -1/d$ with $c_2 = -1/d$, we get

$$\delta_\Lambda A_{ij} = \partial_i\partial_j\Lambda - \frac{1}{d}\delta_{ij}\partial^2\Lambda\,, \qquad \delta_\Lambda E_{ij} = 0\,. \tag{4.64}$$

Note that in the latter case, the trace $A_{ii}$ is gauge invariant. Depending on the values of $c_1$ and $c_2$, the theory thus splits into three classes[6]. When $c_1 = c_2 = -1/d$, the

---

[6]We do not consider the case $c_1 = -1/d$ and $c_2 \neq -1/d$ as in this case the combination $\dot{A}_{ij} + c_1\delta_{ij}\dot{A}_{kk} - \partial_i\partial_j\phi - c_2\delta_{ij}\partial_k\partial_k\phi$ in (4.57) is not gauge invariant.

field strength $F_{ijk}$ as defined above is no longer invariant. Rather, it transforms as follows under (4.64)

$$\delta F_{ijk} = \frac{2}{d}\delta_{k[i}\partial_{j]}\partial^2\Lambda\,, \tag{4.65}$$

and since $A_{ii}$ is gauge invariant, we cannot redefine the field strength by adding a term to it that makes it gauge invariant. Instead, taking $\mathcal{H}$ to be given by (4.30), gauge invariance requires that the coefficients $h_1$ and $h_2$ be related as

$$h_1 = -h_2(d-1)\,. \tag{4.66}$$

This condition for $d = 2$ reads $h_1 = -h_2$, but equation (4.54) implies that in this case the magnetic terms add up to zero. Hence the traceless theory with a nontrivial magnetic sector requires $d \geq 3$.

Similarly, the electric field strength $F_{0ij}$ as defined above is no longer gauge invariant. Instead, the invariant electric field strength is now

$$\tilde{F}_{0ij} := \dot{A}_{ij} - \partial_i\partial_j\phi - \frac{1}{d}\delta_{ij}(\dot{A}_{kk} - \partial^2\phi)\,, \tag{4.67}$$

which is gauge invariant under (4.64). The Lagrangian of the traceless theory is obtained by integrating out $E_{ij}$ from (4.57), which turns out to imply the condition $g_1 + dg_2 = 0$, and produces the result

$$\mathcal{L}_{\text{traceless}}[A_{ij}, \phi] = \frac{1}{2g_1}\tilde{F}_{0ij}\tilde{F}_{0ij} - \frac{h_1}{4}F_{ijk}F_{ijk} + \frac{h_1}{2(d-1)}F_{ijj}F_{ikk}\,, \tag{4.68}$$

which is traceless in the sense of (4.56). This is intimately linked to the conservation of the trace of the quadrupole moment. Furthermore, the fact that the Lagrangian is independent of $A_{ii}$ gives rise to a Stückelberg symmetry $\delta A_{ij} = \delta_{ij}\chi$ with parameter $\chi$ that allows us to set $A_{ii} = 0$.

The remaining two cases arise when either $c_1 \neq -1/d$ and $c_2 = -1/d$, or $c_1, c_2 \neq -1/d$ but otherwise arbitrary. These cases do not give rise to new theories. This we demonstrate in Appendix B.

## 4.8   Spectrum of the scalar charge gauge theory

We now study the spectrum of the scalar charge gauge theory, starting with the traceful case. We are going to do this by analysing the Fourier decomposition of the gauge invariant objects $F_{ijk}$ and $F_{0ij}$. We will need the equations of motion as well as Bianchi identities. The Bianchi identities[7] are given by

$$\partial_{[i}F_{jk]l} = 0 \tag{4.69a}$$

$$2\partial_{[i}F_{0j]k} - \partial_0 F_{ijk} = 0\,. \tag{4.69b}$$

---

[7]The second may be checked using the explicit expressions in terms of $A_{ij}$. In order to maintain a light notation, since the 0 index in $F_{0jk}$ is fixed, we use the convention $2\partial_{[i}F_{0j]k} \equiv \partial_i F_{0jk} - \partial_j F_{0ik}$.

It should be noted that the antisymmetrisation in (4.69b) only involves $i$ and $j$ and *not* 0. The equations of motion of the traceful theory can be obtained from equations (4.38a)–(4.38c) and can be expressed as follows

$$\partial_i\partial_j F_{0ij} - \frac{g_2}{g_1 + dg_2}\partial_i\partial_i F_{0jj} = 0 \tag{4.70a}$$

$$\frac{1}{g_1}\left(\dot{F}_{0ij} - \frac{g_2}{g_1 + dg_2}\delta_{ij}\dot{F}_{0mm}\right) = h_1\partial_m F_{m(ij)} + h_2\delta_{ij}\partial_m F_{mnn} - h_2\partial_{(i}F_{j)nn} \, . \tag{4.70b}$$

The goal is to decouple the equations and find wave-like equations for the various components of $F_{0ij}$. If we take the time derivative of equation (4.70b) and apply the Bianchi identity (4.69b), we get the following equation for $F_{0ij}$

$$\partial_0^2 F_{0ij} = \left[g_1 h_2 + g_2 h_1 + (d-1)g_2 h_2\right]\delta_{ij}\left(\partial_m\partial_m F_{0ll} - \partial_m\partial_l F_{0ml}\right) + g_1 h_1\partial_m\partial_m F_{0ij}$$
$$- g_1 h_1\partial_m\partial_{(i}F_{0j)m} - g_1 h_2\partial_i\partial_j F_{0mm} + g_1 h_2\partial_m\partial_{(i}F_{0j)m} \, , \tag{4.71}$$

which no longer involves the magnetic field strength.

The Fourier transformation of the equation above gives

$$\omega^2\hat{F}_{0ij} = \left[g_1 h_2 + g_2 h_1 + (d-1)g_2 h_2\right]\delta_{ij}\left(k_m k_m \hat{F}_{0ll} - k_m k_l \hat{F}_{0ml}\right) + g_1 h_1 k^2\hat{F}_{0ij}$$
$$- g_1 h_1 k_m k_{(i}\hat{F}_{0j)m} - g_1 h_2 k_i k_j \hat{F}_{0mm} + g_1 h_2 k_m k_{(i}\hat{F}_{0j)m} \, , \tag{4.72}$$

where $\hat{F}_{0ij}(\omega, k)$ is the Fourier transform of $F_{0ij}(t, x)$. If we take the trace of this equation and apply the Gauss constraint (4.70a) we get

$$\omega^2\hat{F}_{0jj} = (g_1 + (d-1)g_2)(h_1 + (d-1)h_2)\,k^2\hat{F}_{0jj} \, . \tag{4.73}$$

It can be shown, using the strict version of the bounds for the coupling constants found in (4.34) that $(g_1 + (d-1)g_2) > 0$ and $(h_1 + (d-1)h_2) > 0$. Hence the velocity squared of this mode, i.e., $(g_1 + (d-1)g_2)(h_1 + (d-1)h_2)$ is indeed positive. We restrict ourselves to the strict versions of the inequalities involving $h_1$ and $h_2$ in order that the magnetic sector is nontrivial which is needed for propagation.

To find the rest of the modes it is useful to introduce the following projector

$$P_{ij} = \delta_{ij} - \frac{k_i k_j}{k^2} \, . \tag{4.74}$$

$P_{ij}$ projects along the directions perpendicular to $k_i$. Using this along with equations (4.70a), (4.73) and (4.72) we find the following modes

$$\omega^2(k_i k_j \hat{F}_{0ij}) = v_1^2 k^2(k_i k_j \hat{F}_{0ij}) \tag{4.75a}$$

$$\omega^2(P_{ij}\hat{F}_{0ij}) = v_1^2 k^2(P_{ij}\hat{F}_{0ij}) \tag{4.75b}$$

$$\omega^2\left(P_{im}k_n\hat{F}_{0mn}\right) = v_2^2 k^2\left(P_{im}k_n\hat{F}_{0mn}\right) \tag{4.75c}$$

$$\omega^2(P_{li}P_{nj} - \frac{1}{(d-1)}P_{ln}P_{ij})\hat{F}_{0ij} = v_3^2 k^2(P_{li}P_{nj} - \frac{1}{(d-1)}P_{ln}P_{ij})\hat{F}_{0ij} \, , \tag{4.75d}$$

where the velocities are given by

$$v_1^2 = (g_1 + (d-1)g_2)(h_1 + (d-1)h_2) \tag{4.76a}$$

$$v_2^2 = \frac{1}{2}g_1(h_1 + h_2) \tag{4.76b}$$

$$v_3^2 = g_1 h_1 . \tag{4.76c}$$

It follows from the strict versions of the inequalities in (4.34) that $g_1(h_1 + h_2) > 0$ as well as $g_1 h_1 > 0$, so we see that we get three classes of modes with three different velocities. We also know that the Gauss constraint in (4.70a) relates $k_i k_j \hat{F}_{0ij}$ to $P_{ij}\hat{F}_{0ij}$. Using this we find that there are $d(d+1)/2 - 1$ independent modes, as is to be expected.

We have exclusively focused on the electric sector in this analysis but one can see from the Bianchi identities and the equations of motions that an oscillating electric field strength leads to an oscillating magnetic field strength. Furthermore, we observe that there is no universal velocity as the velocities are not all equal which chimes well with the earlier observation that these fields are defined on an Aristotelian geometry. It would be interesting to study the energy-momentum tensor for these theories and the different states of polarisation.

Next, we turn to the traceless case whose Lagrangian is given in (4.68). Now the equations of motion are given by

$$0 = \partial_i \partial_j \tilde{F}_{0ij} \tag{4.77a}$$

$$0 = \frac{1}{g_1}\partial_0 \tilde{F}_{0ij} - h_1 \partial_m F_{m(ij)} + \frac{h_1}{d-1}(\delta_{ij}\partial_l F_{lmm} - \partial_{(i}F_{j)ll}) , \tag{4.77b}$$

where $\tilde{F}_{0ij}$ is defined in equation (4.67). We now express the Bianchi identities in terms of $\tilde{F}_{0ij}$

$$\partial_{[i}F_{jk]l} = 0 \tag{4.78a}$$

$$2\partial_{[i}\tilde{F}_{0j]k} + \frac{1}{d-1}\left(\delta_{jk}\partial_l \tilde{F}_{0il} - \delta_{ik}\partial_l \tilde{F}_{0jl}\right) = \dot{F}_{ijk} - \frac{2}{d-1}\delta_{k[j}\dot{F}_{i]ll} . \tag{4.78b}$$

If we differentiate (4.77b) with respect to time and apply (4.78b) and (4.77a) we get

$$\partial_0^2 \tilde{F}_{0ij} = g_1 h_1 \left(\partial_m \partial_m \tilde{F}_{0ij} - \frac{d}{d-1}\partial_m \partial_{(i}\tilde{F}_{0j)m}\right) . \tag{4.79}$$

The Fourier transformation of this equation is given by

$$\omega^2 \check{F}_{0ij} = g_1 h_1 \left(k^2 \check{F}_{0ij} - \frac{d}{d-1}k_m k_{(i}\check{F}_{0j)m}\right) , \tag{4.80}$$

where we have defined $\check{F}_{0ij}(\omega, k)$ to be the Fourier transform of $\tilde{F}_{0ij}(t, x)$. This then leads to the following decomposition of the modes

$$\omega^2(k_j\check{F}_{0ij}) = \frac{d-2}{2(d-1)}v_3^2 k^2(k_j\check{F}_{0ij}) \tag{4.81}$$

$$\omega^2(P_{im}P_{jn}\check{F}_{0mn}) = v_3^2 k^2(P_{im}P_{jn}\check{F}_{0mn}), \tag{4.82}$$

where $v_3^2$ is given in equation (4.76c). In order to arrive at this result we have used that the Gauss constraint in (4.77a) is given by $k_i k_j \check{F}_{0ij} = 0$ when expressed in momentum space. Due to the tracelessness of $\check{F}_{0ij}$ this also means $P_{ij}\check{F}_{0ij} = 0$. Using this we find that there are $d(d+1)/2 - 2$ independent modes.

As can be seen from the dispersion relations above something special happens for $d = 2$. There is only 1 degree of freedom and it is given by $k_j\check{F}_{0ij}$ but it does not propagate. This is related to the fact that $\mathcal{H}_{\mathrm{mag}} = 0$ for the traceless case in $d = 2$. We can see this from equation (4.54) in combination with the condition in (4.66).

## 4.9 Similarities and differences with partially massless gravitons

The gauge structure of scalar charge theory bears a striking resemblance to the linear theory of partially massless gravitons [25, 26], although they are not the same theories. In this section, we elucidate the similarities and the differences between these theories (we follow Section 1 of [51]).

Theories of partially massless gravitons were originally developed to address the cosmological constant problem (i.e., why the cosmological constant is small and nonzero by relating its value to the mass of a massive graviton via a gauge symmetry).

On a maximally symmetric curved spacetime, there exists the possibility of considering particles which are neither fully massive nor fully massless. In particular, in de Sitter space, where such theories where first developed, it was observed that a theory of gravitons with more degrees of freedom than a massless theory, but fewer than in a theory of massive gravity, could be written down [25]. Concretely, such theories are obtained from massive theories of gravity by imposing a scalar gauge symmetry that removes one degree of freedom.

Let us now describe the linear theory of partially massless gravitons using the Stückelberg field approach of [52], which is almost identical to the Stückelberg approach of Sections 2.7 and 4.2 (see for further details [51]).

The dynamics of a massive graviton $\tilde{H}_{\mu\nu}$ of mass $m$ on a $(3 + 1)$-dimensional maximally symmetric Lorentzian background with metric $\bar{g}_{\mu\nu}$ is described by Fierz–Pauli theory [53]. For generic $m^2 \neq 0$, this has five degrees of freedom, while for $m^2 = 0$, the now massless graviton field $\tilde{H}$ enjoys linearised diffeomorphism invariance, which leads to the two degrees of freedom of a massless graviton.

Regardless of the value of $m^2$, we can introduce gauge redundancy into the theory via Stückelberg fields $\mathcal{A}_\mu$ and $\psi$, in terms of which we write $\tilde{H}_{\mu\nu}$ as

$$\tilde{H}_{\mu\nu} = H_{\mu\nu} + \bar{\nabla}_{(\mu}\mathcal{A}_{\nu)} + \bar{\nabla}_\mu \bar{\nabla}_\nu \psi \,, \tag{4.83}$$

where $\bar{\nabla}$ is the Levi-Civita connection of $\bar{g}$. The new gauge symmetries $\Sigma$ and $\Lambda$ act as

$$\delta H_{\mu\nu} = \bar{\nabla}_{(\mu}\Sigma_{\nu)} \qquad\qquad \delta\mathcal{A}_\mu = \bar{\nabla}_\mu\Lambda - \Sigma_\mu \qquad\qquad \delta\psi = -\Lambda \,. \tag{4.84}$$

It can be shown [51] that after performing a field redefinition that untangles $\phi$ and $H_{\mu\nu}$, and then writing the theory in terms of this redefined $H'_{\mu\nu}$, for the special choice of background Ricci scalar $\bar{R} = 6m^2$ the action becomes independent of the field $\psi$. As in Section 2.7, we can then gauge fix $\mathcal{A}_\mu = 0$ (while keeping the field $\psi$ free), leading to

$$\delta H'_{\mu\nu} = \bar{\nabla}_\mu \bar{\nabla}_\nu \Lambda + \frac{m^2\Lambda}{d-1}\bar{g}_{\mu\nu} \,, \tag{4.85}$$

on a $(d+1)$-dimensional background.

Although this procedure is very similar to what we described in Sections 2.7 and 4.2, there are some crucial differences. First and foremost, the additional Stückelberg field $\psi$ is a new ingredient that is not part of the construction in (2.70), and the gauge transformation itself is also different since $H'_{\mu\nu}$ contains a term linear in $\Lambda$ that has no derivatives acting on $\Lambda$. Second, the role of time is different: the partially massless graviton $H_{\mu\nu}$ has both temporal and spatial components, while the fracton gauge field $A_{ij}$ only has spatial components. In the same vein, there is no analogue of the Lagrange multiplier $\phi$ in the theory of partially massless gravitons.

It would be interesting to explore this analogy further. In particular, there is a non-linear theory of partially massless gravitons (see, e.g., [51]), and it could be worthwhile to investigate if a similar construction exists for fractons.

## 5    Aristotelian geometry

For the remainder of this paper we will concern ourselves with coupling the scalar field theory and the scalar charge gauge theory to curved spacetime. As explained in Section 2.6, the proper geometric framework is that of Aristotelian geometry, the details of which we provide in this section.

The motivations to place these theories on a curved spacetime are the same as for relativistic field theories. In no particular order – and without being exhaustive – understanding the coupling to curved space helps with computing correlation functions of for example the energy momentum tensor, it can aid the search for Weyl-type anomalies, it helps with formulating a theory of fluid dynamics that obeys the same conservation equations, etc.

Originally coined by Penrose [54], Aristotelian geometry captures the geometry of absolute time and space. In the context of gravitational theories, Aristotelian geometry plays the same role in Hořava–Lifshitz gravity, see e.g., [55, 56], and Einstein–æther theory[8] [58] as Lorentzian geometry plays in Einstein gravity.

## 5.1 Geometric data

The first systematic treatment of Aristotelian geometry in the formulation we will employ was given in [34], where it was used in the description of boost-agnostic fluids. An Aristotelian geometry on a $(d+1)$-dimensional manifold $\mathcal{M}$ consists of a 1-form $\tau_\mu$ – the clock form – and a co-rank 1 symmetric tensor $h_{\mu\nu}$ of Euclidean signature, whose kernel is spanned by a vector $v^\mu$, i.e., $h_{\mu\nu}v^\nu = 0$. As above, Greek indices $\mu, \nu, \ldots = 0, \ldots, d$ are spacetime indices. The degeneracy of $h_{\mu\nu}$ implies the following decomposition

$$h_{\mu\nu} = \delta_{ab}e_\mu^a e_\nu^b, \tag{5.1}$$

where $a, b = 1, \ldots, d$ are purely spatial tangent space indices, where the vielbeins $e_\mu^a$ transform under local $SO(d)$ rotations. Crucially, neither $h_{\mu\nu}$ nor $\tau_\mu$ are assigned particular tangent space transformations. Hence, Aristotelian geometry can be viewed as a "proto-geometry" in the sense that Lorentzian, Galilean and Carrollian geometries all arise from Aristotelian geometry via the introduction of the appropriate boost symmetry.[9] Together $(\tau_\mu, e_\mu^a)$ form a square matrix with inverse $(v^\mu, e_a^\mu)$, where the following relations are satisfied

$$e_a^\mu e_\mu^b = \delta_a^b \qquad v^\mu e_\mu^a = 0 = \tau_\mu e_a^\mu \qquad v^\mu \tau_\mu = -1 \qquad e_\mu^a e_a^\nu - v^\nu \tau_\mu = \delta_\mu^\nu. \tag{5.2}$$

The last of these relations – the completeness relation – will prove particularly useful in our considerations of Aristotelian geometry below. The volume form is locally given by $\mathrm{vol} = e\, d^{d+1}x$, where $e$ is the determinant of $(\tau_\mu, e_\mu^a)$.

At this stage, let's explicitly exhibit the equivalence between Aristotelian geometry and the geometric description of Einstein–æther theory. In [58], Einstein–æther theory is described by a metric $g_{\mu\nu}$ and a vector $u^\mu$ satisfying $g_{\mu\nu}u^\mu u^\nu = -1$. By defining $u_\mu = g_{\mu\nu}u^\nu$, we can formally identify $u^\mu = v^\mu$, $u_\mu = \tau_\mu$ and $g_{\mu\nu} + u_\mu u_\nu = h_{\mu\nu}$, which completes the identification between Aristotelian geometry and the geometric data of Einstein–æther theory.

## 5.2 Aristotelian connections and intrinsic torsion

We now seek an affine connection satisfying the following Aristotelian analogue of metric compatibility

$$\nabla_\mu \tau_\nu = \nabla_\mu h_{\nu\rho} = 0, \tag{5.3}$$

---

[8]For the relation between Hořava–Lifshitz gravity and Einstein-æther theory, see [57].

[9]Sometimes the realisation of the boost symmetry is accompanied by the introduction of additional gauge fields, such as the mass gauge field in Newton–Cartan geometry.

which, via the completeness relation (5.2), also imply that

$$\nabla_\mu v^\nu = \nabla_\mu h^{\nu\rho} = 0 \,. \tag{5.4}$$

Under infinitesimal general coordinate transformations parameterised by $\xi^\mu$, an affine connection $\Gamma$ transforms as

$$\delta_\xi \Gamma^\rho_{\mu\nu} = \pounds_\xi \Gamma^\rho_{\mu\nu} + \partial_\mu \partial_\nu \xi^\rho \,, \tag{5.5}$$

where $\pounds_\xi \Gamma^\rho_{\mu\nu}$ represents the tensorial part of the transformation. In terms of the Aristotelian data, this transformation property can be achieved by the following affine connection

$$\Gamma^\rho_{\mu\nu} = -v^\rho \partial_\mu \tau_\nu + \frac{1}{2} h^{\rho\lambda} \left( \partial_\mu h_{\lambda\nu} + \partial_\nu h_{\lambda\mu} - \partial_\lambda h_{\mu\nu} \right) + Y^\rho_{\mu\nu} \,, \tag{5.6}$$

where $Y^\rho_{\mu\nu}$ is an arbitrary tensor, and where the first two terms are required to obtain the non-tensorial piece $\partial_\mu \partial_\nu \xi^\rho$ in (5.5). Imposing (5.3) leads to constraints on the tensor $Y$. Starting with the condition $\nabla_\mu \tau_\nu = 0$, we find that

$$0 = \nabla_\mu \tau_\nu = \partial_\mu \tau_\nu - \Gamma^\rho_{\mu\nu} \tau_\rho \Rightarrow Y^\rho_{\mu\nu} \tau_\rho = 0 \,. \tag{5.7}$$

Similarly, the condition $\nabla_\mu h_{\nu\rho} = 0$ translates to

$$0 = \nabla_\mu h_{\nu\rho} = \partial_\mu h_{\nu\rho} - 2\Gamma^\lambda_{\mu(\nu} h_{\rho)\lambda} = -2\tau_{(\rho} K_{\nu)\mu} - 2Y^\lambda_{\mu(\nu} h_{\rho)\lambda} \,, \tag{5.8}$$

where

$$K_{\mu\nu} = -\frac{1}{2} \pounds_v h_{\mu\nu} \tag{5.9}$$

is the extrinsic curvature,[10] which satisfies

$$v^\mu K_{\mu\nu} = 0 \,. \tag{5.10}$$

The property (5.8) thus implies that

$$Y^\lambda_{\mu\nu} = -h^{\lambda\kappa} \tau_\nu K_{\mu\kappa} + C^\lambda_{\mu\nu} \,, \tag{5.11}$$

with

$$C^\lambda_{\mu(\nu} h_{\rho)\lambda} = 0 \,. \tag{5.12}$$

In summary, metric compatibility in the sense of (5.3) can be achieved with the following affine connection (which also featured in [34])

$$\Gamma^\rho_{\mu\nu} = -v^\rho \partial_\mu \tau_\nu + \frac{1}{2} h^{\rho\lambda} \left( \partial_\mu h_{\lambda\nu} + \partial_\nu h_{\lambda\mu} - \partial_\lambda h_{\mu\nu} \right) - h^{\rho\lambda} \tau_\nu K_{\mu\lambda} + C^\rho_{\mu\nu} \,, \tag{5.13}$$

---

[10]The name "extrinsic curvature" is perhaps a bit of a misnomer, since in general it is not an extrinsic curvature of anything. In the case where $\tau$ obeys the Frobenius condition $\tau \wedge d\tau = 0$, and so defines a foliation, $K_{\mu\nu}$ becomes the extrinsic curvature of the leaves of the foliation.

where the tensor $C^{\rho}_{\mu\nu}$ is such that

$$C^{\rho}_{\mu\nu}\tau_{\rho} = 0 \qquad\qquad C^{\rho}_{\mu(\nu}h_{\lambda)\rho} = 0\,. \qquad\qquad (5.14)$$

This is a torsionful connection with torsion given by

$$2\Gamma^{\rho}_{[\mu\nu]} = -v^{\rho}\tau_{\mu\nu} + 2h^{\rho\lambda}\tau_{[\mu}K_{\nu]\lambda} + 2C^{\rho}_{[\mu\nu]} =: T^{\rho}_{\mu\nu} + 2C^{\rho}_{[\mu\nu]}\,, \qquad (5.15)$$

where we defined

$$\tau_{\mu\nu} = 2\partial_{[\mu}\tau_{\nu]}\,. \qquad\qquad (5.16)$$

In the language of [59], the intrinsic torsion of an Aristotelian geometry[11] is precisely captured by $\tau_{\mu\nu}$ and $K_{\mu\nu}$ – in other words, the intrinsic torsion is $T^{\rho}_{\mu\nu}$. If we require that the connection we employ is *minimal* in the sense that the torsion is given only by the intrinsic torsion, we must have $C^{\rho}_{[\mu\nu]} = 0$. In this case the conditions in (5.14) imply that $C^{\rho}_{\mu\nu} = 0$, that is to say, the symmetric part of the $C$ tensor vanishes as well. To see this, note that the second equation of (5.14) implies that $C^{\rho}_{\mu\nu}v^{\nu} = 0$ and the fact that $C^{\rho}_{[\mu\nu]} = 0$ tells us that also $C^{\rho}_{\mu\nu}v^{\mu} = 0$. The first equation of (5.14) implies that $C^{\rho}_{\mu\nu} = h^{\rho\lambda}C_{\lambda\mu\nu}$. Hence, without loss of generality we can assume that $C_{\lambda\mu\nu}$ is entirely spatial, i.e., all contractions with $v^{\rho}$ vanish. In terms of $C_{\lambda\mu\nu}$ the second equation of (5.14) and the vanishing of the intrinsic torsion tell us that $C_{(\mu\nu)\rho} = C_{\mu[\nu\rho]} = 0$. These two conditions can only be satisfied if $C_{\mu\nu\rho} = 0$. Hence, demanding that the torsion is intrinsic and that the affine connection is metric compatible leads to our final result for the connection

$$\Gamma^{\rho}_{\mu\nu} = -v^{\rho}\partial_{\mu}\tau_{\nu} + \frac{1}{2}h^{\rho\lambda}\left(\partial_{\mu}h_{\lambda\nu} + \partial_{\nu}h_{\lambda\mu} - \partial_{\lambda}h_{\mu\nu}\right) - h^{\rho\lambda}\tau_{\nu}K_{\mu\lambda}\,. \qquad (5.17)$$

A particularly simple class of Aristotelian geometries are those with vanishing intrinsic torsion, namely those for which

$$d\tau = 0\,, \qquad K_{\mu\nu} = 0\,. \qquad\qquad (5.18)$$

In this case $\tau$ is locally exact: $\tau = dt$, but we will assume that this is true globally so that there is a foliation of the geometry where each leaf is described by $t = $ constant with $t$ being the absolute time. In this case the elapsed time $T_1 = \int_{\gamma_1}\tau$ along a given path $\gamma_1$ with fixed endpoints is the same as the elapsed time $T_2 = \int_{\gamma_2}\tau$ along any other path $\gamma_2$ with the same endpoints.

**ADM type description of torsion-free Aristotelian geometry**

We can write the torsion-free Aristotelian data in ADM type variables, i.e.,

$$\tau = dt \qquad h = h_{ij}\left(dx^i + N^i dt\right)\left(dx^j + N^j dt\right) \qquad v = -\partial_t + N^i\partial_i \qquad (5.19)$$

---

[11]Aristotelian geometry can be viewed as the intersection of Carroll and Newton–Cartan geometry, which have intrinsic torsion described by $K_{\mu\nu}$ and $\tau_{\mu\nu}$, respectively [59].

and $h^{tt} = h^{ti} = 0$ with $h^{ij}$ the inverse of $h_{ij}$. The extrinsic curvature $K_{\mu\nu} = -\frac{1}{2}\mathcal{L}_v h_{\mu\nu}$ is then given by

$$K_{ij} = \frac{1}{2}\partial_t h_{ij} - \frac{1}{2}\mathcal{L}_N h_{ij} \,, \tag{5.20}$$

where the second Lie derivative is a $d$-dimensional Lie derivative along $N^i$. The other components follow from $v^\mu K_{\mu\nu} = 0$. When the intrinsic torsion vanishes we have that $K_{ij} = 0$. Equation (5.19) is obviously not the most general ADM-type parametrisation of an Aristotelian geometry which would have a general unconstrained $\tau$ and $h_{ij}$.

## 5.3  Field theory on Aristotelian backgrounds

Consider a generic field theory described by the action $S[\Phi; \tau_\mu, h_{\mu\nu}]$ with field content abstractly denoted by $\Phi$ on a $(d+1)$-dimensional Aristotelian background given by $\tau_\mu$ and $h_{\mu\nu}$. The variation of the action (see also [34]) is given by

$$\delta S[\Phi; \tau_\mu, h_{\mu\nu}] = \int d^{d+1}x \, e \left( -T^\mu \delta\tau_\mu + \frac{1}{2}T^{\mu\nu}\delta h_{\mu\nu} + \mathcal{E}_\Phi \delta\Phi \right) \,, \tag{5.21}$$

where $T^\mu$ is the energy current, $T^{\mu\nu}$ the momentum-stress tensor[12] and $\mathcal{E}_\Phi$ is the Euler-Lagrange equation for $\Phi$, and where $\delta\tau_\mu$, $\delta h_{\mu\nu}$ and $\delta\Phi$ are arbitrary variations. For simplicity we have assumed that $\Phi$ is a scalar. Out of the energy current and the momentum-stress tensor, we can build the energy-momentum tensor $T^\mu{}_\nu$, which is a $(1,1)$-tensor given by [34]

$$T^\mu{}_\nu = -T^\mu \tau_\nu + T^{\mu\rho} h_{\rho\nu} \,. \tag{5.22}$$

Invariance of the action (5.21) under general coordinate transformations infinitesimally parameterised by the vector $\xi^\mu$ implies that

$$\delta_\xi S = \int d^{d+1}x \, e \left( -T^\mu \delta_\xi \tau_\mu + \frac{1}{2}T^{\mu\nu}\delta_\xi h_{\mu\nu} + \mathcal{E}_\Phi \delta_\xi \Phi \right) = 0 \,, \tag{5.23}$$

leading to the Ward identity

$$0 = e^{-1}\partial_\mu(eT^\mu{}_\nu) + T^\mu \partial_\nu \tau_\mu - \frac{1}{2}T^{\mu\rho}\partial_\nu h_{\mu\rho} - \mathcal{E}_\Phi \partial_\nu \Phi \,. \tag{5.24}$$

This can also be written in the following form

$$0 = \nabla_\mu T^\mu{}_\nu - \Gamma^\mu_{[\mu\sigma]}T^\sigma{}_\nu + \Gamma^\sigma_{[\mu\nu]}T^\mu{}_\sigma - \mathcal{E}_\Phi \partial_\nu \Phi \,, \tag{5.25}$$

---

[12]Since $v^\mu h_{\mu\nu} = 0$ the momentum-stress tensor is determined up to a term proportional to $v^\mu v^\nu$. The projection of $T^{\mu\nu}$ along $\tau_\mu$ and $h_{\nu\rho}$ gives the momentum, while the projection along $h_{\mu\rho}h_{\nu\sigma}$ gives the stress tensor which is symmetric as a result of the symmetry of $T^{\mu\nu}$.

where we used the connection given in (5.13) that satisfies the Aristotelian analogue of metric compatibility. On shell, using the equation of motion of the matter fields $\Phi$, the Ward identity becomes

$$0 = \nabla_\mu T^\mu{}_\nu - \Gamma^\mu_{[\mu\sigma]} T^\sigma{}_\nu + \Gamma^\sigma_{[\mu\nu]} T^\mu{}_\sigma \,, \tag{5.26}$$

which expresses energy-momentum conservation.

If our theory enjoys anisotropic Weyl invariance, we once more obtain a corresponding Ward identity. Under anisotropic Weyl transformations infinitesimally parameterised by $\Omega$, the fields $\tau_\mu$, $h_{\mu\nu}$ and $\Phi$ transform as

$$\delta_\Omega \tau_\mu = z\Omega\tau_\mu \qquad \delta_\Omega h_{\mu\nu} = 2\Omega h_{\mu\nu} \qquad \delta_\Omega \Phi = -D_\Phi \Omega \Phi \,, \tag{5.27}$$

where $D_\Phi$ is the scaling dimension of $\Phi$. Invariance amounts to the statement that

$$\delta_\Omega S = \int d^{d+1}x \, e \left( -T^\mu \delta_\Omega \tau_\mu + \frac{1}{2} T^{\mu\nu} \delta_\Omega h_{\mu\nu} + \mathcal{E}_\Phi \delta_\Omega \Phi \right) = 0 \,, \tag{5.28}$$

which to the following ward identity:

$$- z\tau_\mu T^\mu + T^{\mu\nu} h_{\mu\nu} - \mathcal{E}_\Phi D_\Phi \Phi = 0 \,. \tag{5.29}$$

This can also be expressed as

$$- z\tau_\mu v^\nu T^\mu{}_\nu + h^{\nu\rho} h_{\rho\mu} T^\mu{}_\nu - \mathcal{E}_\Phi D_\Phi \Phi = 0 \,. \tag{5.30}$$

On shell, using the matter field equations of motion, this leads to the vanishing of the $z$-deformed trace of the energy-momentum tensor

$$- z\tau_\mu v^\nu T^\mu{}_\nu + h^{\nu\rho} h_{\rho\mu} T^\mu{}_\nu = 0 \,. \tag{5.31}$$

In order to compute the currents in (5.21) it is important that the field theory is defined on an arbitrary Aristotelian geometry. If we couple a field theory to a restricted class of geometries such as the torsion-free geometries discussed above then the restriction to vanishing intrinsic torsion (5.18) has implications for the field theoretic quantities that we are able to extract as responses to varying the background sources since imposing conditions on the background also constrains the allowed variations of the sources. In other words, when we impose that the background is such that the intrinsic torsion vanishes, the variations we are allowed to make must preserve this condition and so are no longer arbitrary (see [60] for a similar discussion in the context of Newton–Cartan geometry). For example, if $\tau_\mu$ is exact so that $\tau_\mu = \partial_\mu T$ for some scalar field $T$, then $\delta\tau_\mu$ must be exact as well, $\delta\tau_\mu = \partial_\mu \delta T$. This means that we only have access to the divergence of the energy current (which is proportional to the variation of $T$) when we assume the torsion to be zero.

Below we will put the complex scalar field theory on an arbitrary Aristotelian geometry with general intrinsic torsion. For the case of the scalar charge gauge theory we will for simplicity restrict ourselves to the case of vanishing intrinsic torsion.

# 6 Coupling the scalar fields to curved spacetime

We will illustrate the method of coupling the complex scalar theories of Section 2 to an arbitrary Aristotelian geometry for one specific model. The other theories can be coupled in a similar fashion.

The particular Lagrangian of a scalar field theory with global dipole symmetry that we will consider is

$$\mathcal{L} = \dot{\Phi}\dot{\Phi}^\star - m^2 \left|\Phi\right|^2 - \lambda(\partial_i\Phi\partial_j\Phi - \Phi\partial_i\partial_j\Phi)(\partial_i\Phi^\star\partial_j\Phi^\star - \Phi^\star\partial_i\partial_j\Phi^\star)\,, \tag{6.1}$$

where $\Phi$ is a complex scalar of mass $m$ and $\lambda$ is a coupling constant. This Lagrangian is invariant under the global transformation

$$\Phi \to e^{i(\alpha + \beta_i x^i)}\Phi\,, \tag{6.2}$$

where $\alpha$ is the parameter of a global $U(1)$ transformation, while $\beta_i$ is the parameter of the dipole transformation. Following the Noether procedure of Section 2.7 we can gauge this symmetry with the help of $A_{ij}$ and $\phi$. To this end we define

$$\hat{X}_{ij} = \partial_i\Phi\partial_j\Phi - \Phi\partial_i\partial_j\Phi + iA_{ij}\Phi^2\,, \tag{6.3}$$

which transforms as $\hat{X}_{ij} \to e^{2i\Lambda}\hat{X}_{ij}$ under the gauge transformation (4.3), in which case the gauge invariant Lagrangian reads

$$\mathcal{L} = (\partial_t\Phi - i\phi\Phi)(\partial_t\Phi^\star + i\phi\Phi^\star) - m^2 \left|\Phi\right|^2 - \lambda\hat{X}_{ij}\hat{X}_{ij}^\star\,. \tag{6.4}$$

The curved space generalisation of the Lagrangian above is

$$\mathcal{L}_{\text{scalar}} = e\left[(v^\nu\partial_\nu\Phi + i\phi\Phi)(v^\mu\partial_\mu\Phi^\star - i\phi\Phi^\star) - m^2 \left|\Phi\right|^2 - \lambda h^{\mu\nu}h^{\rho\sigma}\hat{X}_{\mu\rho}\hat{X}_{\nu\sigma}^\star\right]\,, \tag{6.5}$$

where

$$\hat{X}_{\mu\nu} = P_{(\mu}^\rho P_{\nu)}^\sigma (\partial_\rho\Phi\partial_\sigma\Phi - \Phi\nabla_\rho\partial_\sigma\Phi) + iA_{\mu\nu}\Phi^2\,, \tag{6.6}$$

in which $\nabla_\rho$ is covariant with respect to the Aristotelian connection (5.17) and where the spatial projector $P_\mu^\rho$ is defined by

$$P_\nu^\mu = h^{\mu\rho}h_{\rho\nu} = \delta_\nu^\mu + v^\mu\tau_\nu\,. \tag{6.7}$$

The symmetric gauge field $A_{\mu\nu}$ is defined to be purely spatial, i.e., we demand that

$$v^\mu A_{\mu\nu} = 0\,. \tag{6.8}$$

The Lagrangian (6.5) is gauge invariant under the curved space generalisation of the gauge transformations (2.63):

$$\delta\phi = -v^\mu\partial_\mu\Lambda\,, \qquad \delta A_{\mu\nu} = P_{(\mu}^\rho P_{\nu)}^\sigma \nabla_\rho\partial_\sigma\Lambda\,. \tag{6.9}$$

The transformation of the matter field $\Phi$ is unchanged, i.e., $\delta\Phi = i\Lambda\Phi$.

As we will see in the next section and as was discussed in [31] there are restrictions on the background geometry when coupling the scalar charge gauge theory to a curved Aristotelian geometry. However, we see here that there are no constraints on the kind of the Aristotelian backgrounds we can couple the scalar theory to. In other words, if we are happy to consider the gauge fields $\phi$ and $A_{\mu\nu}$ as background fields, we can put the fracton field theory on any Aristotelian background we like. As per the discussion in Section 5.3 we can obtain both the energy current and the momentum-stress tensor by varying the background geometry in (6.5) and similar Lagrangians for other complex scalar models.

For $m = 0$, we can generalise (6.5) to be invariant under the following (anisotropic) Weyl transformations

$$\delta\tau_\mu = z\Omega\tau_\mu \qquad\qquad \delta h_{\mu\nu} = 2\Omega h_{\mu\nu} \qquad\qquad \delta\Phi = -D_\Phi\Omega\Phi \qquad\qquad (6.10\text{a})$$

$$\delta\phi = -z\Omega\phi \qquad\qquad \delta A_{\mu\nu} = 0\,, \qquad\qquad (6.10\text{b})$$

where $z = (d+4)/3$ and $D_\Phi = -(d-2)/3$. Note that in $d = 2$ dimensions $z = d = 2$ and $D_\Phi = 0$. In this case the action whose Lagrangian is (6.5) is anisotropic Weyl invariant. For $d = 3$ we need to add curvature terms (non-minimal couplings) to make the theory anisotropic Weyl invariant. First of all we notice that for $m = 0$ we have

$$\delta_\Omega\mathcal{L}_{\text{scalar}} = -D_\Phi e v^\nu\partial_\nu\left(\Phi\Phi^\star\right)v^\mu\partial_\mu\Omega - D_\Phi\lambda h^{\mu\nu}h^{\rho\sigma}\left(\Phi^{\star 2}X_{\nu\sigma}\nabla_\mu\partial_\rho\Omega + \Phi^2 X^\star_{\mu\rho}\nabla_\nu\partial_\sigma\Omega\right)\,. \tag{6.11}$$

We have the following useful results

$$\delta_\Omega K_{\mu\nu} = (2-z)\Omega K_{\mu\nu} - h_{\mu\nu}v^\rho\partial_\rho\Omega \tag{6.12a}$$

$$\delta_\Omega\Gamma^\rho_{\mu\nu} = -zv^\rho\tau_\nu\partial_\mu\Omega - h^{\rho\sigma}h_{\mu\sigma}\tau_\nu v^\lambda\partial_\lambda\Omega$$
$$\qquad\qquad + h^{\rho\lambda}\left(h_{\lambda\nu}\partial_\mu\Omega + h_{\lambda\mu}\partial_\nu\Omega - h_{\mu\nu}\partial_\lambda\Omega\right) \tag{6.12b}$$

$$\delta_\Omega R_{\mu\sigma} = -\nabla_\mu\delta\Gamma^\rho_{\rho\sigma} + \nabla_\rho\delta\Gamma^\rho_{\mu\sigma} + 2\Gamma^\lambda_{[\rho\mu]}\delta\Gamma^\rho_{\lambda\sigma} \tag{6.12c}$$

$$h^{\mu\alpha}h^{\sigma\beta}\delta_\Omega R_{(\mu\sigma)} = -h^{\mu\alpha}h^{\sigma\beta}\left((d-2)\nabla_{(\mu}\partial_{\sigma)}\Omega + h_{\mu\sigma}h^{\rho\lambda}\nabla_\rho\partial_\lambda\Omega\right)\,. \tag{6.12d}$$

Hence for $d = 3$ we have

$$h^{\mu\alpha}h^{\sigma\beta}\delta_\Omega\left(R_{(\mu\sigma)} - \frac{1}{4}h_{\mu\sigma}h^{\alpha\beta}R_{\alpha\beta}\right) = -h^{\mu\alpha}h^{\sigma\beta}\nabla_{(\mu}\partial_{\sigma)}\Omega\,. \tag{6.13}$$

If we define $\tilde{X}_{\nu\sigma}$ for $d = 3$ (so that $D_\Phi = -1/3$) as

$$\tilde{X}_{\nu\sigma} = \hat{X}_{\nu\sigma} - \frac{1}{3}\Phi^2 P^\mu_\nu P^\rho_\sigma\left(R_{(\mu\rho)} - \frac{1}{4}h_{\mu\rho}h^{\alpha\beta}R_{\alpha\beta}\right) \tag{6.14}$$

then $\tilde{X}_{\nu\sigma}$ transforms homogeneously under $\Omega$ (i.e., without derivatives).

Using furthermore that

$$v^\mu \partial_\mu \Phi - \frac{D_\phi}{d} K \Phi \,, \tag{6.15}$$

where $K$ is the trace of $K_{\mu\nu}$, scales homogeneously under $\Omega$ we can write down the following anisotropic Weyl invariant theory in $d = 3$ dimensions

$$\mathcal{L}_{\text{scalar}} = e \left[ \left( v^\nu \partial_\nu \Phi + i\phi\Phi + \frac{1}{9} K\Phi \right) \left( v^\mu \partial_\mu \Phi^\star - i\phi\Phi^\star + \frac{1}{9} K\Phi^\star \right) - \lambda h^{\mu\nu} h^{\rho\sigma} \tilde{X}^\star_{\mu\rho} \tilde{X}_{\nu\sigma} \right] . \tag{6.16}$$

If we compute the energy-momentum tensor of this theory it will obey the $z$-deformed traceless condition (5.31).

Referring back to equation (2.15) and the discussion of improvements of the Noether energy-momentum tensor, it is this result, the Weyl invariant coupling to an arbitrary Aristotelian space, that guarantees the existence of $\Theta^\mu{}_\nu$, the improved energy-momentum tensor used in equation (2.15).

If we consider the complex scalar field theory on a fixed curved background we can ask if it still has a global dipole symmetry. This will be the case provided that we can set both the gauge fields and their transformations (6.9) to zero. In other words, the scalar field theory whose Lagrangian on a curved spacetime is given by (6.5) in which we set $\phi = 0 = A_{\mu\nu}$ admits a global symmetry of the form $\delta\Phi = i\Lambda\Phi$ provided $\Lambda$ obeys the conditions

$$v^\mu \partial_\mu \Lambda = 0 \,, \qquad P^\rho_{(\mu} P^\sigma_{\nu)} \nabla_\rho \partial_\sigma \Lambda = 0 \,. \tag{6.17}$$

On a generic background there need not exist any non-constant $\Lambda$ that obeys these equations.

# 7   Scalar charge gauge theories on Aristotelian geometry

We will now couple the scalar charge gauge theory to curved Aristotelian spacetime. Unlike for the case of the complex scalar fields, the coupling of the scalar charge gauge theory to curved spacetime is less straightforward. Perhaps the analogy with partially massless gravitons makes this somewhat less surprising. The coupling of the scalar charge gauge theory to curved space (but not spacetime) has previously been considered in [31]. To facilitate comparison, we begin by coupling the scalar charge gauge theory to a partially gauge fixed torsion-free Aristotelian background that admits a timelike foliation whose leaves are a priori arbitrary Riemannian geometries. We recover previous results that require the spatial geometry to obey certain conditions in order for the theory to maintain gauge invariance. We then generalise this coupling to Aristotelian space*time*. We summarise our results regarding the coupling to curved space in Table 1 and to curved spacetime in Table 2.

## 7.1 Coupling the scalar charge gauge theory to curved space

In order to get started we will first look at a special class of torsion-free Aristotelian geometries for which the Riemannian geometry on constant time slices is time-independent. We will partially gauge fix the $(d+1)$-dimensional diffeomorphism invariance so that $\tau = dt$ and $h_{\mu\nu}dx^\mu dx^\nu = h_{ij}dx^i dx^j$ with $\partial_t h_{ij} = 0$. In other words we consider the simpler problem of curving up the geometry on constant $t$ slices. This is a $d$-dimensional Riemannian geometry, and we denote by $D_i$ the Levi–Civita connection of this geometry.

### 7.1.1 The magnetic sector

In this subsection we only consider the magnetic part of the Lagrangian, i.e., the curved generalisation of

$$\mathcal{L}_{\text{mag}} = -\frac{h_1}{4}F_{ijk}F_{ijk} - \frac{h_2}{2}F_{ijj}F_{ikk}\,. \tag{7.1}$$

Let us define $F_{ijk}$ to be

$$F_{ijk} = D_i A_{jk} - D_j A_{ik}\,, \tag{7.2}$$

and the gauge transformation to be

$$\delta A_{ij} = D_i \partial_j \Lambda\,. \tag{7.3}$$

Note that the right-hand side is symmetric in $(ij)$ since we are using the Levi-Civita connection. The object $F_{ijk}$ is covariant under $d$-dimensional general coordinate transformations of the form $x^i \to x'^i = x'^i(x)$ but is no longer invariant under the $\Lambda$ gauge transformations. Instead we now find that

$$\delta_\Lambda F_{ijk} = D_i D_j D_k \Lambda - D_j D_i D_k \Lambda = R_{ijk}{}^l D_l \Lambda\,. \tag{7.4}$$

One way to possibly deal with this is to introduce a new gauge field $A_i$ that transforms as $\delta A_i = \partial_i \Lambda$ and then to define

$$\check{F}_{ijk} = D_i A_{jk} - D_j A_{ik} - R_{ijk}{}^l A_l = F_{ijk} - R_{ijk}{}^l A_l\,. \tag{7.5}$$

However, we will show in appendix C that this procedure leads to a Stückelberging of the dipole symmetry in that $A_{ij}$ now always appears in the combination $A_{ij} - D_{(i}A_{j)}$. We show that this remains true if we include a complex scalar field that is minimally coupled to $(\phi, A_i)$. The field $A_{ij} - D_{(i}A_{j)}$ is not a gauge field and so its presence does not correspond to any genuine gauge invariance. This procedure therefore does away with the need to introduce $A_{ij}$ in the first place and is thus unwanted.

Here we will show that for a specific relation between $h_1$ and $h_2$ the magnetic Lagrangian can be coupled to a curved geometry without invoking $A_i$ provided the geometry on the constant time slices is a space of constant sectional curvature, thus

reproducing a result found in [31]. The curved generalisation of the magnetic part of the Lagrangian is

$$\mathcal{L}_{\text{mag}} = -\sqrt{h}\left(\frac{h_1}{4}h^{jm}h^{kn} + \frac{h_2}{2}h^{jk}h^{mn}\right)h^{il}F_{ijk}F_{lmn}\,, \tag{7.6}$$

where $F_{ijk}$ is as in (7.2) and where $h = \det h_{ij}$. The gauge variation of the Lagrangian is

$$\delta_\Lambda \mathcal{L}_{\text{mag}} = -\sqrt{h}\left(\frac{h_1}{2}h^{jm}h^{kn} + h_2 h^{jk}h^{mn}\right)h^{il}R_{ijka}F_{lmn}\partial^a\Lambda\,. \tag{7.7}$$

For $d = 3$ the Riemann tensor can be written as

$$R_{ijkl} = h_{ik}R_{jl} - h_{jk}R_{il} + h_{jl}R_{ik} - h_{il}R_{jk} - \frac{R}{2}\left(h_{ik}h_{jl} - h_{jk}h_{il}\right)\,. \tag{7.8}$$

In this case the variation can be written as

$$\delta\mathcal{L}_{\text{mag}} = \sqrt{h}\left((h_1 + h_2)h^{mn}\left(R^l_{\ a} - \frac{R}{3}\delta^l_a\right)\right.$$
$$\left. + h_1\left(R^{mn} - \frac{R}{3}h^{mn}\right)\delta^l_a + \frac{h_1 + 2h_2}{6}Rh^{mn}\delta^l_a\right)F_{lmn}\partial^a\Lambda\,. \tag{7.9}$$

Hence, in order to have invariance in $d = 3$ we need to assume that the Ricci tensor is pure trace, i.e., $R_{ij} = \frac{R}{3}h_{ij}$ and furthermore we need to take $h_2 = -h_1/2$. This is precisely the value for which the magnetic part is independent of the trace of $A_{ij}$.

In fact we can generalise this result to general dimension $d$. If the Riemann tensor for a $d$-dimensional manifold is given by

$$R_{ijkl} = \frac{R}{d(d-1)}(h_{ik}h_{jl} - h_{il}h_{jk})\,, \tag{7.10}$$

then substitution into equation (7.7) shows that the Lagrangian is invariant if we take $h_1 = -(d-1)h_2$, which is the same condition we met in (4.66) in the traceless theory. Due to (4.54), this also implies that the $d = 2$ scalar charge gauge theory *cannot* be coupled to any curved background in a gauge-invariant way. We come back to the case $d = 2$ in Section 7.1.4. Given (7.10) it follows that the Einstein tensor is $G_{ij} = -\frac{d-2}{2d}Rh_{ij}$. Using the twice contracted Bianchi identity $D^iG_{ij} = 0$ this tells us that $R$ must be constant for $d \geq 3$. Therefore spaces of the form (7.10) are spaces of constant sectional curvature.

Consider again the case $d = 3$ with $h_2 = -h_1/2$. In this case the variation of the magnetic part of the Lagrangian is

$$\delta\mathcal{L}_{\text{mag}} = \sqrt{h}h_1\left(R^{mn} - \frac{R}{3}h^{mn}\right) \tag{7.11}$$
$$\times \left(\frac{1}{4}h^{ij}F_{mij}\partial_n\Lambda + \frac{1}{4}h^{ij}F_{nij}\partial_m\Lambda + h^{ij}F_{i(mn)}\partial_j\Lambda - \frac{1}{2}h_{mn}h^{ij}h^{kl}F_{ikl}\partial_j\Lambda\right)\,,$$

where the second parenthesis has been made symmetric and traceless in $m$ and $n$. We can make $\mathcal{L}_{\text{mag}}$ gauge invariant by adding a Lagrange multiplier term

$$\mathcal{L}_{\text{LM}} = \sqrt{h} h_1 \left( R^{mn} - \frac{R}{3} h^{mn} \right) \mathcal{X}_{mn} \,, \tag{7.12}$$

where $\mathcal{X}_{mn}$ is a traceless symmetric Lagrange multiplier that transforms as

$$\delta_\Lambda \mathcal{X}_{mn} = - \left( \frac{1}{4} h^{ij} F_{mij} \partial_n \Lambda + \frac{1}{4} h^{ij} F_{nij} \partial_m \Lambda + h^{ij} F_{i(mn)} \partial_j \Lambda - \frac{1}{2} h_{mn} h^{ij} h^{kl} F_{ikl} \partial_j \Lambda \right) \,. \tag{7.13}$$

In higher dimensions, the Riemann tensor is no longer determined only in terms of the Ricci tensor, and we generically expect that a similar procedure would involve a Lagrange multiplier with four indices, i.e., $\mathcal{X}_{ijkl}$.

### 7.1.2 The electric sector

We next consider the electric sector with $g_1 + dg_2 > 0$, i.e., the traceful electric theory. For the moment we still restrict ourselves to geometries of the form $\tau = dt$ and $h_{\mu\nu} dx^\mu dx^\nu = h_{ij} dx^i dx^j$ with $\partial_t h_{ij} = 0$. On such a geometry the Lagrangian of the electric sector of the scalar charge gauge theory is given by

$$\mathcal{L}_{\text{elec}} = \sqrt{h} \left( \frac{1}{2g_1} h^{ik} h^{jl} F_{0ij} F_{0kl} - \frac{g_2}{g_1(g_1 + dg_2)} (h^{ij} F_{0ij})^2 \right) \,, \tag{7.14}$$

where we defined

$$F_{0ij} = \dot{A}_{ij} - D_i \partial_j \phi \,, \tag{7.15}$$

which, unlike $F_{ijk}$, is invariant under the gauge transformations $\delta\phi = \partial_t \Lambda$ and $\delta A_{ij} = D_i \partial_j \Lambda$, i.e.,

$$\delta F_{0ij} = 0 \,. \tag{7.16}$$

It is thus straightforward to put the traceful electric theory on the curved space described by $\tau = dt$ and $h_{\mu\nu} dx^\mu dx^\nu = h_{ij} dx^i dx^j$. The traceless electric theory has $g_1 + dg_2 = 0$ and needs to be treated independently.

### 7.1.3 The traceless scalar charge gauge theory on curved space

To end our considerations of scalar charge gauge theories on curved space, let us explicitly demonstrate how the traceless scalar charge gauge theory (4.68) couples to curved space, thereby reproducing the results of [31] for $d = 3$. In this case both the electric and the magnetic sector are traceless. The gauge transformations of the gauge fields of the traceless theory on curved space are

$$\delta A_{ij} = D_i D_j \Lambda - \frac{1}{d} h_{ij} D^2 \Lambda \qquad \text{and} \qquad \delta\phi = \dot{\Lambda} \,, \tag{7.17}$$

where $D^2\Lambda = h^{ij}D_iD_j\Lambda$. The electric and magnetic field strengths are given by

$$\tilde{F}_{0ij} = \dot{A}_{ij} - D_iD_j\phi - \frac{1}{d}(h^{kl}\dot{A}_{kl} - D^2\phi)h_{ij} \tag{7.18}$$

$$F_{ijk} = 2D_{[i}A_{j]k}. \tag{7.19}$$

The electric field strength is invariant under (7.17), while the magnetic field strength transforms as

$$\delta F_{ijk} = R_{ijk}{}^l D_l\Lambda + \frac{2}{d}h_{k[i}\partial_{j]}(D^2\Lambda), \tag{7.20}$$

which is the generalisation of (4.65). The curved space traceless theory is given by

$$\mathcal{L}[A_{ij}, \phi] = \sqrt{h}\left[\frac{1}{2g_1}h^{ik}h^{jl}\tilde{F}_{0ij}\tilde{F}_{0kl} - \frac{h_1}{4}h^{il}\left(h^{jm}h^{kn} - \frac{2}{d-1}h^{jk}h^{mn}\right)F_{ijk}F_{lmn}\right]. \tag{7.21}$$

It is easy to verify that this theory is gauge invariant on backgrounds that satisfy the relation (7.10). We also see that the traceless electric theory, which is given by the above Lagrangian with $h_1 = 0$, i.e.

$$\mathcal{L}_{\text{traceless electric}}[A_{ij}, \phi] = \sqrt{h}\frac{1}{2g_1}h^{ik}h^{jl}\tilde{F}_{0ij}\tilde{F}_{0kl}, \tag{7.22}$$

can be coupled to any curved space.

### 7.1.4   2 + 1 dimensions

In Section 4.6 we mentioned that for $d = 2$ we can consider the CS-like theory given in (4.53). If we couple this to curved space we find

$$\mathcal{L} = \frac{k}{4\pi}\epsilon^{ij}h^{kl}\left(A_{ik}\dot{A}_{jk} + \phi D_kF_{ijk}\right), \tag{7.23}$$

where $F_{ijk}$ is now given by (7.2) and where $\epsilon^{ij}$ is the Levi-Civita symbol. Under the gauge transformations $\delta\phi = \partial_t\Lambda$ and $\delta A_{ij} = D_i\partial_j\Lambda$ we find that

$$\delta\mathcal{L} = \frac{k}{4\pi}\Lambda\epsilon^{ij}\partial_i\phi\partial_j R, \tag{7.24}$$

where we used that any 2-dimensional Riemann tensor is of the form $R_{ijkl} = \frac{R}{2}(h_{ik}h_{jl} - h_{il}h_{jk})$ with an arbitrary Ricci scalar $R$. The Lagrangian is (7.23) is gauge invariant provided the spatial geometry has constant curvature $R$ [31, 32].

### 7.1.5 Summary of coupling to curved space

We have summarised the coupling of the scalar charge gauge theory to $(d + 1)$-dimensional Aristotelian geometry with absolute time and time-independent Riemann geometries on the leaves of the foliation in Table 1 below.

The scalar charge gauge theory with a traceful electric sector and a traceless magnetic sector coupled to curved space for $d \geq 3$ is described by the Lagrangian

$$\mathcal{L} = \sqrt{h} \left[ \frac{1}{2g_1} h^{ik} h^{jl} F_{0ij} F_{0kl} - \frac{g_2}{g_1(g_1 + dg_2)} (h^{ij} F_{0ij})^2 \right. $$
$$\left. + h_1 \left( -\frac{1}{4} h^{jm} h^{kn} + \frac{1}{2(d-1)} h^{jk} h^{mn} \right) h^{il} F_{ijk} F_{lmn} \right] . \tag{7.25}$$

The magnetic and electric field strengths, respectively, are defined in (7.2) and (7.15), while the background geometry is subject to the condition (7.10). In particular, the theory (7.25) is not traceless in the sense of (4.56) since the electric part depends on the trace of $A_{ij}$. For $d = 3$, we introduced a Lagrange multiplier $\mathcal{X}_{mn}$ that restricts the background geometry, which led to the following Lagrangian

$$\mathcal{L} = \sqrt{h} \left[ \frac{1}{2g_1} h^{ik} h^{jl} F_{0ij} F_{0kl} - \frac{g_2}{g_1(g_1 + 3g_2)} (h^{ij} F_{0ij})^2 \right. $$
$$\left. - \frac{h_1}{4} \left( h^{jm} h^{kn} - h^{jk} h^{mn} \right) h^{il} F_{ijk} F_{lmn} + h_1 \left( R^{mn} - \frac{R}{3} h^{mn} \right) \mathcal{X}_{mn} \right] . \tag{7.26}$$

Similar Lagrange multiplier terms can be constructed in higher dimensions.

We have summarised the coupling to curved space in the Table 1 below.

| Dim. | Theory | Spatial geometry |
|---|---|---|
| $d = 2$ | magnetic theory with $h_1 + h_2 > 0$ | flat |
| | electric theory (traceful and traceless) | any |
| | CS-like theory | constant sectional curvature |
| $d \geq 3$ | magnetic theory with $h_2 \neq -(d-1)h_1$ | flat |
| | magnetic theory with $h_2 = -(d-1)h_1$ | constant sectional curvature |
| | electric theory (traceful and traceless) | any |

**Table 1**: Summary of the spatial backgrounds to which the scalar charge gauge theories can couple in a gauge-invariant way. The electric theory is given in (7.14) if it is traceful and in (7.22) if it is traceless. The magnetic theory is given in (7.6). Recall that the scalar charge gauge theory, for which $h_2 = -(d-1)h_1$, only has a non-trivial magnetic sector for $d \geq 3$ (c.f., (4.54)). The condition for constant sectional curvature is given in (7.10).

Note that Table 1 agrees with Table 1 of [31] with the understanding that 3-dimensional Einstein spaces must have a constant Ricci scalar (as follows from the covariant constancy of the Einstein tensor) and are therefore spaces of constant sectional curvature (which in turn follows from the fact that the Weyl tensor vanishes identically in 3 dimensions). For the higher dimensional cases we also find that the coupling of traceful theories is restricted to flat backgrounds (due to the magnetic sector), but when the magnetic theory is traceless the theories can be coupled to backgrounds of constant sectional curvature (this generalises the result of [31] since the electric sector can be traceful).

## 7.2 Coupling the scalar charge gauge theory to curved spacetime

In this section, we couple the scalar charge gauge theory to any torsion-free Aristotelian spacetime. This means that we generalise the previous results by allowing for time-dependent $h_{ij}$ and that we furthermore add a shift vector $N^i$ as in (5.19). We will however refrain from using the ADM parametrisation here and instead use a spacetime covariant notation.

We generalise the symmetric tensor gauge field $A_{ij}$ by replacing $A_{ij} \rightarrow A_{\mu\nu}$ satisfying

$$v^\mu A_{\mu\nu} = 0 \qquad \text{and} \qquad A_{[\mu\nu]} = 0 \,. \tag{7.27}$$

The absence of torsion implies that the gauge transformation of the symmetric tensor gauge field can be written as[13]

$$\delta A_{\mu\nu} = P_\mu^\rho P_\nu^\sigma \nabla_\rho \partial_\sigma \Lambda \,, \tag{7.28}$$

which preserves (7.27), while the scalar $\phi$ transforms as

$$\delta\phi = -v^\mu \partial_\mu \Lambda \,. \tag{7.29}$$

We replace the field strengths $F_{0ij}$ and $F_{ijk}$ by the following quantity

$$F_{\mu\nu\rho} = \nabla_\mu A_{\nu\rho} - \nabla_\nu A_{\mu\rho} - 2P_\rho^\sigma \tau_{[\mu} \nabla_{\nu]} \nabla_\sigma \phi \,. \tag{7.30}$$

By construction, this field strength satisfies

$$v^\rho F_{\mu\nu\rho} = 0 \,, \tag{7.31}$$

and transforms under (7.28) as

$$\delta F_{\mu\nu\rho} = R_{\mu\nu\rho}{}^\sigma \partial_\sigma \Lambda \,, \tag{7.32}$$

---

[13]If we drop the assumption of a torsion-free background, we must explicitly symmetrise the projectors since $\nabla_\rho \partial_\sigma \Lambda$ is no longer symmetric, i.e.,

$$\delta A_{\mu\nu} = P_{(\mu}^\rho P_{\nu)}^\sigma \nabla_\rho \partial_\sigma \Lambda \,.$$

where the Riemann tensor of the Aristotelian connection (5.17) is given by (1.4). Furthermore, by explicitly writing out the definition of the field strength, we see that

$$3F_{[\mu\nu\rho]} = F_{\mu\nu\rho} + F_{\rho\mu\nu} + F_{\nu\rho\mu} = 0\,, \tag{7.33}$$

which, together with the fact that $F_{\mu\nu\rho}$ is antisymmetric in its first two indices, implies that the field strength is hook symmetric.

The electric part of the field strength is symmetric in its two indices

$$-v^\mu h^{\nu\kappa} h^{\rho\lambda} F_{\mu\nu\rho} = -h^{\nu\kappa} h^{\rho\lambda} \left(v^\mu \nabla_\mu A_{\nu\rho} + \nabla_\nu \partial_\rho \phi\right)\,, \tag{7.34}$$

which transforms as

$$\delta\left(-v^\mu h^{\nu\kappa} h^{\rho\lambda} F_{\mu\nu\rho}\right) = -v^\mu h^{\nu\kappa} h^{\rho\lambda} R_{\mu\nu\rho}{}^\sigma \partial_\sigma \Lambda\,. \tag{7.35}$$

The magnetic part of the field strength is

$$h^{\mu\sigma} h^{\nu\kappa} h^{\rho\lambda} F_{\mu\nu\rho} = h^{\mu\sigma} h^{\nu\kappa} h^{\rho\lambda} \left(\nabla_\mu A_{\nu\rho} - \nabla_\nu A_{\mu\rho}\right)\,, \tag{7.36}$$

which transforms as

$$\delta\left(h^{\mu\sigma} h^{\nu\kappa} h^{\rho\lambda} F_{\mu\nu\rho}\right) = h^{\mu\sigma} h^{\nu\kappa} h^{\rho\lambda} R_{\mu\nu\rho}{}^\alpha \partial_\alpha \Lambda\,. \tag{7.37}$$

### 7.2.1  The magnetic sector

The Lagrangian with the condition (4.66) (for $d \geq 3$) already implemented is

$$\mathcal{L}_{\mathrm{mag}} = eh_1\left(-\frac{1}{4} h^{\nu\lambda} h^{\rho\kappa} + \frac{1}{2(d-1)} h^{\nu\rho} h^{\lambda\kappa}\right) h^{\mu\sigma} F_{\mu\nu\rho} F_{\sigma\lambda\kappa}\,, \tag{7.38}$$

and the variation is now

$$\delta\mathcal{L}_{\mathrm{mag}} = eh_1\left(-\frac{1}{2} h^{\nu\lambda} h^{\rho\kappa} + \frac{1}{d-1} h^{\nu\rho} h^{\lambda\kappa}\right) h^{\mu\sigma} R_{\mu\nu\rho}{}^\alpha F_{\sigma\lambda\kappa} \partial_\alpha \Lambda\,. \tag{7.39}$$

Since the Aristotelian geometry is taken to be torsion-free we have the usual algebraic Bianchi identity $R_{[\mu\nu\rho]}{}^\alpha = 0$ which implies that the Ricci tensor $R_{\mu\nu} = R_{\mu\rho\nu}{}^\rho$ is symmetric. Using that $\nabla_\mu v^\sigma = 0$ and thus that $0 = [\nabla_\mu, \nabla_\nu]v^\sigma = -R_{\mu\nu\rho}{}^\sigma v^\rho$, it follows that the Ricci tensor is spatial, i.e., $v^\mu R_{\mu\nu} = 0$. Hence the only contraction of $R_{\mu\nu}$ is what we will call the Ricci scalar $R = h^{\mu\nu} R_{\mu\nu}$. The identity $0 = [\nabla_\mu, \nabla_\nu]\tau_\rho = R_{\mu\nu\rho}{}^\sigma \tau_\sigma$ implies that $R_{\mu\nu} = R_{\mu\rho\nu}{}^\sigma P_\sigma^\rho$. The condition that makes the magnetic Lagrangian gauge invariant is

$$h^{\mu\kappa} h^{\nu\lambda} R_{\mu\nu\rho}{}^\sigma = \frac{R}{d(d-1)}\left(P_\rho^\kappa h^{\sigma\lambda} - h^{\sigma\kappa} P_\rho^\lambda\right)\,, \tag{7.40}$$

for any torsion-free Aristotelian spacetime.

### 7.2.2 The electric sector

The Lagrangian of the traceful electric sector is now

$$\mathcal{L}_{\text{elec}} = e \left( \frac{1}{2g_1} h^{\rho\lambda} h^{\sigma\kappa} - \frac{g_2}{g_1(g_1 + dg_2)} h^{\rho\sigma} h^{\lambda\kappa} \right) v^\mu v^\nu F_{\mu\rho\sigma} F_{\nu\lambda\kappa} , \qquad (7.41)$$

and its variation is

$$\delta\mathcal{L}_{\text{elec}} = e \left( \frac{1}{g_1} h^{\rho\lambda} h^{\sigma\kappa} - \frac{2g_2}{g_1(g_1 + dg_2)} h^{\rho\sigma} h^{\lambda\kappa} \right) v^\mu F_{\mu\rho\sigma} v^\nu R_{\nu\lambda\kappa}{}^\alpha \partial_\alpha \Lambda . \qquad (7.42)$$

Equation (7.34) tells us that $v^\mu h^{\rho\kappa} h^{\sigma\lambda} F_{\mu\rho\sigma}$ is symmetric in $\kappa$ and $\lambda$. Furthermore, the algebraic Bianchi identity $R_{[\mu\nu\rho]}{}^\sigma = 0$ and the fact that $R_{\mu\nu\rho}{}^\sigma v^\rho = 0$ tell us that $v^\nu R_{\nu\lambda\kappa}{}^\alpha$ is also symmetric in $\kappa$ and $\lambda$. Hence the electric Lagrangian is gauge invariant provided we demand that

$$v^\nu R_{\nu\lambda\kappa}{}^\alpha = 0 , \qquad (7.43)$$

for any torsion-free Aristotelian spacetime. We do not need to contract the $\kappa$ and $\lambda$ indices with $h^{\sigma\kappa} h^{\rho\lambda}$ because any contraction of $v^\nu R_{\nu\lambda\kappa}{}^\alpha$ with $v^\mu$ is zero. Therefore the condition (7.43) tells us that the Riemann tensor is entirely spatial, i.e., all possible contractions with $v^\mu$ and $\tau_\mu$ now vanish.

### 7.2.3 Summary of coupling to curved spacetime

The combined Lagrangian that describes the full theory with a traceful electric sector and a traceless magnetic sector is given by

$$\begin{aligned}
\mathcal{L} &= \mathcal{L}_{\text{elec}} + \mathcal{L}_{\text{mag}} \\
&= e \bigg[ \left( \frac{1}{2g_1} h^{\rho\lambda} h^{\sigma\kappa} - \frac{g_2}{g_1(g_1 + dg_2)} h^{\rho\sigma} h^{\lambda\kappa} \right) v^\mu v^\nu F_{\mu\rho\sigma} F_{\nu\lambda\kappa} \\
&\quad + h_1 \left( -\frac{1}{4} h^{\nu\lambda} h^{\rho\kappa} + \frac{1}{2(d-1)} h^{\nu\rho} h^{\lambda\kappa} \right) h^{\mu\sigma} F_{\mu\nu\rho} F_{\sigma\lambda\kappa} \bigg] ,
\end{aligned} \qquad (7.44)$$

where $d \geq 3$. The background must now satisfy the conditions (7.40) and (7.43), which can be summarised into one condition as

$$R_{\mu\nu\rho}{}^\sigma = \frac{R}{d(d-1)} \left( h_{\mu\rho} P_\nu^\sigma - h_{\nu\rho} P_\mu^\sigma \right) . \qquad (7.45)$$

This condition generalises (7.10) to the case of an arbitrary torsion-free Aristotelian background and can be imposed using an appropriate Lagrange multiplier as in (7.12). We summarise the state of play in the table below.

| Theory | Curved torsion-free Aristotelian background |
|---|---|
| Magnetic theory (7.38) with $h_2 = -(d-1)h_1$ $(d \geq 3)$ | obeying (7.40) |
| Magnetic theory (7.38) with $h_2 \neq -(d-1)h_1$ $(d \geq 2)$ | flat |
| Traceful electric theory for $d \geq 2$ (7.41) | obeying (7.43) |

**Table 2**: Summary of the torsion-free Aristotelian spacetimes to which the scalar charge theories can couple in a gauge-invariant way.

It would be interesting to generalise the traceless electric theory (7.22) and the Chern–Simons like theory of Section 7.1.4 to curved spacetime and furthermore to drop the assumption that the background geometry is torsion-free.

# 8 Discussion and outlook

In this paper we have shown how to couple fractonic theories to curved spacetime. This spacetime is not the familiar one of Lorentzian geometry, rather, it is an Aristotelian geometry described not by a metric but in terms of an Aristotelian structure consisting of $\tau_\mu$ and $h_{\mu\nu}$ as discussed in Section 5. We have shown how to couple the complex scalar theory with dipole symmetry to an arbitrary curved background, which, to the best of our knowledge, has been an open problem in the theoretical description of fractons. Additionally, we presented how the scalar charge gauge theory couples to torsion-free Aristotelian space*times*, generalising previous results in the literature where the coupling to curved space was considered [31]. In order to couple both the electric and magnetic sector we find that there are two cases. If the magnetic sector depends on the trace of $A_{\mu\nu}$ then the background must be flat. If the magnetic sector is traceless, i.e., obeys the condition (4.66), then the background must satisfy a severe restriction on its curvature, namely (7.45), which can be enforced using a Lagrange multiplier. We have shown that the electric sector of the scalar charge gauge theory, regardless of whether it is traceful or traceless, can be coupled to any torsion-free Aristotelian background.

Along the way, we have derived new results for the complex scalar theory with dipole symmetry that describes fracton matter. In particular, we have found a no-go theorem that tells us that such a theory cannot simultaneously enjoy linearly realised dipole symmetry, contain spatial derivatives, and be Gaussian. The case with linearly realised dipole symmetry that is also Gaussian thus contains no spatial derivatives, and we have shown that this an example of a Carrollian theory. Conversely, if the theory is Gaussian and has spatial derivatives, the dipole symmetry is non-linearly realised and the theory becomes a special case of a Lifshitz theory with polynomial

shift symmetry. We have gauged the dipole symmetry using the Noether procedure, and the dynamics of the resulting symmetric tensor gauge field is described by scalar charge gauge theory, for which we have provided a Faddeev–Jackiw type Hamiltonian analysis and elucidated the gauge structure using generalised differentials.

This opens up a number of interesting avenues for further research, some of which we list below.

**Vector charge gauge theory** There exist other interesting rank 2 symmetric gauge theories, one of them is the "vector charge theory" [21]. It is governed by gauge transformations of the form $A_{ij} \to A_{ij} + 2\partial_{(i}\Lambda_{j)}$ with the corresponding vector-flavoured generalised Gauss law $\partial_i E^{ij} = \rho^j$ and leads to mobility restrictions of a another kind and gives rise to one-dimensional particles that move on a line (lineons). Many of the results and tools of this work should generalise to this case.

**Fracton hydrodynamics** The theory of fracton hydrodynamics has been considered in, e.g., [61–65]. As we have shown in this work, fractons couple to Aristotelian geometry. In [34], the theory of boost-agnostic fluids was coupled to Aristotelian geometry, and it would be very interesting to include dipole symmetry in the approach of [34] and thus develop a theory of fracton hydrodynamics on curved space. As demonstrated in that paper, this would allow us to use the technology of hydrostatic partition functions to extract hydrodynamic information.

**Carroll theories** As mentioned in Section 2.4 the free (or Gaussian) uncoupled matter theory has Carrollian symmetry and the quanta are rather unconventional Carrollian particles. Even though the non-Gaussian and Aristotelian fractonic theories do not inherit these enhanced symmetries it is tempting to ask if we can gain further insights into the physics of fractons by perturbing around the Carrollian theories (see also [37, Appendix A]).[14]

**Charge–Dipole symmetries and their spacetimes** Much of the fascinating fractonic physics emerged by generalising beyond the usual symmetries (see, e.g., [13]). For the prototypical charge and dipole symmetries, i.e., the first two lines in (2.59), this could mean to classify all Lie algebras and their spacetimes with $\mathfrak{so}(d)$ rotations, two vectors, and two scalars. A classification in this direction which also uncovers further coincidental isomorphisms will be given in a future work [75].

**Gauge structure and asymptotic symmetries** Since the scalar charge theory is a gauge theory the question of conserved charges is intimately related to

---

[14]Curiously this resonates with early ideas of perturbations around Carrollian ("zero signature") geometries [66–68]. See also, e.g., [69–74] for more recent interesting works on Carrollian geometry.

asymptotic charges and symmetries. Indeed, in Section 4.2 a first step in this direction is taken when we recover the charge and dipole charge using a Regge–Teitelboim type [45, 46] analysis. A more elaborate analysis of the asymptotic symmetries might be interesting, especially since boosts, that often complicate the analysis for the Lorentzian theories, are absent. For this reason we find it reasonable to expect an enhanced asymptotic symmetry algebra.

**Higher spins** As emphasised repeatedly, see, e.g., [5, 21, 43] a generalisation to higher spins might be an interesting endeavor. Especially since many of the no-go results, as nicely summarised in [76], fail due to the non-Lorentzian symmetries and, what is possibly even more relevant, the absence of asymptotic momentum eigenstates for isolated particles (which helps to circumvent, e.g., the Weinberg–Witten theorem [77]). Higher spin symmetries are also closely tied to gauge symmetries and our elaborations in Section 4.2 uncover the interesting place at which the gauge structure of this fractonic theory sits, see (4.27). This might present a starting point for generalisations to higher rank and dual representations (see for instance [44]).

**Partially massless fractons** In Section 4.9 we have highlighted similarities between the scalar charge theory and partially massless gravitons. It might be interesting to understand if there is more to it and if one can formulate a non-linear theory of "partially massless fractons" (of possibly even higher spin [78–83]).

## Acknowledgments

We want to thank José Figueroa-O'Farrill, Kristan Jensen, Victor Lekeu, Jakob Salzer, Kevin Slagle, and Watse Sybesma for useful discussions.

In particular we want to thank Victor Lekeu for useful discussions concerning the gauge sector of the scalar charge theory and in particular for clearing up a confusion concerning the Bianchi identities and suggesting that the gauge structure is reminiscent of partially massless fields.

The work of LB is supported by the Royal Society Research Fellows Enhancement Award 2017 "Non-Relativistic Holographic Dualities" (grant number RGF\EA\180149). JH is supported by the Royal Society University Research Fellowship "Non-Lorentzian Geometry in Holography" (grant number UF160197). The work of EH is supported by the Royal Society Research Grant for Research Fellows 2017 "A Universal Theory for Fluid Dynamics" (grant number RGF\R1\180017). JM is supported by an EPSRC studentship. SP is supported by the Leverhulme Trust Research Project Grant (RPG-2019-218) "What is Non-Relativistic Quantum Gravity and is it Holographic?". SP acknowledges support of the Erwin Schrödinger

Institute (ESI) in Vienna where part of this work was conducted during the thematic programme "Geometry for Higher Spin Gravity: Conformal Structures, PDEs, and Q-manifolds".

# A    Electrodynamics

The purpose of this of appendix is mainly pedagogical. We will illustrate some of the concepts and ideas of the main text in the simpler and more familiar setting of electrodynamics.

## A.1    The gauge sector

We start by defining the gauge potential $A_i \sim \boxed{i}$ and its canonical conjugate $\pi^i$ with the equal-time Poisson bracket

$$\{A_i(x), \pi^j(y)\} = \delta_i^j \delta(x - y) \,. \tag{A.1}$$

The indices $i, j$ run from 1 to $d$, the spatial dimensions.

The gauge transformations are given by

$$\delta_\Lambda A_i = \partial_i \Lambda \qquad\qquad \delta_\Lambda \pi^i = 0 \tag{A.2}$$

where $\Lambda = \Lambda(t, x) \sim \bullet$. They are generated by the gauge generator

$$\tilde{G}[\Lambda] = \int \mathrm{d}^d x \left[ -\Lambda \, \partial_i \pi^i + \partial_i(\Lambda \pi) \right] . \tag{A.3}$$

Using the differential we can write pure gauge potentials as

$$(d\Lambda)_i = \partial_i \Lambda \,. \tag{A.4}$$

Since the discussion of the gauge sector follows mutatis mutandis from Section 4.2 we will be brief. Let us emphasise that this discussion is restricted to spatial slices, but generalises to spacetimes for Poincaré invariant electrodynamics. We define the gauge invariant "curvature" or "magnetic field"

$$F_{ij} = (dA)_{ij} = 2\partial_{[i} A_{j]} \sim \boxed{\begin{smallmatrix} i \\ j \end{smallmatrix}} \tag{A.5}$$

where the Young tableaux represents the antisymmetry of the indices. The curvature vanishes when the potential is pure gauge

$$F_{ij} = 2\partial_{[i}\partial_{j]}\Lambda = 0 \qquad (\Leftrightarrow \; d^2\Lambda = 0) \,. \tag{A.6}$$

Conversely, a vanishing curvature implies that the potential is pure gauge, i.e., $F = dA = 0 \implies A = d\Lambda$, this shows that the curvatures fully capture the gauge symmetries. The derivative of the antisymmetric tensors is given by

$$(dT)_{ijk} = \partial_{[i} T_{jk]} \,. \tag{A.7}$$

The final class of tensors we want to introduce are totally antisymmetric tensors $T_{[ijk]} = T_{ijk}$ which have the following Young tableaux

$$T_{ijk} \sim \begin{array}{|c|} \hline i \\ \hline j \\ \hline k \\ \hline \end{array}. \tag{A.8}$$

The differential Bianchi identity follows

$$\partial_{[i} F_{jk]} = 2\partial_{[i}\partial_j A_{k]} = 0 \qquad (\Leftrightarrow dF = d^2 A = 0) \tag{A.9}$$

and conversely $\partial_{[i} T_{jk]} = 0$ implies that $T_{jkl}$ is the curvature of a potential. Explicitly

$$\partial_{[i} T_{jk]} = 0 \quad \Longrightarrow \quad T_{ijk} = 2\partial_{[i} A_{j]} \qquad (\Leftrightarrow dT = 0 \Longrightarrow T = F = dA). \tag{A.10}$$

In summary, we have shown that there exists an exact sequence that we can schematically depict as

$$\bullet \xrightarrow{\ d\ } \square \xrightarrow{\ d\ } \begin{array}{|c|} \hline \\ \hline \\ \hline \end{array} \xrightarrow{\ d\ } \begin{array}{|c|} \hline \\ \hline \\ \hline \\ \hline \end{array}. \tag{A.11}$$

For the Hamiltonian density we demand rotational invariance and gauge invariant (basically so that the constraints fulfill a first-class system). To lowest order in derivatives of the gauge invariant quantities this leads to

$$\mathcal{H} = \frac{g}{2}\pi^i \pi_i + \frac{h}{4} F^{ij} F_{ij}. \tag{A.12}$$

In principle the parameters $g$ and $h$ are free and could in principle be set to zero, in contradistinction to the Poincaré invariant theory. We can then write the Hamiltonian action

$$\mathcal{L}[A_i, \pi^i, \phi] = \pi^i \dot{A}_i - \mathcal{H} + \phi \partial_i \pi^i - \partial_i(\phi \pi^i) \tag{A.13a}$$
$$= \pi^i(\dot{A}_i - \partial_i \phi) - \mathcal{H} \tag{A.13b}$$

where we have introduced the Lagrange multiplier $\phi$ which enforces the constraint. This theory has $d - 1$ degrees of freedom in $d$ spatial dimensions. The variation is given by

$$\delta\mathcal{L} = \left(-\dot{\pi}^i + h\partial_k F^{ki} + J^i\right)\delta A_i + \left(\dot{A}_i - g\pi_i - \partial_i\phi\right)\delta\pi^i$$
$$+ \left(\partial_i \pi^i + J^0\right)\delta\phi + \partial_0\theta^0 + \partial_i\theta^i \tag{A.14}$$

where

$$\theta^0 = \pi^i \delta A_i \qquad\qquad \theta^i = \pi^i\delta\phi - hF^{ij}\delta A_j \tag{A.15}$$

We have to add

$$\delta_\Lambda \phi = \partial_0 \Lambda \tag{A.16}$$

to the gauge symmetries (A.2) to show that the action is gauge invariant

$$\delta_\Lambda \mathcal{L} = 0 \,. \tag{A.17}$$

Solving the equations of motion for $\pi$ and substituting it into the action leads to the "covariant form" of the action

$$\mathcal{L}[A_i, \phi] = \frac{1}{2g}(\dot{A}_i - \partial_i \phi)(\dot{A}^i - \partial^i \phi) - \frac{h}{4}F^{ij}F_{ij} \,. \tag{A.18}$$

The fact that we do not write the first term in the usual Poincaré covariant form $-\frac{1}{2g}F^{0i}F_{0i}$ is a manifestation of the Aristotelian structure, where we have no natural nondegenerate Lorentzian metric that would allow for these index manipulations.

Another perspective is to consider emergent low energy $U(1)$ gauge theories. In general they are determined by Aristotelian geometry however there might be emergent Lorentz invariance and a "speed of light" determined by some microscopic Hamiltonian. In that case, like for, e.g., some $U(1)$ spin liquids, there would then be a natural nondegenerate Lorentzian metric.

## A.2   $3+1$ dimensions

In three spatial dimensions we can use the epsilon tensor to define the magnetic field as $B^i = \epsilon^{ijk}\partial_j A_k = \frac{1}{2}\epsilon^{ijk}F_{jk}$, which we can use to write the generic term Hamiltonian (A.12) as $\mathcal{H} = \frac{1}{2}(g\pi^i\pi_i + hB^iB_i)$. However in $3+1$ dimensions there is the option to add another term to the Hamiltonian

$$\mathcal{H}^\theta = \theta\,\pi^i B_i \,. \tag{A.19}$$

We will now show that this term is closely related to the usual $\theta$ term of the Witten effect. We start by adding the term to the generic Hamiltonian and complete the square

$$\mathcal{H}^{d=3} = \mathcal{H} + \mathcal{H}^\theta = \frac{1}{2}\left[\left(\sqrt{g}\pi^i + \frac{\theta}{\sqrt{g}}B^i\right)^2 + \left(h - \frac{\theta^2}{g}\right)B^iB_i\right] \tag{A.20}$$

Next we change coordinates according to

$$Q_i = A_i \qquad\qquad P^i = \pi^i + \frac{\theta}{g}B^i[A] \tag{A.21}$$

where the square brackets indicate that this is the magnetic tensor of $A_i$. This is a canonical transformation as can be seen from

$$\pi^i(\dot{A}_i - \partial_i\phi) - \mathcal{H}^{D=3} = P^i(\dot{Q}_i - \partial_i\phi) - \mathcal{K} - \frac{\theta}{2g}(\partial_0 F^0 + \partial_i F^i) \tag{A.22}$$

where the new Hamiltonian is given by

$$\mathcal{K} = \frac{1}{2}\left[ gP^i P_i + \left(h - \frac{\theta^2}{g}\right)B^i[Q]B_i[Q]\right] \tag{A.23}$$

and the boundary term or "generating function" $F$ is

$$F^0 = Q_i B^i \qquad\qquad F^i = \epsilon^{ijk}Q_j \dot{Q}_k - 2\phi B^i\,. \tag{A.24}$$

This means that the addition of the $\theta$ term to the Hamiltonian leads, after a canonical transformation, to an shift in the coupling constants of the $B^2$ term plus a boundary term. This means the equations of motion stay unaltered (up to the shift). However, the addition of the boundary term has nontrivial effects for the charges and quantum mechanics [84].[15]

For $2+1$ dimensions there is the possibility to add an $\epsilon_{ij}F^{ij}$ term to the Hamiltonian, which is a boundary term that leaves the EOM unaffected.

## B Field redefinitions for cases 2 and 3

In this appendix, we show that the two remaining cases of Section 4.7 do not lead to new theories.

**Case 2:** When $c_1 \neq -1/d$ and $c_2 = -1/d$, the gauge transformation again takes the form (4.64), which implies that we cannot construct a gauge invariant field strength by augmenting $F_{ijk}$ by adding a suitable term involving $A_{ii}$. With the Hamiltonian (4.30), gauge invariance again imposes the condition (4.66). The gauge invariant electric field strength is the same as in case 1 and thus given by (4.67). The Lagrangian in this case, obtained by integrating out $E_{ij}$ from (4.57), therefore takes the form

$$\mathcal{L}_2[A_{ij},\phi] = \frac{1}{2g_1}\tilde{F}_{0ij}\tilde{F}_{0ij} + \frac{(dc_1+1)^2}{2d(g_1+dg_2)}(\dot{A}_{ii})^2 - \frac{h_1}{4}F_{ijk}F_{ijk} + \frac{h_1}{2(d-1)}F_{ijj}F_{ikk}\,, \tag{B.1}$$

and is *not* independent of the trace $A_{ii}$ due to the second term in the above expression for $\mathcal{L}_2$, in contrast to the traceless theory we considered above in case 1. In deriving this result, we have assumed that $g_1 + dg_2 \neq 0$. The trace $A_{ii}$ is gauge invariant and hence it is like adding a scalar field to the theory of case 1. We thus conclude that this case does not lead to an interesting deformation of the scalar gauge theory and we will not consider it any further.

---

[15]To make this more obvious we can write $\partial_0 F^0 + \partial_i F^i = 2B^i(\dot{Q}_i - \partial_i \phi) = 1/4\epsilon^{\mu\nu\rho\sigma}F_{\mu\nu}F_{\rho\sigma}$ where the last equality sign uses a lorentzian metric.

**Case 3:** When $c_1, c_2 \neq -1/d$ the trace $A_{ii}$ is no longer gauge invariant, which can be used to construct a gauge invariant magnetic field strength invariant

$$\hat{F}_{ijk} = 2\partial_{[i}A_{j]k} + 2\delta_{k[i}\partial_{j]}A_{ll}\frac{c_2 - c_1}{1 + dc_2}, \tag{B.2}$$

which is gauge invariant under (4.63). The Hamiltonian is again given by (4.30) written in terms of the field strength above, and there is no longer any constraint on the parameters $g_2$ and $h_2$ in contrast to cases 1 and 2. The gauge invariant electric field strength now reads

$$\hat{F}_{0ij} = \dot{A}_{ij} - \partial_i\partial_j\phi + c_1\delta_{ij}\dot{A}_{kk} - c_2\delta_{ij}\partial^2\phi. \tag{B.3}$$

By integrating out the electric field from (4.57), we see that the Lagrangian in this case is given by

$$\mathcal{L}_3[A_{ij}, \phi] = \frac{1}{2g_1}\hat{F}_{0ij}\hat{F}_{0ij} - \frac{g_2}{2g_1(g_1 + dg_2)}(\hat{F}_{0ii})^2 - \frac{h_1}{4}\hat{F}_{ijk}\hat{F}_{ijk} - \frac{h_2}{2}\hat{F}_{ijj}\hat{F}_{ikk}. \tag{B.4}$$

As in case 2, we have assumed that $g_1 + dg_2 \neq 0$.

The case $c_1 = c_2 \neq -1/d$ has the same gauge transformations as used in the previous sections. In this case the magnetic field strength $\hat{F}_{ijk}$ is the same as in the undeformed case $F_{ijk}$ while the electric field strength can be written as

$$\hat{F}_{0ij} = F_{0ij} + c_1\delta_{ij}F_{0kk}, \tag{B.5}$$

where $F_{0ij}$ is the electric field strength of the undeformed theory. In this case the Lagrangian $\mathcal{L}_3$ becomes

$$\mathcal{L}_3[A_{ij}, \phi] = \frac{1}{2g_1}F_{0ij}F_{0ij} + \left[\frac{(1 + dc_1)^2}{2d(g_1 + dg_2)} - \frac{1}{2dg_1}\right](F_{0ii})^2 - \frac{h_1}{4}F_{ijk}F_{ijk} - \frac{h_2}{2}F_{ijj}F_{ikk}. \tag{B.6}$$

This theory is not essentially different from the undeformed theory (4.43b) studied in the previous sections. It is simply related by a redefinition of the parameter $g_2$. Alternatively, we can start with $g_2 = 0$ and generate its presence by deforming the Poisson bracket and Gauss constraint with $c_1 = c_2 \neq -1/d$.

Finally, we show that the case $c_1 \neq c_2$ and both different from $-1/d$ does not lead to a new theory either. To see this define

$$\check{A}_{ij} = A_{ij} + \frac{c_1 - c_2}{1 + dc_2}\delta_{ij}A_{kk}, \tag{B.7}$$

which transforms as

$$\delta\check{A}_{ij} = \partial_i\partial_j\Lambda. \tag{B.8}$$

Note that the gauge transformation of the Lagrange multiplier $\phi$ remains unchanged. In terms of this redefined gauge field, which transforms in the same way as the original, we can write the invariant magnetic field strength as

$$\hat{F}_{ijk} = 2\partial_{[i}\breve{A}_{j]k} \,, \tag{B.9}$$

while the electric field strength becomes

$$\hat{F}_{0ij} = \dot{\breve{A}}_{ij} - \partial_i\partial_j\phi + c_2\delta_{ij}(\dot{\breve{A}}_{kk} - \partial^2\phi) \tag{B.10a}$$
$$= \breve{F}_{0ij} + c_2\delta_{ij}\breve{F}_{0kk} \,, \tag{B.10b}$$

which is the same as (B.5) but with $c_1$ replaced with $c_2$, allowing us to conclude that this is also the same theory as the undeformed theory (4.43b).

## C   Stückelberging the dipole symmetry

Consider equation (7.5). We introduce a new gauge field $A_i$ which transforms as $\delta A_i = \partial_i\Lambda$ and we use this to build the gauge invariant field strength

$$\breve{F}_{ijk} = D_iA_{jk} - D_jA_{ik} - R_{ijk}{}^l A_l = F_{ijk} - R_{ijk}{}^l A_l \,. \tag{C.1}$$

We can write

$$\breve{F}_{ijk} = D_iA_{jk} - D_jA_{ik} - R_{ijk}{}^l A_l = D_i\left(A_{jk} - D_jA_k\right) - D_j\left(A_{ik} - D_iA_k\right)$$
$$= D_i\left(A_{jk} - D_{(j}A_{k)} - \frac{1}{2}F_{jk}\right) - D_j\left(A_{ik} - D_{(i}A_{k)} - \frac{1}{2}F_{ik}\right) \,, \tag{C.2}$$

where $F_{ij} = \partial_iA_j - \partial_jA_i$. Note that the combination

$$\hat{A}_{jk} = A_{jk} - D_{(j}A_{k)} \tag{C.3}$$

is gauge invariant, so one way of thinking about $A_i$ is as a Stückelberg field that removes the symmetric tensor gauge symmetry from the theory as we can now perform a field redefinition from $A_{jk}$ to $\hat{A}_{jk}$ and in this new theory the magnetic sector is a gauge theory for $A_i$ that contains a symmetric rank 2 tensor field $\hat{A}_{jk}$ which is not a gauge field. Furthermore, the electric field strength of the next section can be written as

$$F_{0ij} = \dot{\hat{A}}_{ij} - \frac{1}{2}\partial_0F_{ij} + D_i\left(\partial_0A_j - \partial_j\phi\right) \,, \tag{C.4}$$

which contains the electric field strength $\partial_0A_j - \partial_j\phi$, $F_{ij}$ and the non-gauge field $\hat{A}_{ij}$.

Using the identity

$$\partial_i\Phi\partial_j\Phi - \Phi D_i\partial_j\Phi + i\Phi^2A_{ij} = \mathcal{D}_i\Phi\mathcal{D}_j\Phi - \Phi\mathcal{D}_{(i}\mathcal{D}_{j)}\Phi + i\Phi^2\hat{A}_{ij} \,, \tag{C.5}$$

we see that also in the matter sector we can formulate things in terms of the non-gauge field $\hat{A}_{ij}$. Here the covariant derivatives $\mathcal{D}_i$ contains $A_i$ (as well as the Levi-Civita connection). We see that introducing the $A_i$ field amounts to Stückelberging the dipole gauge symmetry as there is now no longer an $A_{ij}$ gauge field, and hence this is the same as saying that there is no dipole gauge symmetry. So this is a non-solution to the problem of putting the theory on curved space. In order to recover dipole gauge symmetry on curved space one would have to reinstate the $\Sigma_i$ gauge symmetry but that is equivalent to setting $A_i = 0$ with $A_{ij}$ transforming as usual, which brings us back to square one.

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
