# Peer review of "Fractons, dipole symmetries and curved spacetime"

_SciPost Physics, doi:SciPost Phys. 12, 205 (2022)_

## Round 1 · Referee Report · Anonymous (Referee 1) · 2022-3-3

Strengths

1) Work is generally quite thorough

2) Coupling fractons to curved spacetime is novel

Weaknesses

1) Writing style is sometimes confusing and notation-dense

Report

In this paper, the authors carefully couple the traceful and traceless scalar charge higher-rank gauge theories and dipole-conserving matter theories to curved spacetime, as well as coupling the matter theories to the gauge theories. In the process, they give some classification results for dipole-conserving matter theories. Some of the classification results were informally in the “lore” of the field, but this paper has formalized them sharply (although I have one important piece of confusion about this part of the work – see second requested change). As the authors point out, these gauge theories were coupled to curved space in Ref. 31; the present generalization to curved Aristotelian spacetime is nontrivial. Although I find the writing occasionally confusing and notation-dense, the results appear interesting and suggest that it is important and useful to consider curved spacetime even in theories where there is no Lorentz or Galilean boost symmetry. It would be helpful for the authors to address the following points and questions in the "requested changes" section, but once those points are addressed, I recommend this work for publication. This work prompts a number of interesting follow-up directions.

Requested changes

1) The statement “The restricted mobility of isolated fracton particles can be viewed as a consequence of conservation of their dipole moment” is not particularly generic and should be modified. While dipole conservation is the simplest way to obtain fractonic mobility restrictions, it is not the only way, particularly in more exotic quantum fracton models like Haah’s code.

2) I find the introduction of Eq. (2.27) very confusing. Why have we restricted to this particular expression? Is this the most general expression that obeys the symmetries (2.1-2.2)? This is very important given that the rest of the classification results of section 2 come from this expression.

3) Could the authors clarify the consequences of the constraint Eq. (6.17) for dipolar symmetries? On a curved background, what does the $\Lambda$ corresponding to a dipolar symmetry look like? Is it clear when Eq. (6.17) is satisfied for a dipolar symmetry?

4) The results of Sec. 7 appear to contradict the results of Ref. 31. Ref. 31 states that, when considering only curved space, the traceful scalar charge theory in any dimension can only be gauge-invariant on flat space. The present paper’s Eq. 7.17 appears to allow Einstein manifolds. Could the authors explain the relationship between these results?

5) Is there any generalization of this work’s approach to theories where, for example, continuous rotational symmetry is relaxed to discrete rotation symmetry? Or theories with subsystem symmetries (e.g. the off-diagonal or “hollow” scalar charge theories where $A_{ij}$ has only off-diagonal components)?

6) Two typos: Top of page 21 – “..are described by $B_i = \partial_i \Lambda$$B_i$ should read $\Sigma_i$ . Also, broken reference just after Eq. 4.76c.

  • validity: high
  • significance: good
  • originality: good
  • clarity: ok
  • formatting: perfect
  • grammar: perfect

Author:  Emil Have  on 2022-06-02  [id 2547]

(in reply to Report 1 on 2022-03-03)

We are grateful to the referee for their time and their useful comments. Below we respond to the comments and suggestions of the referee report point by point. We hope that the improvements will satisfy the referee and we believe that they increase the quality of the paper.

  1. We agree with the referee that a conserved dipole moment is just one way of obtaining mobility restrictions. We have modified the sentence: "The restricted mobility of isolated fracton particles can be viewed as a consequence of conservation of their dipole moment:" to "For some theories the restricted mobility of isolated fracton particles can be viewed as a consequence of conservation of their dipole moment:"

  2. The Lagrangian in (2.27) is an example of a dipole-invariant term that appears at fourth order in derivatives and not the most general expression at that order. We took this to be the archetypical Lagrangian since similar expressions have appeared before in the literature. None of the subsequent results rely in a crucial way on the specific form of this Lagrangian: for the no-go theorem, we only use a specific form of the Lagrangian to show that it is possible to build a Gaussian theory that contains spatial derivatives when we allow for a non-linearly realised dipole symmetry. To clarify this point, we have added the following below Eq. (2.37): "The above shows that any theory of a complex scalar with a linearly realised dipole symmetry cannot simultaneously be Gaussian and contain spatial derivatives (i.e., gradient terms). If we allow for a non-linearly realised dipole symmetry, it is possible to build theories that are both Gaussian and contain spatial derivatives, as we illustrate by the following example." We also use this explicit Lagrangian when computing the symmetry algebra since it also has scale invariance, but any other choice of dipole invariant Lagrangian will lead to the same symmetry algebra as can be seen from the general expressions in (2.46) and (2.47), which are true for generic dipole-invariant Lagrangians. To emphasise this point, we have added the following sentence under (2.52): "At this stage, we emphasise that (2.52a) and (2.52b) are independent of the specific choice of dipole-invariant Lagrangian. Only scale invariance is a special property of the Lagrangian (2.43)."

  3. Eq. (6.17) is the general condition for there to be an analogue of global dipole symmetry on curved space. On a generic background, this will have no (non-trivial) solutions. Thus, whether there is a notion of global dipole symmetry on a given background has to be checked on a case-by-case basis and is a property inherent to the background itself.

  4. The interesting result of Reference [31] can be generalised by showing that the Lagrangian (7.17) is indeed gauge invariant and dependent on the trace of :math:` A_{ij}`. It would be nice to see if the argument in [31] can be generalised to include this result.

  5. Generalising our results to cases with discrete rotational symmetry remains an interesting open problem beyond the scope of the present work, but we expect similar techniques to work. To our knowledge, the corresponding geometry has, in contrast to Aristotelian geometry, not previously appeared in the literature.

  6. We thank the referee for pointing out these typos. The broken reference has been changed to a reference to Eq. (4.34).

---

## Round 1 · Referee Report · Anonymous (Referee 2) · 2022-4-1

Strengths

Solid results that a clearly correct.

Weaknesses

The only weakness I find is that the paper is written as a chapter in a textbook. It is hard to look for main results, which could have easily been summarized on a single page, and it is hard to chase some of the definitions to make sense of equations. I would recommend to add a section where the main equations are summarized and all notations needed to understand those are introduced. However, if the authors feel strongly about keeping the paper the way it is I will not object to the publication ``as is''.

Report

The authors explain the following issues:

(1) Gaussian fracton theories have Carroll symmetry. This observation is extremely simple, but, to the best of my knowledge, has not been stated previously and counts as new.

(2) They explain how to couple dipole conserving theories to Aristotelian geometry. They carefully write the covariant version of the complex scalar theory with dipole conservation. The main result is Eq.(65).

(3) They also show how to couple the tensor gauge theory to curved space and recover previous result about compatibility of the gauge structure with Einstein geometry. Incidentally, this was first done in 1712.06600 in 2 spatial dimensions.

These results are very good and should certainly be published.
  • validity: top
  • significance: good
  • originality: high
  • clarity: ok
  • formatting: good
  • grammar: perfect

Author:  Emil Have  on 2022-06-02  [id 2548]

(in reply to Report 2 on 2022-04-01)

We are grateful to the referee for their time and their useful comments. Below we respond to the comments and suggestions of the referee report point by point. We hope that the improvements will satisfy the referee and we believe that they increase the quality of the paper.

  • We have followed the suggestion of the referee and have added a "Summary of main results" at the end of the introduction.
  • We agree that this was also considered in the work 1712.06600. We have added a brief section 7.1.4 about the 2+1 dimensional case. We cite 1712.06600 as Ref. [32].

---

## Round 2 · List of Changes

Warnings issued while processing user-supplied markup:

  • Inconsistency: plain/Markdown and reStructuredText syntaxes are mixed. Markdown will be used.
    Add "#coerce:reST" or "#coerce:plain" as the first line of your text to force reStructuredText or no markup.
    You may also contact the helpdesk if the formatting is incorrect and you are unable to edit your text.

In addition to the changes mentioned in the replies to the referees, we have made the following improvements to the manuscript:

  • The paragraph on notation has been updated to include our conventions for the Riemann and Ricci tensors.
  • We have added a footnote and comment that point out that the worldline actions in Section 3 describe mobile dipoles.
  • The third bullet point in the list of conditions imposed on the Hamiltonian in Sec. 4.3 has been updated to: "Quadratic in :math:E_{ij}, so that we can integrate out :math:E_{ij} and obtain a Lagrangian that is second order in time derivatives."
  • We have included an analysis of the bounds on the coupling constants that make the Hamiltonian of the scalar charge gauge theory bounded from below in any :math:d\geq 3.
  • We have emphasised that the traceless scalar charge gauge theory as we define it is only non-trivial for :math:d\geq 3.
  • The argument in Sec. 5.2 that shows that the requirement of minimal torsion implies that :math:C_{\mu\nu}^\rho has been rephrased.
  • Sect. 7 has been significantly refined. Highlights of the changes made include:
  • Referring to the condition~(7.10) as ``Einstein'' in dimensions different from $d=3$ was inaccurate, and we have modified the language to reflect this.
  • We have added a new section (Sec. 7.1.3) that details the coupling of the traceless scalar charge theory to curved space.
  • When :math:d=2, there exists a traceless Chern--Simons-like theory that was discussed in Ref. [32]. We have added a new section (Sec. 7.1.4) that discusses the coupling of this theory to curved space, which reproduces the results of Refs. [31,32].
  • As an aid to the reader, we have included tables 1 and 2 which summarise the coupling of various scalar charge gauge theories to both curved space and spacetime in various dimensions.
  • The condition (7.39) that the Aristotelian background must satisfy previously included an arbitrary function :math:f. However, our choice of connection implies that this must be zero, and we have corrected the new version to take this into account.
  • We have specified the condition for the electric theory alone to be coupled to curved spacetime, which works out to be :math:v^\mu R_{\mu\nu\rho}{^\sigma} = 0 (see Section 7.2.2). As we now show, this amounts to the requirement that the Riemann tensor is entirely spatial.
  • The introduction, summary and discussion have all been changed to reflect the changes we have listed above.
  • What used to be appendix A has become appendix C to reflect the fact that it is not referred to before we refer to appendices A and B.

---

## Editorial Decision

published